# The Goldilocks of Pragmatic Understanding: Fine-Tuning Strategy Matters for Implicature Resolution by LLMs

**Laura Ruis**[*]
University College London

**Akbir Khan**
University College London

**Stella Biderman**
EleutherAI, Booz Allen Hamilton

**Sara Hooker**
Cohere for AI

**Tim Rocktäschel**
University College London

**Edward Grefenstette**
University College London

## Abstract

Despite widespread use of LLMs as conversational agents, evaluations of performance fail to capture a crucial aspect of communication: interpreting language *in context*—incorporating its pragmatics. Humans interpret language using beliefs and prior knowledge about the world. For example, we intuitively understand the response "I wore gloves" to the question "Did you leave fingerprints?" as meaning "No". To investigate whether LLMs have the ability to make this type of inference, known as an *implicature*, we design a simple task and evaluate four categories of widely used state-of-the-art models. We find that, despite only evaluating on utterances that require a binary inference (yes or no), models in three of these categories perform close to random. However, LLMs instruction-tuned at the example-level perform significantly better. These results suggest that certain fine-tuning strategies are far better at inducing pragmatic understanding in models. We present our findings as the starting point for further research into evaluating how LLMs interpret language in context and to drive the development of more pragmatic and useful models of human discourse.

## 1 Introduction

> User: "Have you seen my phone?"
> GPT-3: "Yes, I have seen your phone."

GPT-3's response[2] is a perfectly fine answer to the question, but a human might answer differently. They might respond "it's in your bag," bypassing the obvious follow-up question ("where is it?"). Giving such a helpful and efficient answer is an example of pragmatic language use that goes beyond the mere production of semantically plausible and consistent utterances. Meaning is not only determined by a combination of words, but also context, beliefs, and social institutions (Wittgenstein, 1953; Grice, 1975; Huang, 2017). Consider another exchange where Esther asks her friend Juan "Can you come to my party on Friday?" and Juan responds "I have to work". We resolve Juan's response as him declining the invitation by using the contextual commonsense knowledge that having to work on a Friday night precludes attendance. Both these exchanges contain an *implicature*—utterances that convey something other than their literal meaning.[3] Implicatures illustrate how context contributes to meaning; distinguishing writing and speaking from communicating (Green, 1996). We cannot fully

---

[*]Correspondence to `laura.ruis.21@ucl.ac.uk`

[2]Appendix D contains details on how this completion was obtained from text-davinci-002

[3]In Appendix E we present an introduction to implicature.

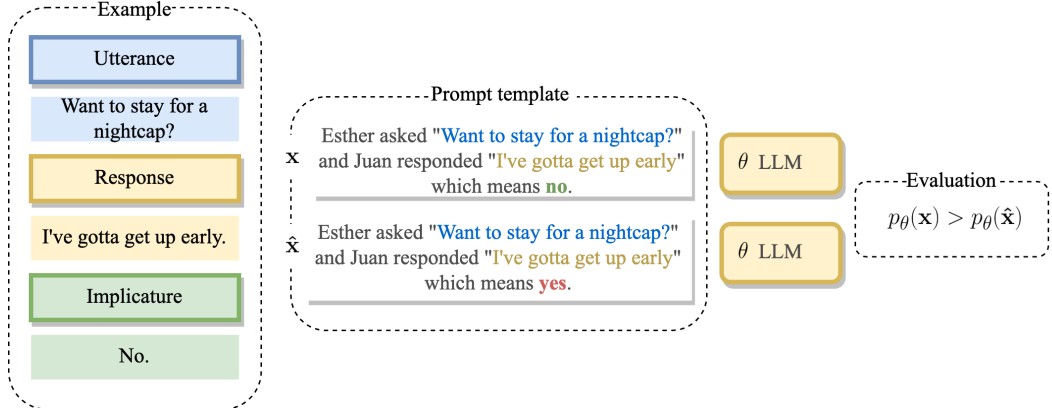

Figure 1: A schematic depiction of the protocol we propose to evaluate whether language models can resolve implicatures. Each example in the test set gets wrapped in templates and transformed into an *incoherent* example by swapping "yes" and "no". The model is said to resolve the implicature if it assigns a higher likelihood to the coherent text than the incoherent text.

understand utterances without understanding their implications. Indeed, the term "communication" presupposes the speaker's implications are understood by the addressee. Being able to resolve completely novel implicatures and, more broadly, engage in pragmatic understanding constitutes an essential and ubiquitous aspect of our every day use of language.

Large language models (LLMs) have demonstrated remarkable ability on a variety of downstream tasks such as planning (Huang et al., 2022), commonsense reasoning (Kojima et al., 2022), information retrieval (Lewis et al., 2020; Kim et al., 2022) and code completion (Austin et al., 2021; Biderman and Raff, 2022), to name a few. When fine-tuned with human feedback, LLMs obtain higher ratings on desiderata like helpfulness (Ouyang et al., 2022; Bai et al., 2022), and are proposed as conversational agents (Thoppilan et al., 2022). Despite the widespread use of LLMs as conversational agents, there has been limited evaluation of their ability to navigate contextual commonsense knowledge.

This raises an important question: *to what extent can large language models resolve conversational implicature?* To answer this question we use a public dataset of conversational implicatures and propose an evaluation protocol on top of it (Figure 1). We evaluate a range of state-of-the-art models that can be categorised into four groups; large-scale pre-trained models, like OPT (Zhang et al., 2022), LLMs fine-tuned on conversational data, like BlenderBot (Ng et al., 2019), LLMs fine-tuned on common NLP benchmarks with natural instructions for each benchmark, like Flan-T5 (Chung et al., 2022), and LLMs fine-tuned on tasks with natural instructions for each example, e.g. versions of OpenAI's InstructGPT-3 series[4]. Our results show that implicature resolution is a challenging task for LLMs. All pre-trained models obtain close to random zero-shot accuracy (around 60%), whereas humans obtain 86%. However, our results suggest that instruction-tuning at the example level is important for pragmatic understanding. Models fine-tuned with this method perform much better than others, and analysis of different model sizes shows that they have the best scaling properties. We further push performance for these models with chain-of-thought prompting, and find that one model in the group (GPT-4) reaches human-level performance. In summary, we conclude that pragmatic understanding has not yet arisen from large-scale pre-training *on its own*, but scaling analysis shows that it might for much larger scale. Fine-tuning on conversational data or benchmark-level instructions does not produce models with pragmatic understanding. However, fine-tuning on instructions at the example-level is a fruitful path towards more useful models of human discourse.

The **main contributions** of this work are: i) we motivate implicature understanding as a crucial aspect of communication that is currently mostly missing from evaluations of LLMs, ii) we design an implicature resolution task and propose a comprehensive evaluation protocol on which we evaluate both humans and LLMs to find that it poses a significant challenge for SotA LLMs, and iii) we provide a thorough analysis of the results and identify one fine-tuning strategy (instruction-tuning at the example-level) as a promising method that produces models with more pragmatic understanding.

---

[4]The precise method is unpublished and differs from the original instructGPT (Ouyang et al., 2022).

## 2   Related Work

LLMs have demonstrated remarkable performance on tasks for which they were not explicitly trained (Brown et al., 2020). Building on the hypothesis that these abilities arise due to implicit multitask learning (Radford et al., 2019), the recent works of Sanh et al. (2022) and Wei et al. (2022) explicitly train LLMs in a supervised multitask fashion, leading to models that are better zero-shot learners with fewer parameters. Besides rapidly saturating language understanding benchmarks (Kiela et al., 2021), these advancements make LLMs beneficial foundations for agents performing a plethora of tasks (Adolphs et al., 2022; Reed et al., 2022). The trend towards using these models as agents brings along with it increased urgency for alignment with human values (Kenton et al., 2021). However, larger models trained with next-word prediction are generally more toxic and unhelpful (Gehman et al., 2020; Bender et al., 2021; Lin et al., 2022). Recent work mitigates this with methods like prompting and finetuning on human-annotated outputs (Askell et al., 2021; Ouyang et al., 2022; Thoppilan et al., 2022). The produced models are more aligned on desiderata such as informativeness when evaluated by dedicated benchmarks and humans. We argue, however, that there is still something missing in these benchmarks. What is helpful and informative, as Kasirzadeh and Gabriel (2022) also point out, depends on the context in which a conversation is held. Consequently, any application that requires communicating with humans will rely on pragmatic communication skills—something that is not explicitly captured by the benchmarks used to evaluate the alignment of LLMs.

There is a large body of work that investigates the interplay between pragmatics and computational modeling (Cianflone et al., 2018; Schuster et al., 2020; Louis et al., 2020; Kim et al., 2021; Li et al., 2021; Jeretic et al., 2020; Parrish et al., 2021; Hosseini et al., 2023). Cianflone et al. (2018) introduce the task of predicting adverbial presupposition triggers, which are words like 'again' that trigger the unspoken presupposition that an event has happened before. Schuster et al. (2020) study the ability of computational models to do scalar inferences, finding that models use linguistic features to make pragmatic inferences. Kim et al. (2021) find that a substantial part of question-answering datasets contain questions that are unanswerable due to false presuppositions (i.e. "which linguist invented the lightbulb"). Hosseini et al. (2023) present a dataset for selecting entities with indirect answers, and find that language models adapted for this task get reasonable accuracy, but that there is room for improvement. The difference with this body of work and ours is that we look at the emergence of pragmatic understanding from large-scale language modeling. Jeretic et al. (2020); Parrish et al. (2021) are early works investigating the emergence of pragmatic understanding in pretrained language models, but they only look at scalar implicatures and presuppositions. Zheng et al. (2021) are the first to evaluate pretrained language models on conversational implicatures. This is important pioneering work highlighting the difficulty of implicature for language models, but their evaluations require task-specific training and the models they evaluate are relatively small. In contrast, our evaluation protocol is applicable out-of-the-box and is much more comprehensive, evaluating models up to 176 billion parameters and using in-context prompting. Additionally, Zheng et al. (2021) benchmark synthetic data whereas this work evaluates performance on naturally occurring implicatures (George and Mamidi, 2020). We believe this to be a better representation of the true distribution of implicatures in natural dialogue.

The standard set of benchmarks LLMs are evaluated on covers many tasks, but even though implicature is one of the most important aspects of language pragmatics (Levinson, 1983), it is only evaluated as part of BIG-bench (Srivastava et al., 2022). Unfortunately, the methodology used by the BIG-bench implicature task contributors has limitations, which call into question the validity of their claims. Firstly, the task contributors discard a subset of the data that is ambiguous according to them. In our view this defeats the point of the benchmark. Implicatures are a type of non-literal, ambiguous language the intended meaning of which humans often easily interpret; comparing the way humans and models do this is precisely what we are interested in. In turn, we expect performance on the BIG-bench task to overestimate the ability of LLMs to resolve naturally occurring implicatures. We keep this challenging subset of the data and instead use human evaluation to deal with examples that are too ambiguous to understand. Secondly, the difference in performance between their average and best rater is 18%, whereas for our evaluations this difference is 6%. This indicates their human evaluation is of low quality, but it is impossible to verify because there are no details available on how the annotation is done. Finally, BIG-bench uses only base LLMs and no SotA fine-tuning methods. In summary, we use a more challenging dataset, and in turn at least six times more evaluations per model, we provide higher-quality human annotations, and evaluate four different categories of LLMs to investigate which aspects of LLMs contribute to their performance on implicature understanding.

# 3 Evaluation Protocol

Here we outline the evaluation protocol we use to answer the research question "To what extent can LLMs resolve conversational implicature?". We focus on binary implicatures that imply "yes" or "no" (see Figure 1). We say a model resolves an implicature correctly if it assigns higher likelihood to a coherent utterance than a similar but incoherent one, detailed below.

**Zero-shot evaluation**. Consider the example from the introduction packed into a single utterance:

> Esther asked "Can you come to my party on Friday?" and Juan responded "I have to work", which means no.

We can transform this example to be *pragmatically incoherent* (in the sense that it will become pragmatically inconsistent with expected use) by replacing the word "no" with "yes":

> Esther asked "Can you come to my party on Friday?" and Juan responded "I have to work", which means yes.

To resolve the implicature, the model should assign higher likelihood to the first of the two sentences above, namely the most coherent one. Importantly, both sentences have exactly the same words except for the binary implicature "yes" or "no", making the assigned likelihood scores directly comparable. Formally, let the coherent prompt be $\mathbf{x}$ and the augmented, incoherent prompt be $\hat{\mathbf{x}}$. A model outputs a likelihood $p$ parameterized by weights $\theta$. We say a model correctly resolves an example $\mathbf{x}$ when it assigns $p_\theta(\mathbf{x}) > p_\theta(\hat{\mathbf{x}})$. This is equivalent to evaluating whether the model assigns a higher likelihood to the correct continuation of the two options. Note that this is a more lenient evaluation protocol than sometimes used for language models, where models are evaluated on on their ability to generate the correct continuation, in this case "no". The greedy decoding approach (evaluating whether "yes" or "no" is generated) is also captured by our approach, but we additionally label an example correct if "no" is not the highest assigned likelihood, but still higher than "yes". We did not opt for greedy decoding because "no" is not the only coherent continuation here, and marginalising over all possible correct continuations is intractable. The more lenient evaluation does capture implicature resolution, because the choice of "no" versus "yes" is only determined by the resolution of the implicature. We guide the models to output "yes" or "no" explicitly in three of the six prompt templates with instructions, such that we can estimate the effect of this guidance on performance. For two model classes (i.e. GPT-3.5-turbo and GPT-4) we do not have access to likelihoods, and for these models we take the greedy decoding approach, guiding the model to output "yes" or "no" explicitly in all prompts (see Table 6 in Appendix F).

We use a dataset of conversational implicatures curated by George and Mamidi (2020)[5]. It contains implicatures that, like in Figure 1, are presented in utterance-response-implicature tuples. Of these, 718 are binary implicatures that we can convert into an incoherent sentence. We randomly sample 600 examples for the test set and keep the remaining 118 as a development set to improve implicature resolution after pre-training through in-context prompting or fine-tuning.

**Few-shot in-context evaluation**. We add $k$ examples of the task to the prompt, e.g. with $k = 2$:

> Esther asked "Have you found him yet?" and Juan responded "They're still looking", which means no.
> Esther asked "Are you having fun?" and Juan responded "Is the pope Catholic?", which means yes.
> Finish the following sentence:
> Esther asked "Can you come to my party on Friday?" and Juan responded "I have to work", which means no.

We evaluate the models' $k$-shot capabilities for $k \in \{1, 5, 10, 15, 30\}$ by randomly sampling $k$ examples from the development set for each test example. We opt for a random sampling approach to control for two sources of randomness. Firstly, examples have different levels of informativeness. Secondly, recent work found that the order in which examples are presented matters (Lu et al., 2022). Ideally, to marginalise over these random factors, we would evaluate each test example with all

---

[5]Published under a CC BY 4.0 license.

permutations of $k$ examples from the development set. This requires $\frac{118!}{(118-k)!}$ evaluations for each test example, which is intractable. Instead, we estimate performance per test example by randomly sampling from the development set. In this way we control for some of the variance in performance, but avoid extra evaluations. We ensure each model sees the same few-shot examples per test example.

**Controlling for prompt sensitivity**. It has been shown language models are sensitive to prompt wording (Efrat and Levy, 2020; Tan et al., 2021; Reynolds and McDonell, 2021a; Webson and Pavlick, 2021). To control for this factor of randomness we manually curate six different template prompts and measure performance across these. One of the templates has been presented above, namely "Esther asked *<utterance>* and Juan responded *<response>*, which means *<implicature>*". Another template is: "Question: *<utterance>*, response: *<response>*, meaning: *<implicature>*". The former we call *natural* prompts and the latter *structured* prompts. Each group has three templates that only differ slightly in wording. This grouping allows us to look at the variance due to slight changes in wording as well as performance difference due to a completely different way of presenting the example. The full list of prompts can be found in Appendix F.

## 4 Experiments

The set of large language model classes we evaluate can be grouped into four distinct categories:

1. *Base models*: large-scale pre-trained models; RoBERTa (Liu et al., 2019), BERT (Devlin et al., 2018), GPT-2 (Radford et al., 2019), EleutherAI (Wang and Komatsuzaki, 2021; Black et al., 2022), BLOOM (BigScience, 2022), OPT (Zhang et al., 2022), Cohere's base models, and GPT-3 (Brown et al., 2020)

2. *Dialogue FT*: LLMs fine-tuned on dialogue, BlenderBot (Ng et al., 2019).

3. *Benchmark IT*: LLMs fine-tuned on tasks with natural instructions for each benchmark or "benchmark-level instruction-tuned models"; T0 (Sanh et al., 2022) and Flan-T5 (Chung et al., 2022).

4. *Example IT*: LLMs fine-tuned on tasks with natural instructions for each example or "example-level instruction-tuned models"; a subset of OpenAI's API models and Cohere's API models).

For Benchmark IT models, annotators write a single instruction for an entire dataset. The models are then fine-tuned on each example from the dataset with the same instruction. We distinguish this from example-level IT; for that type of fine-tuning each example in a dataset gets a new instruction, resulting in a more diverse dataset. Each group contains model classes for which we evaluate a range of sizes. A detailed categorization of the models and their attributes can be found in appendix G.[6] We make use of the OpenAI and Cohere APIs as well as the pretrained models in the transformers library (Wolf et al., 2020) and EleutherAI's framework to evaluate them (Gao et al., 2021). All code used for this paper can be found on GitHub[7] and the dataset is made publicly available on HuggingFace[8]. Below, we present zero-shot and few-shot results, discussing patterns of performance of the models in the four different groups. We further look at the results for different model sizes of each model class and the variance over the prompt templates. We contrast the models' performance with human performance. To this end, each test example gets annotated by five humans. We split the test set in four and assign each annotator a subset, leaving us with twenty annotators in total. The average human performance is 86.2%, and the best performance is 92%. Some of the errors humans make uncover examples that have multiple interpretations, and others uncover annotation errors. The nature of the task of implicature resolution means we do not expect models to perform better than human best performance. Details on the human experiment can be found in the Appendix H (also containing an analysis of human errors), and detailed results per model and prompt template in Appendix K.10. We also test for spurious correlations present in the benchmark (like lexical cues the model can rely on), and find no indication (Appendix K.8).

***Insight 1: Models instruction-tuned at the example level outperform all others.*** Table 1 shows the best 0-, 1-, and 5-shot accuracy each model class achieved on the implicature task. The best overall

---

[6]Note that there are several important aspects unknown for models behind APIs, like OpenAI's model sizes.

[7]`https://github.com/LauraRuis/do-pigs-fly`

[8]`https://huggingface.co/datasets/UCL-DARK/ludwig`

Table 1: The k-shot accuracy ($k \in \{0, 1, 5\}$) for the best performing model of each class. For each model, we select the model size to show by choosing the one that achieves the best 5-shot performance. The std is over prompt templates for the models and over annotators for humans. FT stands for fine-tuning and IT for instruction-tuning. We find that the models in the *Example IT* class obtain significantly higher performance than all others. ⋆ means size unknown.

| | Model | 0-shot | 1-shot | 5-shot |
|---|---|---|---|---|
| **Baselines and Toplines** | Random | | 50% | |
| | Human avg. | | 86.2% ± 2.3 | |
| **Base models** | BERT-110M | 54.8% ± 1.6 | 51.7% ± 1.7 | 53.3% ± 2.2 |
| | RoBERTa-355M | 55.6% ± 2.0 | 54.1% ± 0.9 | 57.1% ± 1.5 |
| | GPT-2-xl | 51.3% ± 2.9 | 57.4% ± 3.3 | 57.7% ± 1.1 |
| | EleutherAI-20B | 57.5% ± 3.3 | 55.9% ± 2.3 | 61.1% ± 4.9 |
| | BLOOM-176B | 54.2% ± 1.2 | 61.1% ± 2.7 | 65.4% ± 3.4 |
| | OPT-13B | 61.0% ± 5.5 | 60.6% ± 2.7 | 67.4% ± 2.1 |
| | Cohere-52B | 58.5% ± 4.0 | 63.0% ± 3.8 | 65.1% ± 2.9 |
| | GPT-3-175B | 57.7% ± 4.4 | 65.7% ± 1.4 | 68.7% ± 1.5 |
| **Dialogue FT** | BlenderBot-2.7B | 53.4% ± 0.3 | 53.3% ± 0.1 | 53.3% ± 0.1 |
| **Benchmark IT** | T0-11B | 55.6% ± 7.0 | 47.8% ± 0.5 | 47.0% ± 0.2 |
| | Flan-T5-11B | 60.8% ± 2.4 | 57.4% ± 5.0 | 61.7% ± 4.8 |
| **Example IT** | text-davinci-001-⋆ | 72.3% ± 2.8 | 72.7% ± 1.3 | 74.5% ± 1.0 |
| | text-davinci-002-⋆ | 70.6% ± 2.3 | 75.6% ± 2.8 | 79.6% ± 2.0 |
| | text-davinci-003-⋆ | 71.2% ± 2.8 | 74.3% ± 1.4 | 79.7% ± 0.6 |
| | ChatGPT-⋆ | 72.1% ± 5.9 | 75.1% ± 1.5 | 73.9% ± 6.3 |
| | GPT-4-⋆ | **81.8% ± 1.8** | **82.3% ± 1.4** | **82.0% ± 1.7** |
| | Cohere-command-52B | 60.2% ± 5.2 | 72.8% ± 1.3 | 75.4% ± 1.8 |

accuracy is achieved by GPT-4 (the size of this model is unknown) at $82.3\% \pm 1.4$. This leaves a gap of 3.9% with human average performance. All models in the class *Example IT* perform significantly better than any of the other models for all $k$, except Cohere-command-52b at 0-shot. This result is more clearly seen in Figure 2, where we present the average accuracy for each model group. The performance for the other model classes across $k$ ranges from 47.0% by BlenderBot-2.7b at $k = 5$ and 68.7% by GPT-3-175b at $k = 5$. Even though base models benefit from few-shot examples, their performance remains mostly closer to random than to humans for all $k$, showing a gap of at least 17.5% with the average human. We observe a decrease in performance for $k > 0$ for the group *Benchmark IT*. This is not surprising, as these kind of models are specifically fine-tuned to be better at zero-shot generalisation (Sanh et al., 2022; Chung et al., 2022). BlenderBot, in the group *Dialogue FT*, performs barely better than random for all $k$. We hypothesise that the lower performance which Cohere-command-52b achieves 0-shot is not due to a lack of implicature understanding, but due to a failure to calibrate the yes/no likelihoods without examples. For this model, we observe a sharp rise in performance from $k = 0$ to $k = 1$ (see Table 1 or Figure 2). Since it is unlikely that one example of an implicature induces pragmatic understanding, we hypothesise that few-shot prompting mostly serves to clarify the task format. We test this hypothesis in Appendix K.6 by repeating the 1- and 5-shot experiment with random labels for Cohere-command-52B and text-davinci-001. We find that the performance does not degrade, which confirms that the few-shot examples mainly serve to prime the model towards producing outputs following the yes/no structure.

***Insight 2: The results are robust to different prompt templates.*** As detailed in Section 3, each example in the test set is wrapped in six different prompt templates. The standard deviation in Table 1 and in Figure 2 shows the sensitivity to different prompt wording. The standard deviation ranges from 0.3 for BlenderBot to 7.0 for T0-11B. All in all, the sensitivity to prompt wording does not seem to be a problem for this task; when taking into account the confidence intervals the result remains that models in the group *Example IT* perform significantly better than all other models, but worse than humans. In Appendix K.4 another analysis is presented that shows how different prompt templates benefit from in-context examples. The takeaway from the analysis is that in-context prompting can mitigate the fact that some models are better at natural prompts and others better at structured prompts by improving performance on the type of prompt the model struggles with zero-shot. Again, when

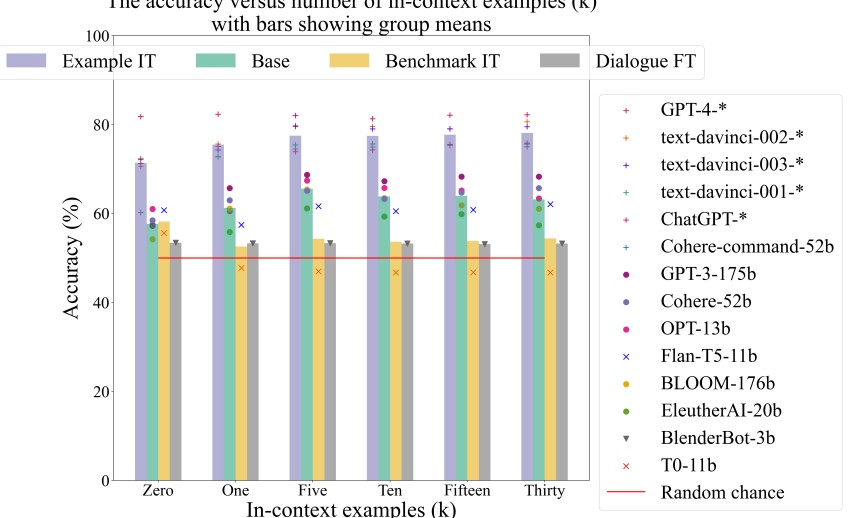

Figure 2: The few-shot accuracy for the best model of each class (e.g. the best performing model in the class Cohere-command is the 52b model, whereas the best model in the class OPT is the 13b model). The bars show the group means. Models fine-tuned on example-level instructions perform better than most other models, especially for $k > 0$. For all models there is a significant gap between best accuracy and human accuracy (which is 86.2%). * means size unknown.

Table 2: Scaling results for OpenAI's text-<engine>-001-series, for which we do not know the number of non-embedding parameters but do know the ordering in terms of size. The colors indicate whether going up in size (from left-to-right) increases performance significantly or not.

| Engine | Ada | Babbage | Curie | Davinci |
|--------|-----|---------|-------|---------|
| 0-shot | 56.5% ± 5.8 | 64.5% ± 1.8 (+8.0%) | 69.0% ± 2.9 (+4.5%) | 72.3% ± 2.8 (+3.3%) |
| 5-shot | 57.6% ± 2.8 | 66.1% ± 0.3 (+8.5%) | 71.3% ± 1.3 (+5.2%) | 74.5% ± 1.0 (+4.0%) |

only looking at the best prompt type for each model class (i.e. structured or natural), the results remain that models in the group *Example IT* perform best.

***Insight 3: Models instruction-tuned at the example-level have the most favourable scaling properties, but some base models also show positive correlation with scale.*** Figure 3 shows the scaling behaviour of the model classes for which we know the number of non-embedding parameters. We highlight 0- and 5-shot results, because for $k > 5$ the accuracy of most models plateaus (see Figure 2). However, detailed results for other $k$ can be found in Appendix K.10. Note that we do not know the number of parameters for OpenAI's 'text-<engine>-001'-series, but we do know the order of the engines in size, and we separately present its scaling results in Table 2. Except OpenAI's 'text-<engine>-001'-series, none of the models show significant performance increase with model size for the 0-shot evaluations. However, for $k$-shot evaluations with $k \geq 1$ we observe significant positive correlation with size for the models in the *Example IT* class for which we have multiple sizes (Cohere-command and 'text-<engine>-001') as well as some models in the base model class. Not only do the models in the *Example IT* class exhibit higher performance for the same model size, these models also have a steeper performance increase with size than the base models. Comparing the scaling properties of the best base model (GPT-3) with Cohere-command, we see that the increase in performance from the second-largest to the largest model is 0.04% per billion parameters from GPT-3-6.7B to GPT-3-175B and 0.15% per billion parameters for Cohere-command-6B to Cohere-command-52b (exact numbers used to calculate the slope can be found in Appendix K.10). If performance is linearly extrapolated from this curve GPT-3 reaches human-level performance at 642b parameters where Cohere-command would need 125b parameters.

***Insight 4: GPT-4 reaches average human-level performance with chain-of-thought prompting.*** For the model groups that benefit from in-context examples, we attempt to push performance further

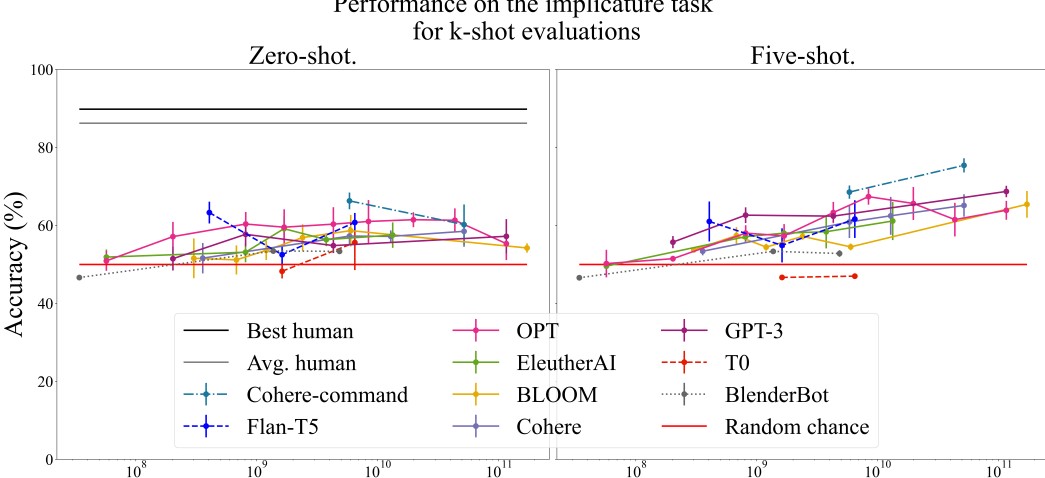

Figure 3: Scaling results for the model classes of which we know the number of non-embedding parameters. The error bars show standard deviation over prompt templates. Cohere's command models instruction-tuned at the example-level perform better than all other models. For all models there is still a significant gap between best accuracy and human accuracy.

Table 3: Results of the chain-of-thought (CoT) experiment for models in the group *Example IT*. The numbers between brackets show the difference in performance with the number on the same row one column to the left. Most models benefit from CoT-prompting, but not all. Additionally, GPT-4 reaches average human-level performance with CoT prompting. ★ means size unknown.

| Model | 0-shot | 5-shot | 5-shot CoT |
|---|---|---|---|
| text-davinci-001-★ | 72.3% ± 2.8 | 74.5% ± 1.0 (+2.2%) | 67.3% ± 2.6 (-7.2%) |
| text-davinci-002-★ | 70.6% ± 2.3 | 79.6% ± 2.0 (+9.0%) | 80.1% ± 0.8 (+0.5%) |
| text-davinci-003-★ | 71.2% ± 2.8 | 79.7% ± 0.6 (+8.5%) | 83.6% ± 0.6 (+4.0%) |
| ChatGPT-★ | 72.1% ± 6.0 | 73.9% ± 6.3 (+1.8%) | 77.2% ± 1.0 (+3.3%) |
| GPT-4-★ | 81.8% ± 1.8 | 82.0% ± 1.7 (+0.2%) | **86.5%** ± **1.0** (+4.5%) |
| Cohere-command-52b | 60.2% ± 5.2 | 75.4% ± 1.8 (+15.2%) | 75.3% ± 0.5 (-0.1%) |

with chain-of-thought prompting. We manually write a five-shot chain-of-thought prompt for all six prompt templates, and evaluate model performance using this prompt. One of the six chain-of-thought prompts can be found in Table 4 in Appendix F, and the other five are provided in the supplementary material. We only present the results for the group Example IT here, since CoT prompting did not improve performance for two of the base model classes we tried (see Appendix K.7). Consequently, we decided not to apply this experiment to the other models in the base group to save compute costs. The results of are shown in Table 3. We find that chain-of-thought prompting does not help for all models, but is nonetheless able to boost performance of GPT-4 to 86.5% ± 1.0. This is on-par with average human-level performance, and slightly below human best performance at 89.8%. To illustrate how explicit reasoning helps implicature understanding, we highlight a CoT generated by GPT-4 for an example from the dataset that models persistently get wrong. "A: Is there a bus I can get to the station? B: You can't rely on it". The implicature is yes, there is a bus, you just cannot rely on it. GPT-4 five-shot gets this wrong for all six templates. With CoT it gets it right for five of six templates. The generated CoT for one template is the following:

> Alice says 'You can't rely on it.' Alice must be implying that there is a bus, but it may not be dependable or timely. This means the response to Bob's question is yes, but with a caution about reliability. Answer: yes

More completions can be found in Appendix J.

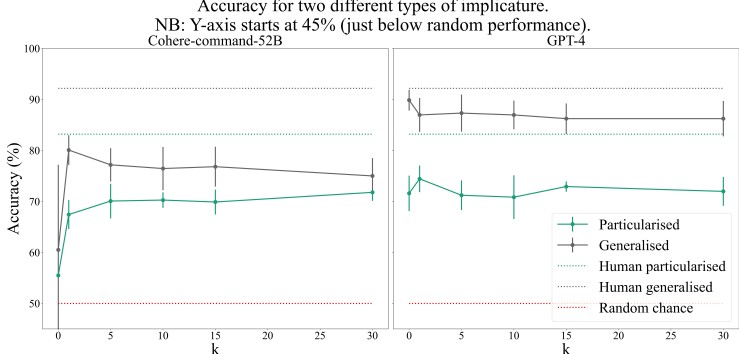

Figure 4: The accuracy v. k for the generalised and particularised examples obtained by the Example IT models Cohere-command and GPT-4. Particularised (context-heavy) examples are often significantly more difficult than generalised (context-free) examples for both models and humans. For most models, in-context prompting can mitigate this, but for others (like GPT-4), a significant gap remains. We see that Cohere-command-52b achieves similar performance as GPT-4 on the particularised examples, but significantly lower on the generalised examples.

Table 4: An example from the dataset for two types of implicature found in the test set. The rightmost column shows the amount of that type we manually found in the test set.

| Type | Example Utterance | Example Response | Impl. | # |
|---|---|---|---|---|
| Generalised | You know all these people? | Some. | No. | 47 |
| Particularised | Want to stay for a nightcap? | I've gotta get up early. | No. | 94 |

***Insight 5: Models often struggle with the same type of examples humans struggle with.*** We manually labeled 217 examples of the 600 examples in the test set according to a taxonomy. The remaining 383 examples do not fall as clearly within a category and are grouped together as type *other*. In Table 4 the two types of examples that occur frequently in the dataset are exemplified. *Generalised* implicatures require little or no context to be understood. They are the simplest type of example in the test set, and generally imply the same thing ("some" almost always implies "not all"). *Particularised* implicatures, by contrast, do require context to be resolved. For example, from Table 4, we need the context that it is undesirable to stay up late drinking when one has to get up early (see in Appendix E for more on generalised vs. particularised). In these type of examples, the context needed to resolve it is different every time. We label three other types of implicatures in the dataset, but since the analysis of these examples does not show significant patterns, we present it in Appendix K.9. We show the accuracy broken down per example type for two models from the Example IT group, as these patterns hold more broadly for almost all models evaluated (see the detailed results broken down per example type in Appendix K.9). Figure 4 shows that for lower $k$, the models often have a significantly worse performance for particularised examples than for generalised examples, just like humans do. For some, like Cohere-command-52b, this is mitigated by few-shot prompting, which brings particularised and generalised performance closer together (sometimes at the cost of generalised performance). For others, like GPT-4, the gap between particularised and generalised performance remains large for all $k$. From the bottom row in Figure 4 we observe that the edge GPT-4 has over Cohere-command-52b seems mostly driven by a higher accuracy on generalised examples. The accuracy on the particularised examples is comparable between those two models.

# 5 Discussion

In this study we use prompting to evaluate whether different groups of LLMs can resolve implicatures. In designing our experimental protocol, we carefully considered various alternatives, and here we discuss limitations of the chosen approach. Firstly, evaluating LLM competencies is inherently uncertain and sensitive to prompt choice. Nonetheless, we are confident our evaluation is comprehensive enough to assess implicature understanding: we apply six different prompt templates per test example, each used in three different prompting techniques (zero-shot, few-shot, chain-of-thought). Addition-

ally, in the appendix we present alternative zero-shot prompts and task specifications (Appendix K.3 and K.1 respectively), but since these did not improve performance they were not further considered. Another limitation is the fact that a subset of the models we evaluate are behind APIs. This means models are subject to change (affecting reproducibility) and certain details about these models are unknown. This affects the group instruction-tuned at the example-level, which is the group we find outperforms all others and has the most favourable scaling properties. How do we know instruction-tuning at the example-level is the main driver behind these findings without controlled A/B testing? Unfortunately, due to the secrecy surrounding the exact implementation of these models we cannot be certain, but we can be relatively confident. We evaluated ten models across six model classes and two APIs in the group example-level instruction tuned. Within this group, models probably vary significantly in other training and architecture details (especially Cohere-command models versus OpenAI models). The most salient commonality they share with each other and none of the other models is multi-task instruction-tuning at the example level, making it likely that this is the driving factor of their performance. A further datapoint in favour of this conclusion can be seen in Figure 3 (right); base models at similar scales as Example IT models perform significantly worse. We see that Cohere-command 52B significantly outperforms Cohere-base 52B, and the only difference between those models is instruction-tuning at the example level (Cohere-command is fine-tuned from Cohere-base). In fact, Cohere-command 52B outperforms other base models more than 3 times the size by a large margin (e.g. GPT-3 175B, BLOOM-176B, OPT-175B). We are therefore confident that instruction-tuning at the example-level is important for pragmatic understanding, an insight which can guide the development of open-source models capable of pragmatic understanding. Investigating the exact effect of this type of instruction-tuning on pragmatic understanding in a controlled setting is an interesting future work direction (e.g. by isolating the effect of data diversity from instructions). Another limitation is that some evaluations are subject to API stochasticity, which we address in Appendix K.5. After running the zero-shot experiment ten times through each API we conclude there is some stochasticity, but it is too small to impact our conclusions. We publish exact timestamps at which we queried APIs in Appendix L. Further, a downside of doing a comprehensive analysis on many models is compute costs. In Appendix M we publish a list of exact compute used (time and hardware), as well as estimated carbon emissions for each of the models that are not behind an API. Finally, the likelihood ranking approach we take limits our study to implicatures with clear alternative. However, implicatures in natural language can entail more complex propositions. For example, imagine Esther now asking "Can I use your stapler?" and Juan responding "Here's the key to my office.". Juan is implicating that (1) Esther can use the stapler, (2) the stapler is located in the office, and (3) the office is currently locked. This leaves ample room for the design of benchmarks with implicatures entailing multiple non-binary propositions.

## 6  Conclusion

LLMs have made remarkable progress on fluency and coherence in recent years. We argue however that a central aspect of language understanding is missing from evaluations. To understand language means to understand its pragmatics: its usage in a context that incorporates commonsense understanding, goals, objectives, and so on. We design a protocol that evaluates LLMs on binary implicature resolution and establish a significant gap with human understanding for SotA LLMs in three categories; large-scale pre-trained models, models fine-tuned on conversations, and models fine-tuned with benchmark-level instructions. By contrast, we find that models fine-tuned on example-level instructions perform significantly better. This group also exhibits the best correlation between accuracy and model size. Scaling analysis shows that for some large-scale pre-trained models accuracy also positively correlates with model size, but the best model in this group would need at least five times more parameters to reach similar performance. From these results, we conclude that instruction-tuning at the example level is important for pragmatic understanding. We hypothesise that there is something about the multi-task data diversity obtained from example-level instructions (i.e. each example a new task) that makes pragmatic understanding appear at smaller scale.

### Acknowledgements

We would like to thank members of the UCL DARK lab, members of the UCL NLP group, Pontus Stenetorp, Sebastian Borgeaud, Philipp Jettkant, Robert Kirk, and Max Bartolo for fruitful discussions and comments on an earlier draft of this paper. We would also like to thank the anonymous reviewers for engaging actively in discussion and providing feedback that has improved this work. This work was supported by the EPSRC Grant EP/S021566/1 and UCL International Scholar Award for Doctoral Training Centres.

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

# Appendices

## Contents

# A    Contributions

*Laura Ruis*: project proposal and leadership, dataset development, code writing, human experiment, manual error analysis, paper writing and editing.
*Akbir Khan*: code writing, model evaluations, human experiment, paper writing and editing.
*Stella Biderman*: model evaluations, compute usage, paper writing and editing, advisor.
*Sara Hooker*: compute usage, paper writing and editing, advisor.
*Tim Rocktäschel*: paper writing and editing, advisor.
*Edward Grefenstette*: initial idea, manual error analysis, paper writing and editing, advisor.

# B    Reproducibility Statement

We share all the data, human annotations, code used for the evaluations, and the raw results in the supplementary material. Additionally, in Appendix K.5 we estimate the variance due to stochasticity in the API's of OpenAI and Cohere. Of course, if either OpenAI or Cohere decides to change the models behind the API, the results might look different. We publish the exact date and time each API was queried for the results in Appendix L. Finally, in Appendix K.2 we estimate the variance over the prompt order of the in-context examples. The compute used for the experiments is detailed in Appendix M. The remaining experiments were done with API credits received as a research grant from OpenAI and Cohere.

# C    Ethics Statement

In this work, we conduct a study with human subjects (see Appendix H for details). To get matched with participants, we used the platform Prolific. Prolific complies with ethical standards according to UK law (e.g. complying with the GDPR). We compensated participants with a UK living wage at 15 GBP an hour, which is 6 GBP an hour more than Prolific recommends at 9 GBP per hour.
Implicature is an aspect of pragmatics, and pragmatic language impairments are universal in Autism Spectrum Disorder (ASD) (American Psychiatric Association, 2013). Difficulties in understanding scalar implicatures are claimed to be present in people with ASD (Volden, 2017), although the nature of the relation has proven hard to establish and has recently been debated (Katsos et al., 2011; Schaeken et al., 2018). For the purposes of this work, whether or not implicature understanding relates to ASD is not important. We took the following steps to make sure no sensitive data is collected or published. The human annotations we obtain are anonymous, related to a participant only by their Prolific ID for the purposes of compensation. In publishing the human annotations, we will not publish the Prolific ID of participants or anything else related to the participants. Additionally, we did not collect or request any personal or demographic characteristics of the participants apart from that they are all native English speakers.
Additionally, in this work we run a lot of compute-intensive experiments. We publish the estimated emissions per experiment in Appendix M. The total amount of GPU hours is estimated at maximally 966. How this is broken down per experiment can be seen in Appendix M.

# D Opener example with InstructGPT

This quote was obtained through the OpenAI playground for text-davinci-002 on May 15th 2023. The model text-davinci-001 consistently generates better responses for the same prompt. GPT-3 itself (i.e. davinci) mainly gives nonsensical answers. In the following, the prompt is *italic* and the completion **bold**. The completion was generated with a maximum of 10 tokens and a temperature of 0:

> *User: "Have you seen my phone?"*
> *GPT-3:* **"Yes, I have seen your phone."**

The opener example is used to introduce the problem we are looking at, and not to judge the model used to generate it. In fact, although text-davinci-002 sometimes completes conversations in a way that is unexpected according to pragmatic language usage, it is one of the better models when evaluated few-shot.

# E Background on implicature

The first influential consideration of implicature is Grice (1975). In his work, Grice continues the trend of moving away from purely logical accounts of language started by Wittgenstein (1921) by hypothesising implicatures arise in conversation when some mutually agreed upon maxims seem to be violated. For example, if we agree on only making relevant contributions to conversation, Juan's response in the introduction seemingly violates this maxim—after all, he starts talking about work when Esther asks him about a party. However, because Juan agreed to be relevant he must be implying that having to work means he cannot come to the party. Grice contrasts conversational implicatures that arise through context with conventional implicatures. These are implicatures where the *conventional* meaning of the word determines what is implicated. An example given by Grice is the following sentence: "he is an Englishman; he is therefore brave.". Grice notes that this sentence does not literally state that an Englishman being brave is a direct consequence of him being English, but it's implied by the conventional meaning of the word 'therefore'.

Since then, issues with the Gricean cooperative principle have been pointed out by many (Levinson, 1983; Sperber and Wilson, 1986; Davis, 1998; Lepore and Stone, 2014). The most influential alternative theory is relevancy theory by Sperber and Wilson (1986). They do away with the cooperative principle and instead theorise implicatures arise because speakers try to produce utterances that are both as relevant as possible and require the least effort to process. Another point of contention is the incorporation of conventional implicatures on the pragmatics side. Bach (1999) argues that there is no such thing as conventional implicatures, and they are simply instances of something else. Based on a thorough treatment of what Grice calls conventional implicatures, Bach argues all examples of it can be filed under other concepts within semantics, like utterance modifiers (called "utterance modifiers" instead of "sentence modifiers" because they go against the semantic content of the rest of the sentence). Potts (2005) also argues that to explain conventional implicatures we can stay on semantic turf. Indeed, even Grice himself says conventional implicatures derive from the meaning of the words, not from conversational context. However, Potts does not claim conventional implicatures do not exist, but instead argues they arise by a combination of lexical meaning and novel ways of combining words—the latter being the well-known principle of compositionality, an important part of semantics, not of pragmatics. Potts provides us with an illuminating demarcation between conventional and conversational implicatures. Conventional implicatures are never negotiable by context, whereas conversational implicatures are context-dependent and can always be cancelled without causing incoherent discourse. Consider again the sentence "he is an Englishman; he is therefore brave." and the sentence "Eddie has three bicycles" (implicating that Eddie has exactly three bicycles and not more). The former sentence can not be cancelled by new context without contradiction, whereas for the latter, if we continue saying "In fact, Eddie has 10 bicycles, he is a bicycle junkie", we have cancelled the implicature. This demarcation clearly puts conventional implicatures on the semantic side, and conversational implicatures on the pragmatic side. Potts goes on by providing a formal theory for conventional implicatures.

In later work, Potts (2006) describes how pragmatic pressures interacting with context cause conversational implicature to arise. He shows how sensitive conversational implicatures are to small changes in the context. Novel information about a speaker's belief state might completely change what is implied. There are many more models of implicature that aim to explain how humans understand language in context. Most notably, Frank and Goodman (2012) formalise the view that speakers produce utterances that are helpful and not longer than necessary with a Bayesian model called the rational speech act (RSA). Many variants on the RSA framework have since been proposed. For example, Goodman and Frank (2016) extend it to handle nonliteral uses of language, like irony, and metaphor. In the context of computational models, prior work uses insights from pragmatics to show that the use of certain words can make a language model produce biased completions (Patel and Pavlick (2021), e.g. saying someone "claimed" something rather than "said" something), and inform bias and sentiment classifiers (Greene and Resnik, 2009; Recasens et al., 2013).

In this work, we focus on conversational implicatures and not on conventional implicatures. All conversational implicatures are negotiable by context, but the way they depend on context can be different. Grice (1975) identifies generalised conversational implicatures and particularised conversational implicatures. The former require little or no context to be resolved. For example, "some athletes smoke" can imply "not all athletes smoke", but might also imply "I do not know whether all athletes smoke" when it is a response to the question "do you know whether all athletes

smoke?" (Davis, 2019). The latter only arise in certain contexts. For example, the response "I have an early morning" to the question "do you want to stay for a drink?".

# F Detailed prompt templates

Table 5 contains the full prompt templates we used for the main evaluation and Table 7 contains the extra prompt templates.

Table 5: *Ranking prompt templates*. The six templates we wrap the test examples in to present to the models. Template 1, 3, and 4 are of the type *structured*, and 2, 5, and 6 of the type *natural*. Within the type of prompt template they only differ slightly in wording.

| # | Prompt template |
|---|---|
| 1 | Does the following response to the question imply yes or no?

question: *<utterance>*
response: *<response>*
implicature: *<implicature>* |
| 2 | Finish the following text:

Esther asked "*<utterance>*" and Juan responded "*<response>*", which means *<implicature>* |
| 3 | Is the implied meaning of the following response yes or no:

question: *<utterance>*
response: *<response>*
meaning: *<implicature>* |
| 4 | What is the intent of the following response, yes or no?

question: *<utterance>*
response: *<response>*
intent: *<implicature>* |
| 5 | Finish the following text:

Karen asked "*<utterance>*" and William responded "*<response>*", which means *<implicature>* |
| 6 | Finish the following text:

Bob asked "*<utterance>*" and Alice responded "*<response>*", which means *<implicature>* |

Table 6: *Completion prompt templates*. The six adjusted templates we wrap the test examples in to present to the models when we use completion instead of likelihood ranking. Template 1, 3, and 4 are of the type *structured*, and 2, 5, and 6 of the type *natural*. Within the type of prompt template they only differ slightly in wording.

| # | Prompt template |
|---|---|
| 1 | Does the following response to the question imply yes or no? Only output 'yes' or 'no'. Even if you're uncertain, choose either 'yes' or 'no'. 

 question: *\<utterance\>* 
 response: *\<response\>* 
 implicature: *\<implicature\>* |
| 2 | Finish the following text. Only output 'yes' or 'no'. Even if you're uncertain, choose either 'yes' or 'no'. 

 Esther asked "*\<utterance\>*" and Juan responded "*\<response\>*", which means *\<implicature\>* |
| 3 | Is the implied meaning of the following response yes or no. Only output 'yes' or 'no'. Even if you're uncertain, choose either 'yes' or 'no'. 

 question: *\<utterance\>* 
 response: *\<response\>* 
 meaning: *\<implicature\>* |
| 4 | What is the intent of the following response, yes or no? Only output 'yes' or 'no'. Even if you're uncertain, choose either 'yes' or 'no'. 

 question: *\<utterance\>* 
 response: *\<response\>* 
 intent: *\<implicature\>* |
| 5 | Finish the following text. Only output 'yes' or 'no'. Even if you're uncertain, choose either 'yes' or 'no'. 

 Karen asked "*\<utterance\>*" and William responded "*\<response\>*", which means *\<implicature\>* |
| 6 | Finish the following text. Only output 'yes' or 'no'. Even if you're uncertain, choose either 'yes' or 'no'. 

 Bob asked "*\<utterance\>*" and Alice responded "*\<response\>*", which means *\<implicature\>* |

Table 7: The three additional templates we wrap the test examples in to present to the models, adapted from Glaese et al. (2022).

| # | Prompt template |
|---|---|
| 7 | The following text shows an interaction between two humans called Esther and Juan. In the interaction, Esther will ask Juan a question, and Juan will give an answer that contains an implicature. An implicature is an utterance that means something other than the literal meaning of the words. The implicature of Juan's response is yes or no. You, the AI assistant, are asked to finish the text with yes or no. The task begins:

Esther asked "*\<utterance\>*" and Juan responded "*\<response\>*", which means *\<implicature\>* |
| 8 | The following text shows an interaction between two humans called Esther and Juan. In the interaction, Esther will ask Juan a question, and Juan will give an answer that has a meaning besides the literal meaning of the words. That meaning is either yes or no. You, the AI assistant, are asked to finish the text with the correct meaning, either yes or no. The task begins:

Esther asked "*\<utterance\>*" and Juan responded "*\<response\>*", which means *\<implicature\>* |
| 9 | The following text shows an interaction between two humans called Esther and Juan. In the interaction, Esther will ask Juan a question, and Juan will give an answer that has a meaning besides the literal meaning of the words. That meaning is either yes or no. You, a highly intelligent and knowledgeable AI assistant, are asked to finish the text with the correct meaning, either yes or no. The task begins:

Esther asked "*\<utterance\>*" and Juan responded "*\<response\>*", which means *\<implicature\>* |

Table 8: *Chain-of-thought (CoT) prompt templates*. One of the six chain-of-thought prompt templates we use for the CoT experiment. Note that this is a 5-shot prompt. Each prompt variation contains five CoT examples. The other five variations are separately added to the supplementary materials

| # | Prompt template |
|---|---|
| 1 | Bob asks Alice a question, and Alice responds with an implicature. This means that Alice's response does not literally contain the answer to Bob's question, but implies an answer. Assuming that Alice is a cooperative conversational partner, what is the implicated answer to the question? For example:

Bob: You invented fire?
Alice: I told you that.
Implicature: Alice says 'I told you that'. Alice's response must be relevant to Bob's question because Alice is a cooperative conversational partner. Alice must mean that she told Bob that she invented fire. Alice's response to Bob's question 'You invented fire?' is yes.
Answer: yes

Bob: That cake looks delicious. Aren't you going to have some with me?
Alice: But that was a huge meal we just had.
Implicature: Alice's response must be relevant to Bob's question because Alice is a cooperative conversational partner. Alice must mean that the meal they just had was so huge she is not hungry anymore, and this must be relevant to Bob's question: 'Aren't you going to have some with me?' Alice's response to the question must therefore be no.
Answer: no

Bob: Could you perform well?
Alice: Being bilingual would help put me ahead of the pack.
Implicature: Alice says being bilingual would help put her ahead of the pack. Alice's response must be relevant to Bob's question because Alice is a cooperative conversational partner. Alice must be implying that she could perform well because she is bilingual. This means the response to Bob's question is yes.
Answer: yes

Bob: Have you any news for me?
Alice: I've made progress
Implicature: Alice says she has made progress. Alice's response must be relevant to Bob's question because Alice is a cooperative conversational partner. If Alice would not have any news for Bob, Alice would not have said she would have made progress because that would be misleading. The answer to Bob's question 'Have you any news for me?' must therefore be yes.
Answer: yes

Bob: You looked out for him?
Alice: He looked out for me. He taught me.
Implicature: Bob asks Alice 'You looked out for him?' and Alice's response says that the person that is being referred to by 'him' here looked out for Alice. If Alice meant yes to Bob's question, Alice would have said something like 'he also looked out for me'. Stating the response like this implies that the answer to Bob's question is no.
Answer: no

Only output a 'yes' or 'no' as a final answer. Write your reasoning after 'Implicature:' and then output either 'Answer: yes' or 'Answer: no'.

Bob: *<utterance>*
Alice: *<response>*
Implicature: |

# G Model categorization

Table 9 contains details on the model classes that are a part of each group of models we evaluate, along with their model sizes.

Table 9: Model categorization for each of the models. DL stands for dialogue, FT for fine-tuning, BL for benchmark-level, EL for example-level, and IT for instruction-tuning.

| Group | Model class | Model IDs | Model size | Instruct |
|-------|-------------|-----------|------------|----------|
| Base | BERT | base uncased | 110M | No |
| | RoBERTa | base, large | 125M, 355M | No |
| | GPT-2 | GPT-2 medium, large, xl | 354M, 774M, 1.6B | No |
| | EleutherAI | GPT-J, GPT-NeoX | 125M, 1.3B, 2.7B, 6B, 20B | No |
| | BLOOM | - | 560M, 1B1, 3B, 7B1, 176B | No |
| | OPT | - | 125M, 350M, 1.3B, 13B, 30B, 66B, 175B | No |
| | Cohere | small, medium, large, XL | 409.3M, 6.067B, 13.12B, 52.4B | No |
| | GPT-3 | ada, babbage, curie, davinci | Est. 350M, 1.3B, 6.7B, 175B | No |
| DL FT | BlenderBot | - | 90M, 2.7B, 9.4B | No |
| BL IT | T0 | - | 3B, 11B | Yes |
| | Flan-T5 | - | 780M, 3B, 11B | Yes |
| EL IT | Cohere-command | medium, xlarge | 6.067B, 52.4B | Yes |
| | text-davinci-001 | ada, babbage, curie, davinci-1 | Unknown, left-to-right increasing in size | Yes |
| | text-davinci-002 | - | Unknown | Yes |
| | text-davinci-003 | - | Unknown | Yes |
| | ChatGPT | gpt-3.5.turbo | Unknown | Yes |
| | GPT-4 | gpt-4 | Unknown | Yes |

# H Human evaluation

The participants for the human evaluation in this paper were recruited using Prolific (`www.prolific.co`). The setup of the experiment is as follows. We divide the test set of 600 examples into four non-overlapping subsets of 150 examples. Each set of 150 examples was given to five unique annotators. This means each example in the test set is labeled five times by different people, and we have in total twenty annotators for the whole test set (five different ones for each of the four subsets). The only constraint for the annotators is that they are native English speakers. In Figure 5 the screen shown to potential participants on Prolific is shown. Participants are paid 15 pounds an hour, which was the living wage at the time of the experiment and more than the 12 dollars an hour Prolific recommends. The total amount spent on the human evaluation is 236 pounds. This amount came to be from four subsets, each costing about 30 minutes to label per annotators, and having 5 annotators per subset: 15 * 4 * 0.5 * 5 = 150. The extra costs were for the annotator that didn't pass the attention check which we paid nonetheless, and for prolific as a platform.

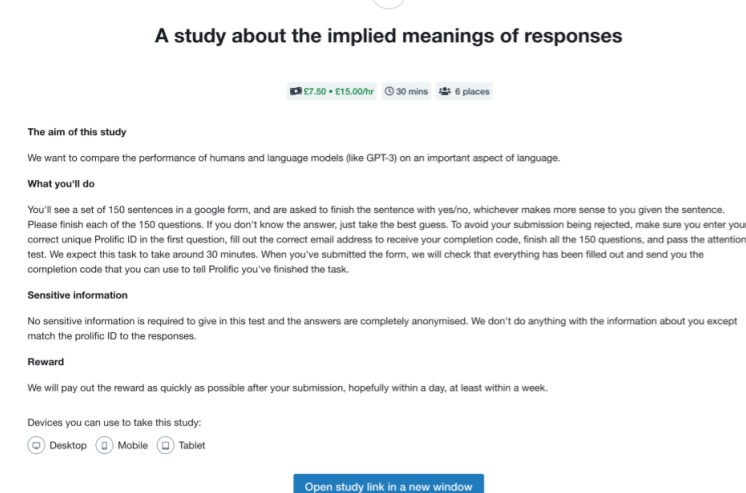

Figure 5: A screenshot of how the experiment is presented to potential annotators on Prolific (`www.prolific.co`).

The 150 test examples are wrapped in prompt template 2 (see Table 5) and presented in a Google form. We opted to wrap all examples in prompt template 2 to make the full human study directly comparable to the model's results on template 2. If we had done a mix of all templates we either had to spent six times as much on the human evalyations (which was not within our budget) or subsample evaluations, making it less comparable to part of the model study. Although models have been shown to be very sensitive to prompt wording, humans are less likely to perform differently for different prompt templates. All templates are coherent natural language that any native English speaker will understand. That said, this is speculative, and to confirm this hypothesis future work should investigate the effect of different wordings on implicature resolution by humans. The participants are asked to choose the correct continuation, yes or no (see Figure 6a). As recommended by Prolific, we subject the participants to an attention test (see Figure 6b). At three random places in the form, we add a question that does not contain an implicature and obviously maps to "yes". In this way, if the participants fails at least two of these questions, we can conclude they were not paying attention and remove their answers from the result. In practice, this happened once and we decided to pay the participant regardless, but discard their results, which were close to random.

Table 10 shows the performance of each annotator on the subset they annotated. The average human performance across subsets and annotators is 86.2% ± 2.3, the best performance is 89.8% ± 2.2, and the worst performance is 83.5% ± 1.5. The column "IAA" shows the average Cohen's Kappa coefficient which is the pairwise inter-annotator agreement for each annotator per subset. All agreements are substantial according to the interpretation guidelines for Cohen's Kappa (between 0.61–0.80).

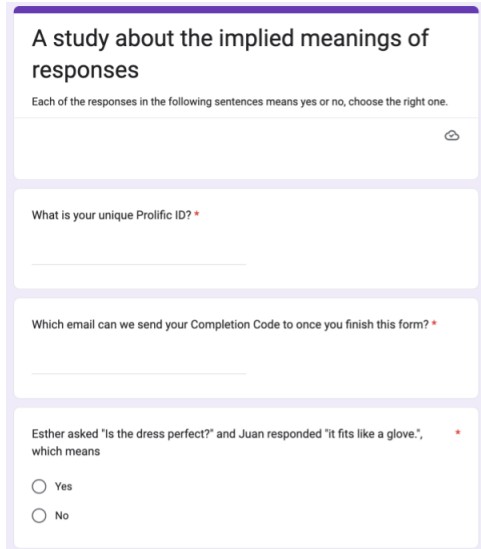
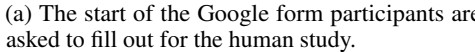

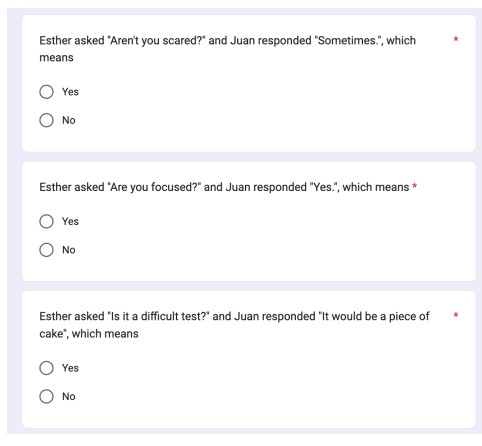

(a) The start of the Google form participants are asked to fill out for the human study.

(b) Part of the Google form the participants are asked to fill out. The second question in this image is part of the attention test. Juan's response does not contain an implicature but simply gives away the correct answer.

Figure 6: Screenshots of the Google form participants fill out as part of the implicature study.

Table 10: The performance of the human annotators on the subsets of the test set. Subset 1 through 4 are non-overlapping and cover the whole test set. Annotator X for subset Y might be a different human than annotator X for subset Z. IAA is the average pairwise inter-annotator agreement (Cohen's kappa coefficient) between annotators per subset.

| Annotator | 1 | 2 | 3 | 4 | 5 | Mean | Best | Worst | IAA |
|---|---|---|---|---|---|---|---|---|---|
| Subset 1 | 86.0% | 92.0% | 90.7% | 90.6% | 86.0% | 89.1% | 92.0% | 86.0% | 0.73 |
| Subset 2 | 84.7% | 83.3% | 87.3% | 86.0% | 86.0% | 85.5% | 87.3% | 83.3% | 0.64 |
| Subset 3 | 84.0% | 85.3% | 88.0% | 86.0% | 82.7% | 85.2% | 88.0% | 82.7% | 0.78 |
| Subset 4 | 85.3% | 82.7% | 84.0% | 82.0% | 92.0% | 85.2% | 92.0% | 82.0% | 0.71 |
| Total | - | - | - | - | - | 86.2% | 89.8% | 83.5% | 0.72 |
| Std | - | - | - | - | - | 2.3 | 2.2 | 1.5 | 0.1 |

**Human source of disagreement with ground-truth**. We do an analysis of the source of disagreement with the ground-truth. We explicitly do not call this error, as in some cases examples might allow multiple interpretations, and both could be right. In other cases, the ground-truth might be wrong.

*Annotation errors and multiple interpretations*: We analyse the examples for which most humans choose a different answer than the ground-truth. For 30 out of 600 examples in the test set, only one or zero people choose the same answer as the ground-truth. Of these examples, most are annotated wrongly (18 of 30). For example: 'Are you busy?', 'I'm drowning in work.', implicature: 'no'. Some are examples that can have multiple different interpretations (12 of 18), and the ground-truth answer likely just chooses one that is unnatural to humans. For example: 'You don't remember them?', 'Leave me alone!', implicature: 'yes'. 6 of the 30 examples are particularised, and 1 is generalised.

*Examples for which all humans agree with the ground-truth*: There are 409 out of 600 examples that all humans get correct. This set of examples contains most of the generalised implicatures (39 out of 47). These contain 58 out of 94 particularised examples.

*Examples most humans agree with the ground-truth*: When we look at examples that 3 or more humans got correct, that comprises most of the test set (530 of 600), and all of the generalised examples (47 of 47). This subset has 78 of 94 particularised examples, so for 16 particularised examples 3 or more humans disagree with the ground-truth.

# I Comparison with BIG-bench implicatures task

One of the BIG-bench tasks is related to the task in this work[9]. It uses the same underlying dataset we use (George and Mamidi, 2020). With the below we aim to discuss our contribution in light of the BIG-bench result. To summarise; the methodology used by the BIG-bench task contributors has limitations, which call into question the validity of their claims. Further, some of the BIG-bench results are irreproducible due to missing details in the tech report and the use of proprietary models. Considering this, our work is an important contribution validating the BIG-bench results in a reproducible and methodologically sound way, and above that providing insight into what aspects of LLM training are crucial for the ability to do pragmatic inferences.

**Limitations of the methodological approach of the task contributors in BIG-bench implicatures**. Our benchmark has 30% more data, which the BIG-bench task contributors discard. In this section we motivate the crucial importance of that data for evaluating implicature understanding (Section I.1), and why BIG-bench in turn might be overestimating the performance of LLMs on implicature resolution (Section I.2). Moreover, the human performance on the BIG-bench task indicates low quality human annotation, which we will also elaborate upon below, noting that this is impossible to verify because the BIG-bench report does not detail how the evaluation was done for this task (Section I.3).

## I.1 Discarding ambiguous examples

The BIG-bench task preprocesses the 1001 examples that George and Mamidi (2020) curate by keeping only yes/no questions, discarding any examples that are ambiguous according to the task contributors, and discarding remaining examples to create a 50/50 distribution of yes/no answers. This leaves them with 492 examples. Of these examples, 81 appear in our development set and the remaining 411 appear in our test set. Our test set has 600 examples, so BIG-bench effectively discarded 189 ambiguous examples compared to our test set; a bit more than 30% of the benchmark. To illustrate the importance of not discarding this data, we cherry picked a few examples that the BIG-bench authors discarded from the data.

- Utterance: "Can you lend me hundred dollars?", Response: "Is this supposed to be some kind of a joke?", Implicature: "No"
- Utterance: "Do you know, how long is Uncle Arthur staying with us?", Response: "Ask your father.", Implicature: "No"

Indeed, these examples are ambiguous. Asking whether the request for a hundred dollars is a joke does not literally mean you're saying no to the request. The response "ask your father" does not mean the speaker does not actually know, maybe they just do not want to respond. The humans in our study all infer the intended ground truth implicature. This shows a general property of implicatures; they are ambiguous, but often humans do infer the intended meaning. Ambiguity is not a discrete property. Some examples may be so vague that no one gets it. The following are examples the BIG-bench task discards that the humans in our study did struggle with:

- Utterance: "Got any more of those?", Response: "Nothing I'm at liberty to reveal here.", Implicature: "Yes"
- Utterance: "Have you finished sight-seeing?", Response: "Sorry. I should've come to see you first.", Implicature: "Yes"

In the first of these the implicature is "yes" because the person responding is implying that they do have more, they just cannot reveal them. Otherwise they would most likely simply say no. In the second example it feels more natural that someone says this when they are finished sight-seeing, otherwise they would've probably said something to the effect of "I'm still out, but I'm sorry..". In any case, humans in our study did not understand these responses like that. This illustrates another aspect of implicature; sometimes communication will go wrong over it. Removing implicatures that are ambiguous though, defeats the purpose of the task, as they are all ambiguous to a certain degree. The purpose of this study is to compare how humans resolve this type of non-literal language

---

[9]https://github.com/google/BIG-bench/tree/main/bigbench/benchmark_tasks/implicatures

compared to how models do it. The human baseline of 86% accuracy that humans achieve on our test set deals more naturally with examples that are too ambiguous for models to understand than discarding examples based on the subjective opinion of a few people.

## I.2 Overestimation of performance on implicature understanding

On the overlapping part of our test set and theirs the humans in our study achieve 92.8% accuracy. The best model on the BIG-bench task is PaLM, achieving a zero-shot performance of 64.4%. Note that this performance is on their full test set (not the overlapping part) and hence not directly comparable. Nonetheless, the missing examples are randomly sampled for our development set, and we can be pretty confident this number indicates a large gap with human performance. Two-shot PaLM comes very close to human performance with 91.7% accuracy, but of course this does not take into account the 189 more challenging examples that are part of our benchmark. Humans achieve 71.9% performance on this subset of ambiguous data, indicating that these data are more difficult than average, but nonetheless performance is higher than random. Without access to the models used to evaluate the BIG-bench task we cannot say anything for certain, but we expect the performance of PaLM to be overestimated as it does not account for a large part of the type of implicatures found in natural discourse; ambiguous ones.

## I.3 Other limitations

*Poor quality human evaluation*. The average human evaluator on BIG-bench implicatures achieves around 82% performance (where ours achieves on average 86% on a more challenging dataset), and their human best rater achieves 100% (where our human best is 92%). This difference between human average and best hints at poor quality average rating. This is impossible to verify because there is no information in the BIG-bench tech report on how the human evaluation was done exactly, or even which examples where evaluated.

*No fine-tuned models*. BIG-bench uses only base LLMs and no SotA fine-tuning methods. A question that remains is therefore, what aspects of LLMs contribute to their performance on implicatures? In our work we find that implicature performance emerges at a much smaller scale in models instruction fine-tuned at the example level, and that scale and prompting techniques are important.

## J Chain-of-thought completions by GPT-4

GPT-4 reaches human-level performance with CoT prompting. In this section we look at a few completions to help illuminate how CoT-prompting improves implicature resolution.

Specifically, we look at some of the examples for which GPT-4 5-shot gets 0% accuracy (i.e. wrong for all templates), and GPT-4 5-shot CoT gets 100% accuracy (i.e. right for all templates). This is a set of 10 examples. We only look at the first template, but all templates get a similar performance with CoT.

*Utterance*: 'Do you sell calculators?'
*Response*: 'We are all sold out. We wont probably have any more in for a few weeks.'
*Implicature*: Yes. But we dont have any calculators to sell now.

This is an example that GPT-4 gets wrong for all templates 5-shot, probably because the response has a negative sentiment, and also it means they do not sell calculators right now. However, the implicature is yes, because they do sell them normally. The CoT completion is:

*CoT completion*: 'Alice says they are all sold out and won't have any more in for a few weeks. Alice's response must be relevant to Bob's question because Alice is a cooperative conversational partner. Alice must be implying that they do sell calculators, but they currently don't have any in stock. The answer to Bob's question is yes. Answer: yes'

This example is similar to the one given in the main text in Section 4, and there are more like these in the dataset: the sentiment is negative, and an interpretation can be negative, but strictly the implicature is positive (yes).

*Utterance*: 'Do you know where he is?'
*Response*: 'I'll find him.'
*Implicature*: No.

GPT-4 gets this wrong for all templates 5-shot, and right for all of them with CoT:

*CoT completion*: 'Alice says she will find him. Alice's response must be relevant to Bob's question because Alice is a cooperative conversational partner. If Alice knew where he is, she would have provided that information directly. Instead, she says she will find him, implying that she does not currently know his location. The answer to Bob's question 'Do you know where he is?' must therefore be no. Answer: no'

It's unclear why GPT-4 gets this one wrong 5-shot (all humans get this one right), but the CoT is very relevant and helpful.

*Utterance*: 'It seems Cathy is not happy at all with her new room mate. Did you ask her why?'
*Response*: 'Believe me. I tried to find out. But I simply could not pin it down.'
*Implicature*: Yes. But Cathy did not give specific reasons for her feelings.

Again an example where the sentiment of the response seems to hint at no. Of 5 human annotators, also only 2 interpret this response as "yes". With a CoT, GPT-4 gets it right six times:

*CoT completion*: 'Alice says she tried to find out but couldn't pin it down. Alice's response must be relevant to Bob's question because Alice is a cooperative conversational partner. Alice must mean that she attempted to ask Cathy why she is unhappy with her new roommate, but was unable to get a clear answer. The answer to Bob's question 'Did you ask her why?' must therefore be yes. Answer: yes'

A helpful reasoning trace. All CoT completions by the models we have run CoT on are available in the GitHub: `https://github.com/LauraRuis/do-pigs-fly`.

# K  Additional results

## K.1  Contrastive experiment

In this section we reframe the implicature resolution task to a contrastive one, allowing the model to contrast the coherent to the incoherent sentence in a single prompt.

**Contrastive task**. In the ranking task the model is required to assign higher likelihood to the coherent utterance than the incoherent one ($p_\theta(\mathbf{x}) > p_\theta(\hat{\mathbf{x}})$). In assigning a likelihood to $\mathbf{x}$, the model has no knowledge of $\hat{\mathbf{x}}$, and vice-versa. We hypothesize that the task might become easier if we reformulate it as a contrastive task. Consider the following prompt $\mathbf{p}$.

> Which of the following sentences is coherent:
>
> A: Esther asked "Can you come to my party on Friday?" and Juan responded "I have to work", which means no.
>
> B: Esther asked "Can you come to my party on Friday?" and Juan responded "I have to work", which means yes.
>
> Answer:

We can now evaluate the models' ability to understand which is the coherent sentence by evaluating whether it assigns $p_\theta(A \mid \mathbf{p}) > p_\theta(B \mid \mathbf{p})$. Note that this can again be framed in a ranking task of assigning a higher likelihood to the coherent prompt. If we finish the above prompt $\mathbf{p}$ by adding "A" to make a coherent prompt $\mathbf{x}$ and "B" to make an incoherent prompt $\hat{\mathbf{x}}$ we can again formulate the task by $p_\theta(\mathbf{x}) > p_\theta(\hat{\mathbf{x}})$. The difference is that within both the coherent and the incoherent prompt, the model can contrast the coherent and incoherent utterance to each other. We randomise the assignment of A and B to the utterances.

We do a small experiment with the contrastive task with one of the best performing models overall, OpenAI's text-davinci-002, for $k = \{0, 1, 5\}$. We use two prompt templates and for each template try three different multiple choice answers: A and B like above, one and two, or the full text of the answer. For the last option the coherent prompt $\mathbf{x}$ would look as follows:

> Which of the following sentences is coherent:
>
> A: Esther asked "Can you come to my party on Friday?" and Juan responded "I have to work", which means no.
>
> B: Esther asked "Can you come to my party on Friday?" and Juan responded "I have to work", which means yes.
>
> Answer: Esther asked "Can you come to my party on Friday?" and Juan responded "I have to work", which means no.

Table 11: Performance on the implicature task framed contrastively by OpenAI's text-davinci-002. The mean and standard deviation are reported over two different prompt templates (template 1 and 2).

| k | Non-contrastive | Rank one, two | Rank A, B | Rank full text |
|---|---|---|---|---|
| 0 | 71.3% ± 1.75 | 53.9% ± 0.9 | 59.3% ± 1.3 | 48.9% ± 0.6 |
| 1 | 76.1% ± 2.6 | 59.4% ± 1.6 | 63.2% ± 2.0 | 66.9% ± 0.9 |
| 5 | 80.5% ± 2.3 | 61.4% ± 1.3 | 64.0% ± 1.3 | 67.9% ± 2.1 |

In Table 11, perhaps surprisingly, we can see that the contrastive task is much more difficult than the original ranking task. For $k = 0$, the result is random except for the prompt where the multiple choice options are A and B. For $k = \{1, 5\}$ the full text ranking does best, but is still significantly worse than the original ranking setup. Because of these disappointing results, we did not evaluate the other models contrastively. Future work must establish whether the contrastive setup is worse across all model classes and sizes.

## K.2 Variance over prompt ordering

As mentioned in Section 3 in the main text, models are sensitive to the ordering of the $k$ examples in the prompt. Instead of marginalising over this random factor by evaluating all possible prompt orderings, we randomly sampled an ordered set of examples from the development set for each test example. Throughout experiments, we kept this randomly sampled order the same, meaning if you re-run the 5-shot evaluation you get exactly the same orderings. The reason for this is that we want evaluate each model equally. In this section we ask how the performance chances for the best performing model if we select another random order. We do this for the 5-shot evaluation, because the results show that adding more in-context examples barely helps performance.

Table 12: Variance over prompt ordering for 5-shot evaluation per prompt template (P.T.) for text-davinci-002

| Seed | P. T. 1 | P. T. 2 | P. T. 3 | P. T. 4 | P. T. 5 | P. T. 6 | Mean |
|------|---------|---------|---------|---------|---------|---------|-------|
| 0 | 80.17 | 78.17 | 82.83 | 80.50 | 79.17 | 76.50 | 79.56 |
| 1 | 80.17 | 76.17 | 81.33 | 81.83 | 76.00 | 76.33 | 78.64 |
| 2 | 79.50 | 78.17 | 81.17 | 80.17 | 78.17 | 76.50 | 78.94 |
| mean | 79.94 | 77.50 | 81.78 | 80.83 | 77.78 | 76.44 | - |
| std | 0.31 | 0.94 | 0.75 | 0.72 | 1.32 | 0.08 | - |

Table 12 shows the results of this experiment. Some prompt templates seem to be more sensitive to prompt example ordering than others, but for none of them the variance is high enough to change any conclusions.

## K.3 Different zero-shot instruction prompts

There is a narrative around large language models that if they fail a task, it might be that the prompt was not the right one (through works like Reynolds and McDonell (2021b); Kojima et al. (2022)). The idea is that they can be prompted to simulate almost anything, if you set them up correctly. Because implicature resolution is a ubiquitous result of learning language, we hold the view that a model should be able to do this task if a prompt is given in coherent natural language. Nonetheless, in an additional effort to find the "let's think step-by-step" (Kojima et al., 2022) of zero-shot implicature resolution we try three more prompt templates. We evaluate a base large language model and two instructable models: GPT-3-175B, text-davinci-001, and text-davinci-002. The prompts we use are taken from recent work that proposes a dialogue agent trained with human feedback (Glaese et al., 2022), but adapted to the task of implicature resolution. The full prompts are presented in Table 7 and Table 13 shows the results. The new templates do not improve performance for any of these models. The variance over the prompt templates for text-davinci-002 is

Table 13: Zero-shot accuracy over three additional prompt templates for a base LLM and two instructable models.

| Model | Templates |
|-------|-----------|
| GPT-3-175b | 59.2% ± 4.5 |
| text-davinci-001-? | 66.1% ± 3.2 |
| text-davinci-002-? | 67.7% ± 9.6 |

high, and the best prompt template of these three does achieve a slightly higher accuracy than the others: 74.5%. These results do not change the picture sketched so far.

## K.4 The effect of in-context examples on sensitivity to prompt wording

Figure 7 shows the relative performance increase due to in-context prompting broken down per prompt template. For text-davinci-001, most templates benefit similarly from more in-context examples, except for template 1. Perhaps surprisingly, we see that this template already achieves a performance of 76.5% at the zero-shot evaluation and does not improve much with few-shot prompting. For Cohere-52B and OPT-175B we see a clear grouping between the structured prompts (dashed lines) and natural prompts (dotted lines). Cohere struggles significantly more with the structured prompts than with the natural prompts in the zero-shot evaluation, and few-shot prompting can mitigate that, lowering the standard deviation over prompt templates to 1.89 at $k = 30$ from 4 at $k = 0$. OPT benefits from prompting for the natural prompts, but not for the structured prompts.

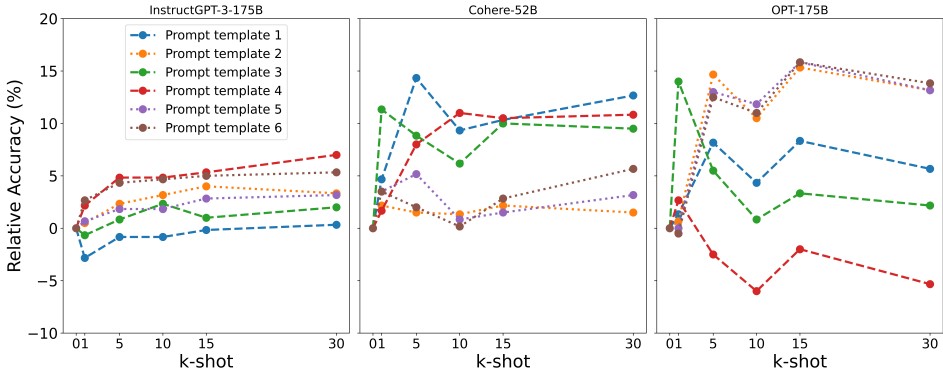

Figure 7: Relative performance increase over 0-shot due to in-context prompting. Structured prompt templates are dashed lines (1, 3, 4) and natural prompt templates dotted lines (2, 5, 6).

### K.5   Variance over API runs

In this section we comment on the reproducibility of research done using APIs. OpenAI and Cohere have their models behind an API, meaning we do not have control over what happens to the prompt before the model processes it. We run the zero-shot evaluation ten more times for two models of OpenAI and Cohere, text-davinci-002 and Cohere-52B. The results from this experiment are shown in Table 14 and 15. From this we can conclude that there is some stochasticity in the API that we have no control over, a bit more for OpenAI than for Cohere, but again we can be relatively confident that the conclusion will not be different because of it. The results from this work are therefore reproducible with access to the same models behind the API now. Unfortunately, when OpenAI or Cohere changes the models behind the API, these results are not exactly reproducible anymore.

For completeness, we add the timestamp that each result was obtained below (Appendix L).

Table 14: Results per prompt template (P.T.) for 10 different runs from text-davinci-002 for 0-shot evaluation.
Each evaluation has exactly the same text, so the variance in performance is due to API stochasticity.

| API-run | P. T. 1 | P. T. 2 | P. T. 3 | P. T. 4 | P. T. 5 | P. T. 6 | Mean |
|---|---|---|---|---|---|---|---|
| 0 | 73.50 | 68.83 | 73.00 | 71.17 | 67.17 | 68.83 | 70.42 |
| 1 | 73.83 | 69.00 | 72.83 | 71.50 | 67.67 | 68.33 | 70.53 |
| 2 | 73.67 | 68.67 | 73.17 | 71.33 | 67.50 | 68.50 | 70.47 |
| 3 | 73.83 | 68.17 | 73.17 | 71.00 | 67.67 | 68.17 | 70.33 |
| 4 | 73.67 | 68.83 | 73.33 | 71.17 | 67.00 | 68.33 | 70.39 |
| 5 | 73.83 | 68.50 | 73.00 | 71.00 | 67.00 | 68.17 | 70.25 |
| 6 | 73.67 | 69.00 | 73.00 | 71.17 | 67.33 | 68.50 | 70.44 |
| 7 | 73.67 | 68.67 | 72.83 | 71.33 | 67.50 | 68.67 | 70.44 |
| 8 | 73.83 | 69.17 | 72.83 | 71.17 | 67.33 | 68.00 | 70.39 |
| 9 | 73.50 | 68.50 | 72.83 | 71.00 | 67.50 | 68.67 | 70.33 |
| 10 | 73.67 | 69.50 | 73.00 | 71.33 | 67.50 | 68.50 | 70.58 |
| mean | 73.70 | 68.80 | 73.00 | 71.20 | 67.38 | 68.42 | - |
| std | 0.12 | 0.35 | 0.16 | 0.16 | 0.23 | 0.24 | - |

### K.6   Experiment with random in-context labels

This paper presents the thesis that instruction-tuning at the example level ("Example IT") is important for pragmatic understanding in LLMs. However, the 0-shot result that one of the models in the Example IT group achieves is similar to that of base models; Cohere-command-52b obtains a zero-shot performance of 60.2%. From the sharp rise in performance observed for the $k = 0$ to $k = 1$ result (from 60.2% to 72.8%) we hypothesise that the k-shot in-context examples in this task do not necessarily teach the model pragmatics in-context, but prime the model for the task format (namely,

Table 15: Results per prompt template (P.T.) for 10 different runs from Cohere-52B for 0-shot evaluation.
Each evaluation has exactly the same text, so the variance in performance is due to API stochasticity.

| API-run | P. T. 1 | P. T. 2 | P. T. 3 | P. T. 4 | P. T. 5 | P. T. 6 | Mean |
|---------|---------|---------|---------|---------|---------|---------|------|
| 0 | 56.00 | 62.67 | 54.33 | 54.00 | 62.17 | 62.17 | 58.56 |
| 1 | 56.00 | 62.83 | 54.33 | 54.00 | 62.33 | 62.33 | 58.64 |
| 2 | 56.00 | 62.83 | 54.33 | 54.00 | 62.17 | 62.33 | 58.61 |
| 3 | 56.00 | 62.83 | 54.33 | 54.00 | 62.17 | 62.33 | 58.61 |
| 4 | 55.83 | 62.67 | 54.33 | 54.00 | 62.17 | 62.33 | 58.56 |
| 5 | 56.00 | 62.83 | 54.33 | 54.00 | 62.17 | 62.17 | 58.58 |
| 6 | 56.00 | 62.83 | 54.33 | 54.00 | 62.17 | 62.17 | 58.58 |
| 7 | 56.00 | 62.67 | 54.33 | 54.00 | 62.33 | 62.17 | 58.58 |
| 8 | 56.00 | 62.83 | 54.33 | 54.00 | 62.00 | 62.33 | 58.58 |
| 9 | 56.00 | 62.83 | 54.00 | 53.83 | 62.17 | 62.17 | 58.50 |
| mean | 55.98 | 62.78 | 54.30 | 53.98 | 62.18 | 62.25 | - |
| std | 0.05 | 0.08 | 0.10 | 0.05 | 0.09 | 0.08 | - |

outputting either "yes" or "no" as detailed in Section 3 in the main text). If this hypothesis is true, we would observe similar performance regardless of whether the labels given in the prompt for the few-shot examples are true. We test this empirically for two base models (GPT-3, Cohere-52b) and two Example IT models (text-davinci-001, Cohere-command-52b) for 1-shot and 5-shot evaluation. The results can be found in Table 16. We find that for the Example IT models in-context prompts with random labels obtain the same results (i.e. within confidence intervals) as the experiments with ground-truth labels in the in-context examples. For base models however we do observe a drop in performance; for GPT-3-175b at 5-shot, and Cohere-52b both at 1- and 5-shot. Taken together, we can conclude that for base models the content of the in-context prompt seems important, whereas for models in the example IT group the in-context examples mainly serve as a primer for the task structure.

Table 16: The results of the 1- and 5-shot experiment with random labels for the few-shot examples as opposed to the the true labels. We find that performance does not degrade for the models in the Example IT group, which implies that for these models not the content of the examples is important for performance, but the structure.

| Model | 1-shot | 1-shot rand labels | 5-shot | 5-shot rand labels |
|-------|--------|--------------------|--------|--------------------|
| GPT-3-175b | $65.7\% \pm 1.4$ | $65.4\% \pm 1.2$ | $68.7\% \pm 1.5$ | $64.7\% \pm 1.9$ |
| Cohere-52b | $63.0\% \pm 3.8$ | $58.3\% \pm 3.3$ | $65.1\% \pm 2.9$ | $60.5\% \pm 1.9$ |
| text-davinci-001 | $72.7\% \pm 1.3$ | $73.9\% \pm 1.7$ | $74.5\% \pm 1.0$ | $73.4\% \pm 1.2$ |
| Cohere-command-52b | $72.8\% \pm 1.3$ | $72.0\% \pm 1.6$ | $75.4\% \pm 1.8$ | $73.5\% \pm 2.7$ |

Table 17: Results of the chain-of-thought (CoT) experiment for models in the base group. The numbers between brackets show the difference in performance with the number on the same row one column to the left. These models do not benefit from CoT-prompting. The reason Cohere-6b achieves such a low score for CoT-prompting is because it is not able to adhere to the correct output format (yes/no).

| Model | 0-shot | 5-shot | 5-shot CoT |
|---|---|---|---|
| GPT-3-350m | 51.5% ± 3.0 | 55.7% ± 1.6 (+4.2%) | 55.0% ± 3.5 (-0.7%) |
| GPT-3-1.3b | 57.7% ± 3.1 | 62.6% ± 2.0 (+4.9%) | 54.4% ± 5.8 (-8.2%) |
| GPT-3-6.7b | 54.8% ± 1.9 | 62.4% ± 1.5 (+7.6%) | 61.0% ± 2.3 (+4.0%) |
| GPT-3-175b | 57.2% ± 4.4 | 68.7% ± 1.5 (+11.5%) | 60.3% ± 4.2 (-8.4%) |
| Cohere-6b | 57.3% ± 2.2 | 60.9% ± 4.1 (+3.6%) | 29.2% ± 14.7 (-31.7%) |
| Cohere-52b | 58.5% ± 4.0 | 65.1% ± 2.9 (+6.6%) | 64.7% ± 3.2 (-0.4%) |

## K.7 Chain-of-thought on base models

In the main paper we do a CoT experiment on the models in the Example IT group. Base models also benefit from in-context examples, so it makes sense to also try CoT prompting on these models. After attempting this for two of the model classes in the group, we decided not to apply this prompting technique to the other models, because it decreases performance, sometimes significantly. See the results of the CoT experiment on the two base model classes in Table 17

## K.8 Testing for spurious correlations

In this section, we do a small scale experiment to test whether the benchmark has spurious correlations. Specifically, we run the benchmark with only the utterance or only the response as input. Strictly, getting the implicature right from the response only does not always indicate spurious correlations, as some examples only need the response (e.g. rhetorical questions like 'do pigs fly?'). Utterance-only results do always indicate spurious correlations. We run this experiment for GPT-3.5-turbo and GPT-4 0-shot and 5-shot (see Table 18 and Table 19).

Table 18: Results of running the benchmark with only the utterance as input, to test for spurious correlations with the label.

| Utterance-only | 0-shot | 5-shot |
|---|---|---|
| GPT-3.5-Turbo | 54.3% ± 3.3 | 41.7% ± 12.4 |
| GPT-4 | 48.9% ± 10.5 | 53.7% ± 0.5 |

Table 19: Results of running the benchmark with only the response as input, to test what part of the examples can be resolved without the utterance.

| Response-only | 0-shot | 5-shot |
|---|---|---|
| GPT-3.5-Turbo | 59.2% ± 4.7 | 58.3% ± 6.6 |
| GPT-4 | 62.6% ± 1.7 | 65.5% ± 1.1 |

We find that models mostly perform random for utterance-only, so spurious correlations do not seem to be an issue. For response-only, GPT-4 5-shot gets 65% accuracy. Some examples it gets right are: "do fish swim?" and "let's hope so".

Table 20: An example from the dataset for each type of implicature found in the test set. The rightmost column shows the amount of that type we manually found in the test set.

| Type | Example Utterance | Example Response | Impl. | # |
|------|-------------------|------------------|-------|---|
| Generalised | You know all these people? | Some. | No. | 47 |
| Particularised | Want to stay for a nightcap? | I've gotta get up early. | No. | 94 |
| World knowledge | Did you leave fingerprints? | I wore gloves. | No. | 23 |
| Idiom | Would he fire me? | He's all bark and no bite. | No. | 42 |
| Rhetorical question | Can you drive that far? | Can fish swim? | Yes. | 11 |
| Other | - | - | - | 383 |

## K.9 Detailed results type label analysis

In the main paper we do an analysis of two types of examples that occur frequently in the dataset, namely generalised and particularised implicatures. Here, we detail the full taxonomy of types of examples occurring in the dataset and report detailed results for each type per model (see 21 until Table 34 below). In Table 20 the full taxonomy of the examples is shown, representing types of examples that occur frequently in the dataset. We manually labeled 217 examples of the 600 examples in the test set according to this taxonomy. The remaining 383 examples do not fall as clearly within a category and are grouped together as type *other*. *Generalised* implicatures require little or no context to be understood. They are the simplest type of example in the test set, and generally imply the same thing ("some" almost always implies "not all"). *Particularised* implicatures, by contrast, do require context to be resolved. For example, from Table 20, we need the context that it is undesirable to stay up late drinking when one has to get up early (see in Appendix D more on generalised vs. particularised). The type *world knowledge* requires knowledge of the physical world to be resolved. From the example in Table 20; we need to know that you cannot leave fingerprints when wearing gloves to resolve this implicature. *Idiom* types contain an idiom or a metaphor that one needs to know or understand to resolve the implicature, and finally *Rhetorical question* types contain a question like "Is the Pope Catholic?", often requiring factual knowledge to be resolved.

The following tables contain the detailed results broken down per example type: Table 21 - Table 34. The most interesting pattern in this data is that for almost all models, even the best model (GPT-4 30-shot in Table 32), there is a significant gap between human-level performance on the particularised examples. This gap is larger than the gap for the other labels usually. Few-shot prompting can often mitigate this (e.g. for GPT-3-175b, Cohere-52b, and text-davinci-002), but not always (e.g. for GPT-4 the gap remains large for $k = 30$). However, for GPT-4, chain-of-thought can mitigate the gap as seen in Table 34. Where GPT-4 30-shot obtains 71.97% accuracy on the particularised examples (and humans 83.18%), GPT-4 with 5-shot CoT achieves 81.63%, which is close to human-level. We find that the particularised examples mostly benefit from CoT prompting. Namely, for the generalised type of examples, GPT-4 30-shot already achieves 86.23% accuracy and CoT improves this to 88.66%, which is a much smaller improvement than for the particularised examples.

## K.10 Detailed results per model

This section contains the results used for the zero-shot and few-shot evaluation in the main text in Section 4, broken down per prompt template. See Table 35 until Table 84.

Table 21: Accuracy per label for 0-shot evaluation.

| Model | Mean | World knowledge | Idiom | Rhetorical question |
|---|---|---|---|---|
| OPT-125m | 50.92 | 50.00 +/- 2.17 | 51.52 +/- 9.96 | 57.58 +/- 10.05 |
| OPT-350m | 57.14 | 57.97 +/- 10.25 | 64.77 +/- 3.65 | 65.15 +/- 3.39 |
| OPT-1.3b | 60.36 | 60.14 +/- 5.84 | 68.94 +/- 5.52 | 59.09 +/- 4.55 |
| OPT-2.7b | 59.56 | 60.87 +/- 6.15 | 67.05 +/- 2.18 | 69.70 +/- 6.78 |
| OPT-6.7b | 60.33 | 59.42 +/- 6.95 | 59.47 +/- 2.04 | 53.03 +/- 19.93 |
| OPT-13b | 61.03 | 63.77 +/- 14.78 | 73.86 +/- 7.51 | 66.67 +/- 16.32 |
| OPT-30b | 61.47 | 65.94 +/- 10.48 | 62.88 +/- 8.05 | 74.24 +/- 6.25 |
| OPT-66b | 61.33 | 69.57 +/- 13.75 | 60.23 +/- 4.30 | 59.09 +/- 18.74 |
| OPT-175b | 55.33 | 55.07 +/- 5.42 | 54.55 +/- 9.19 | 63.64 +/- 21.64 |
| BLOOM-560m | 51.58 | 54.35 +/- 5.47 | 54.92 +/- 16.72 | 50.00 +/- 13.64 |
| BLOOM-1b1 | 51.17 | 50.00 +/- 2.17 | 50.38 +/- 11.77 | 53.03 +/- 12.22 |
| BLOOM-1b7 | 53.61 | 52.17 +/- 6.15 | 53.79 +/- 8.77 | 68.18 +/- 6.94 |
| BLOOM-3b | 56.89 | 54.35 +/- 6.02 | 59.85 +/- 4.48 | 63.64 +/- 5.25 |
| BLOOM-7b1 | 58.67 | 63.77 +/- 14.57 | 68.94 +/- 5.82 | 68.18 +/- 4.55 |
| BLOOM-176b | 54.22 | 55.07 +/- 7.39 | 50.38 +/- 11.01 | 62.12 +/- 9.70 |
| EleutherAI-125m | 51.89 | 56.52 +/- 9.72 | 52.65 +/- 8.84 | 63.64 +/- 5.25 |
| EleutherAI-1.3b | 53.14 | 51.45 +/- 3.90 | 53.03 +/- 11.19 | 62.12 +/- 3.39 |
| EleutherAI-2.7b | 59.17 | 60.14 +/- 13.38 | 65.91 +/- 3.94 | 68.18 +/- 4.55 |
| EleutherAI-6b | 56.36 | 57.25 +/- 7.28 | 56.06 +/- 8.87 | 50.00 +/- 17.99 |
| EleutherAI-20b | 57.53 | 51.45 +/- 3.90 | 67.80 +/- 5.93 | 72.73 +/- 5.25 |
| Cohere-409m | 51.61 | 52.17 +/- 4.35 | 53.41 +/- 11.94 | 54.55 +/- 12.86 |
| Cohere-6b | 57.28 | 55.80 +/- 5.28 | 60.23 +/- 5.98 | 72.73 +/- 9.09 |
| Cohere-13b | 57.19 | 59.42 +/- 4.81 | 54.55 +/- 10.82 | 48.48 +/- 10.05 |
| Cohere-52b | 58.50 | 60.14 +/- 13.61 | 65.15 +/- 3.86 | 74.24 +/- 11.03 |
| GPT-3-350m | 51.47 | 51.45 +/- 3.90 | 53.41 +/- 13.56 | 50.00 +/- 13.64 |
| GPT-3-1.3b | 57.72 | 61.59 +/- 11.06 | 64.39 +/- 4.08 | 65.15 +/- 3.39 |
| GPT-3-6.7b | 54.83 | 54.35 +/- 6.99 | 53.79 +/- 7.61 | 62.12 +/- 3.39 |
| GPT-3-175b | 57.22 | 55.80 +/- 7.28 | 68.94 +/- 5.19 | 77.27 +/- 8.70 |
| T0-3b | 48.25 | 54.35 +/- 4.86 | 42.42 +/- 4.29 | 36.36 +/- 0.00 |
| T0-11b | 55.61 | 60.14 +/- 6.84 | 54.92 +/- 14.93 | 36.36 +/- 0.00 |
| BlenderBot-90m | 46.64 | 52.17 +/- 0.00 | 38.64 +/- 0.00 | 36.36 +/- 0.00 |
| BlenderBot-3b | 53.44 | 47.83 +/- 0.00 | 61.36 +/- 1.31 | 63.64 +/- 0.00 |
| BlenderBot-9b | 53.36 | 52.17 +/- 6.64 | 60.98 +/- 4.81 | 63.64 +/- 0.00 |
| Flan-T5-780m | 63.31 | 72.46 +/- 4.10 | 71.97 +/- 5.82 | 54.55 +/- 13.89 |
| Flan-T5-3b | 52.50 | 50.72 +/- 6.95 | 51.89 +/- 4.23 | 42.42 +/- 8.57 |
| Flan-T5-11b | 60.78 | 65.94 +/- 5.84 | 72.35 +/- 7.59 | 65.15 +/- 6.25 |
| Cohere-command-6b | 66.31 | 72.46 +/- 7.80 | 78.41 +/- 4.30 | 37.88 +/- 3.39 |
| Cohere-command-52b | 60.22 | 66.67 +/- 10.85 | 63.64 +/- 10.33 | 77.27 +/- 6.94 |
| text-ada-001-unknown | 56.50 | 63.77 +/- 4.10 | 58.71 +/- 16.04 | 51.52 +/- 10.05 |
| text-babbage-001-unknown | 64.47 | 67.39 +/- 6.02 | 76.52 +/- 1.69 | 60.61 +/- 10.05 |
| text-curie-001-unknown | 68.94 | 76.81 +/- 3.24 | 76.89 +/- 2.76 | 54.55 +/- 12.86 |
| text-davinci-001-unknown | 72.31 | 84.78 +/- 7.43 | 78.79 +/- 4.08 | 59.09 +/- 13.64 |
| text-davinci-002-unknown | 70.58 | 82.61 +/- 9.05 | 75.38 +/- 3.05 | 57.58 +/- 16.32 |
| text-davinci-003-unknown | 71.25 | 86.96 +/- 13.28 | 72.35 +/- 7.35 | 48.48 +/- 8.57 |
| ChatGPT-unknown | 72.08 | 82.61 +/- 12.04 | 83.33 +/- 5.97 | 56.06 +/- 16.11 |
| GPT-4-unknown | 81.78 | 92.03 +/- 2.99 | 90.91 +/- 3.21 | 84.85 +/- 8.57 |
| Humans | 86.23 | 93.04 | 92.73 | 92.73 |

Table 22: Accuracy per label for 0-shot evaluation.

| Model | Mean | Particularised | Generalised | Other |
|---|---|---|---|---|
| OPT-125m | 50.92 | 49.43 +/- 5.52 | 55.07 +/- 21.10 | 50.56 +/- 1.33 |
| OPT-350m | 57.14 | 47.92 +/- 4.37 | 69.20 +/- 8.36 | 56.68 +/- 4.99 |
| OPT-1.3b | 60.36 | 51.52 +/- 6.81 | 74.64 +/- 2.05 | 59.65 +/- 3.51 |
| OPT-2.7b | 59.56 | 50.19 +/- 5.06 | 69.93 +/- 5.53 | 59.22 +/- 6.14 |
| OPT-6.7b | 60.33 | 52.27 +/- 5.90 | 75.36 +/- 2.71 | 60.77 +/- 6.13 |
| OPT-13b | 61.03 | 55.49 +/- 8.79 | 75.00 +/- 5.72 | 58.79 +/- 5.51 |
| OPT-30b | 61.47 | 54.55 +/- 4.15 | 71.38 +/- 5.94 | 61.11 +/- 2.12 |
| OPT-66b | 61.33 | 55.11 +/- 7.33 | 69.93 +/- 12.26 | 61.46 +/- 3.68 |
| OPT-175b | 55.33 | 54.17 +/- 7.70 | 58.33 +/- 18.51 | 55.12 +/- 4.21 |
| BLOOM-560m | 51.58 | 50.76 +/- 5.59 | 48.91 +/- 25.22 | 51.59 +/- 3.50 |
| BLOOM-1b1 | 51.17 | 50.57 +/- 6.32 | 53.26 +/- 27.40 | 51.16 +/- 2.41 |
| BLOOM-1b7 | 53.61 | 50.38 +/- 7.95 | 59.78 +/- 18.82 | 53.23 +/- 1.62 |
| BLOOM-3b | 56.89 | 51.70 +/- 8.27 | 67.39 +/- 10.35 | 56.46 +/- 4.39 |
| BLOOM-7b1 | 58.67 | 46.59 +/- 2.86 | 79.35 +/- 2.74 | 57.11 +/- 4.03 |
| BLOOM-176b | 54.22 | 54.73 +/- 10.61 | 60.14 +/- 16.73 | 53.57 +/- 1.65 |
| EleutherAI-125m | 51.89 | 50.38 +/- 5.71 | 57.25 +/- 20.15 | 50.90 +/- 0.99 |
| EleutherAI-1.3b | 53.14 | 50.57 +/- 7.03 | 55.43 +/- 23.20 | 53.32 +/- 2.05 |
| EleutherAI-2.7b | 59.17 | 50.95 +/- 7.43 | 74.64 +/- 4.97 | 58.10 +/- 2.92 |
| EleutherAI-6b | 56.36 | 53.22 +/- 6.86 | 69.20 +/- 7.36 | 55.73 +/- 2.43 |
| EleutherAI-20b | 57.53 | 49.43 +/- 6.68 | 72.83 +/- 5.99 | 56.20 +/- 3.01 |
| Cohere-409m | 51.61 | 51.33 +/- 4.84 | 52.54 +/- 22.71 | 51.25 +/- 2.93 |
| Cohere-6b | 57.28 | 51.52 +/- 6.68 | 64.49 +/- 16.06 | 57.06 +/- 3.13 |
| Cohere-13b | 57.19 | 52.27 +/- 5.83 | 68.12 +/- 10.99 | 57.45 +/- 3.35 |
| Cohere-52b | 58.50 | 51.52 +/- 7.21 | 73.91 +/- 5.75 | 56.85 +/- 3.81 |
| GPT-3-350m | 51.47 | 50.76 +/- 6.96 | 52.90 +/- 24.07 | 51.29 +/- 1.63 |
| GPT-3-1.3b | 57.72 | 50.00 +/- 6.29 | 67.75 +/- 9.84 | 57.06 +/- 3.78 |
| GPT-3-6.7b | 54.83 | 52.65 +/- 8.16 | 63.41 +/- 15.14 | 54.26 +/- 2.12 |
| GPT-3-175b | 57.22 | 53.03 +/- 1.93 | 71.01 +/- 4.81 | 54.61 +/- 5.58 |
| T0-3b | 48.25 | 55.68 +/- 0.66 | 27.17 +/- 2.74 | 49.83 +/- 1.90 |
| T0-11b | 55.61 | 57.95 +/- 2.18 | 47.10 +/- 17.15 | 56.33 +/- 6.49 |
| BlenderBot-90m | 46.64 | 55.49 +/- 0.42 | 23.91 +/- 0.00 | 48.32 +/- 0.00 |
| BlenderBot-3b | 53.44 | 44.51 +/- 0.42 | 76.09 +/- 0.00 | 51.81 +/- 0.20 |
| BlenderBot-9b | 53.36 | 49.24 +/- 4.81 | 71.01 +/- 5.71 | 51.03 +/- 1.63 |
| Flan-T5-780m | 63.31 | 59.28 +/- 3.90 | 68.84 +/- 7.60 | 62.23 +/- 3.13 |
| Flan-T5-3b | 52.50 | 54.55 +/- 1.61 | 48.19 +/- 11.73 | 53.14 +/- 3.15 |
| Flan-T5-11b | 60.78 | 51.52 +/- 3.39 | 73.19 +/- 7.28 | 59.60 +/- 2.18 |
| Cohere-command-6b | 66.31 | 58.90 +/- 3.62 | 73.19 +/- 2.71 | 66.15 +/- 2.41 |
| Cohere-command-52b | 60.22 | 55.49 +/- 4.66 | 60.51 +/- 16.67 | 59.99 +/- 5.09 |
| text-ada-001-unknown | 56.50 | 52.65 +/- 3.86 | 61.59 +/- 15.96 | 56.24 +/- 5.74 |
| text-babbage-001-unknown | 64.47 | 56.25 +/- 2.52 | 72.46 +/- 9.86 | 63.87 +/- 1.55 |
| text-curie-001-unknown | 68.94 | 66.48 +/- 2.34 | 68.84 +/- 5.98 | 68.48 +/- 3.63 |
| text-davinci-001-unknown | 72.31 | 59.66 +/- 5.07 | 79.35 +/- 9.78 | 73.17 +/- 2.54 |
| text-davinci-002-unknown | 70.58 | 64.20 +/- 3.75 | 80.07 +/- 5.67 | 69.94 +/- 3.69 |
| text-davinci-003-unknown | 71.25 | 63.64 +/- 1.86 | 82.25 +/- 4.77 | 71.23 +/- 2.74 |
| ChatGPT-unknown | 72.08 | 68.75 +/- 2.99 | 69.57 +/- 11.16 | 71.66 +/- 5.79 |
| GPT-4-unknown | 81.78 | 71.59 +/- 3.47 | 89.86 +/- 2.05 | 81.35 +/- 1.66 |
| Humans | 86.23 | 83.18 | 92.17 | 84.86 |

Table 23: Accuracy per label for 1-shot evaluation.

| Model | Mean | World knowledge | Idiom | Rhetorical question |
|---|---|---|---|---|
| OPT-125m | 52.72 | 43.48 +/- 5.02 | 54.92 +/- 7.24 | 59.09 +/- 4.55 |
| OPT-350m | 52.92 | 39.86 +/- 1.62 | 48.11 +/- 2.76 | 59.09 +/- 6.94 |
| OPT-1.3b | 56.31 | 54.35 +/- 4.16 | 58.33 +/- 4.48 | 53.03 +/- 12.22 |
| OPT-2.7b | 56.83 | 64.49 +/- 15.35 | 64.39 +/- 2.51 | 66.67 +/- 4.29 |
| OPT-6.7b | 60.08 | 61.59 +/- 13.84 | 68.94 +/- 6.90 | 56.06 +/- 6.25 |
| OPT-13b | 60.56 | 68.84 +/- 7.70 | 69.70 +/- 6.11 | 54.55 +/- 15.75 |
| OPT-30b | 60.33 | 71.74 +/- 5.47 | 63.26 +/- 4.81 | 51.52 +/- 10.05 |
| OPT-66b | 63.19 | 70.29 +/- 11.06 | 62.50 +/- 2.86 | 48.48 +/- 13.55 |
| OPT-175b | 58.36 | 63.77 +/- 4.81 | 66.67 +/- 8.77 | 57.58 +/- 17.14 |
| BLOOM-560m | 54.83 | 50.00 +/- 5.47 | 64.02 +/- 4.62 | 63.64 +/- 0.00 |
| BLOOM-1b1 | 52.56 | 56.52 +/- 9.05 | 59.47 +/- 2.04 | 59.09 +/- 4.55 |
| BLOOM-1b7 | 52.81 | 54.35 +/- 6.52 | 60.98 +/- 3.57 | 63.64 +/- 5.25 |
| BLOOM-3b | 55.94 | 50.72 +/- 4.10 | 64.39 +/- 4.08 | 59.09 +/- 4.55 |
| BLOOM-7b1 | 57.00 | 50.00 +/- 3.32 | 64.77 +/- 2.86 | 62.12 +/- 6.25 |
| BLOOM-176b | 61.11 | 77.54 +/- 3.90 | 66.67 +/- 6.11 | 50.00 +/- 6.94 |
| EleutherAI-125m | 51.67 | 44.93 +/- 6.95 | 50.76 +/- 4.29 | 57.58 +/- 4.29 |
| EleutherAI-1.3b | 55.72 | 47.10 +/- 4.64 | 55.68 +/- 4.50 | 50.00 +/- 11.44 |
| EleutherAI-2.7b | 55.50 | 54.35 +/- 5.47 | 67.42 +/- 4.67 | 65.15 +/- 9.70 |
| EleutherAI-6b | 54.97 | 57.25 +/- 5.84 | 60.23 +/- 4.30 | 53.03 +/- 8.16 |
| EleutherAI-20b | 55.86 | 69.57 +/- 4.35 | 62.88 +/- 4.85 | 53.03 +/- 6.25 |
| Cohere-409m | 51.89 | 42.75 +/- 6.84 | 51.89 +/- 4.62 | 54.55 +/- 5.25 |
| Cohere-6b | 57.86 | 58.70 +/- 12.74 | 67.05 +/- 5.83 | 68.18 +/- 11.44 |
| Cohere-13b | 61.78 | 71.74 +/- 11.43 | 67.42 +/- 8.47 | 37.88 +/- 9.70 |
| Cohere-52b | 62.97 | 66.67 +/- 6.48 | 70.08 +/- 4.43 | 62.12 +/- 14.29 |
| GPT-3-350m | 55.97 | 50.72 +/- 4.10 | 61.74 +/- 5.15 | 69.70 +/- 12.49 |
| GPT-3-1.3b | 60.75 | 58.70 +/- 4.16 | 65.53 +/- 4.23 | 54.55 +/- 5.25 |
| GPT-3-6.7b | 61.17 | 60.87 +/- 11.77 | 69.32 +/- 3.65 | 56.06 +/- 8.16 |
| GPT-3-175b | 65.72 | 76.81 +/- 3.24 | 73.48 +/- 2.51 | 57.58 +/- 16.32 |
| T0-3b | 48.89 | 54.35 +/- 2.17 | 42.80 +/- 2.04 | 36.36 +/- 0.00 |
| T0-11b | 47.78 | 52.17 +/- 0.00 | 40.53 +/- 2.43 | 36.36 +/- 0.00 |
| BlenderBot-90m | 49.94 | 55.07 +/- 6.48 | 47.73 +/- 9.99 | 51.52 +/- 13.55 |
| BlenderBot-3b | 53.31 | 47.83 +/- 0.00 | 61.36 +/- 0.00 | 63.64 +/- 0.00 |
| BlenderBot-9b | 52.53 | 50.72 +/- 9.61 | 57.20 +/- 12.95 | 66.67 +/- 6.78 |
| Flan-T5-780m | 62.89 | 64.49 +/- 7.70 | 67.42 +/- 13.03 | 46.97 +/- 8.16 |
| Flan-T5-3b | 52.75 | 65.22 +/- 15.47 | 55.30 +/- 9.34 | 45.45 +/- 12.86 |
| Flan-T5-11b | 57.44 | 59.42 +/- 3.24 | 61.36 +/- 12.17 | 48.48 +/- 12.49 |
| Cohere-command-6b | 65.00 | 71.74 +/- 6.99 | 71.59 +/- 3.15 | 36.36 +/- 0.00 |
| Cohere-command-52b | 72.83 | 83.33 +/- 3.90 | 83.33 +/- 2.51 | 71.21 +/- 6.25 |
| text-ada-001-unknown | 57.36 | 60.87 +/- 7.10 | 67.80 +/- 3.81 | 66.67 +/- 6.78 |
| text-babbage-001-unknown | 63.89 | 68.84 +/- 3.90 | 76.89 +/- 2.43 | 50.00 +/- 11.44 |
| text-curie-001-unknown | 64.39 | 66.67 +/- 5.98 | 68.56 +/- 9.94 | 56.06 +/- 6.25 |
| text-davinci-001-unknown | 72.72 | 93.48 +/- 4.16 | 80.68 +/- 2.18 | 57.58 +/- 12.49 |
| text-davinci-002-unknown | 75.61 | 91.30 +/- 2.51 | 87.12 +/- 2.51 | 56.06 +/- 8.16 |
| text-davinci-003-unknown | 74.31 | 90.58 +/- 5.28 | 82.20 +/- 1.56 | 54.55 +/- 7.42 |
| ChatGPT-unknown | 75.11 | 86.23 +/- 2.99 | 85.61 +/- 3.12 | 56.06 +/- 14.29 |
| GPT-4-unknown | 82.31 | 97.10 +/- 3.24 | 88.64 +/- 3.94 | 89.39 +/- 3.39 |
| Humans | 86.23 | 93.04 | 92.73 | 92.73 |

Table 24: Accuracy per label for 1-shot evaluation.

| Model | Mean | Particularised | Generalised | Other |
|---|---|---|---|---|
| OPT-125m | 52.72 | 48.30 +/- 1.83 | 60.87 +/- 13.04 | 52.89 +/- 1.04 |
| OPT-350m | 52.92 | 47.73 +/- 2.37 | 60.87 +/- 11.71 | 54.35 +/- 3.10 |
| OPT-1.3b | 56.31 | 53.41 +/- 2.71 | 52.17 +/- 9.88 | 57.41 +/- 1.38 |
| OPT-2.7b | 56.83 | 49.81 +/- 5.31 | 69.93 +/- 6.33 | 55.17 +/- 3.94 |
| OPT-6.7b | 60.08 | 52.65 +/- 7.44 | 73.55 +/- 3.18 | 59.09 +/- 5.85 |
| OPT-13b | 60.56 | 53.03 +/- 1.82 | 71.01 +/- 7.60 | 59.56 +/- 2.75 |
| OPT-30b | 60.33 | 55.87 +/- 3.31 | 70.65 +/- 8.01 | 59.26 +/- 4.61 |
| OPT-66b | 63.19 | 60.04 +/- 4.12 | 67.39 +/- 8.70 | 63.39 +/- 4.28 |
| OPT-175b | 58.36 | 56.63 +/- 3.96 | 59.42 +/- 7.06 | 57.28 +/- 7.30 |
| BLOOM-560m | 54.83 | 43.94 +/- 3.12 | 66.30 +/- 8.21 | 54.82 +/- 1.85 |
| BLOOM-1b1 | 52.56 | 47.35 +/- 3.12 | 63.04 +/- 12.36 | 51.16 +/- 1.54 |
| BLOOM-1b7 | 52.81 | 45.64 +/- 3.50 | 67.03 +/- 9.92 | 51.29 +/- 1.44 |
| BLOOM-3b | 55.94 | 45.27 +/- 1.21 | 76.09 +/- 1.26 | 55.12 +/- 1.93 |
| BLOOM-7b1 | 57.00 | 49.62 +/- 4.08 | 77.17 +/- 1.66 | 55.56 +/- 3.57 |
| BLOOM-176b | 61.11 | 58.14 +/- 3.31 | 66.67 +/- 5.98 | 59.73 +/- 3.66 |
| EleutherAI-125m | 51.67 | 50.19 +/- 2.49 | 53.99 +/- 8.82 | 52.11 +/- 0.89 |
| EleutherAI-1.3b | 55.72 | 50.57 +/- 4.67 | 57.97 +/- 13.44 | 57.36 +/- 2.66 |
| EleutherAI-2.7b | 55.50 | 48.67 +/- 4.84 | 65.22 +/- 4.86 | 54.22 +/- 2.79 |
| EleutherAI-6b | 54.97 | 49.81 +/- 1.21 | 66.30 +/- 4.82 | 54.01 +/- 3.36 |
| EleutherAI-20b | 55.86 | 53.03 +/- 3.57 | 64.49 +/- 6.36 | 53.83 +/- 2.41 |
| Cohere-409m | 51.89 | 52.84 +/- 3.98 | 48.55 +/- 3.69 | 52.54 +/- 1.96 |
| Cohere-6b | 57.86 | 44.13 +/- 1.66 | 77.54 +/- 2.05 | 57.15 +/- 5.08 |
| Cohere-13b | 61.78 | 53.98 +/- 2.05 | 74.28 +/- 3.64 | 61.41 +/- 1.84 |
| Cohere-52b | 62.97 | 60.42 +/- 8.18 | 69.20 +/- 5.24 | 61.76 +/- 4.21 |
| GPT-3-350m | 55.97 | 50.76 +/- 1.82 | 73.91 +/- 7.94 | 54.31 +/- 1.92 |
| GPT-3-1.3b | 60.75 | 53.79 +/- 2.98 | 68.48 +/- 3.26 | 61.07 +/- 1.82 |
| GPT-3-6.7b | 61.17 | 55.49 +/- 6.83 | 72.10 +/- 2.64 | 60.29 +/- 4.09 |
| GPT-3-175b | 65.72 | 62.31 +/- 4.17 | 64.86 +/- 7.26 | 65.33 +/- 2.00 |
| T0-3b | 48.89 | 56.25 +/- 1.57 | 34.06 +/- 4.29 | 49.83 +/- 0.55 |
| T0-11b | 47.78 | 56.44 +/- 0.54 | 27.54 +/- 1.02 | 49.22 +/- 0.53 |
| BlenderBot-90m | 49.94 | 52.46 +/- 4.27 | 44.57 +/- 15.66 | 50.00 +/- 1.65 |
| BlenderBot-3b | 53.31 | 44.51 +/- 0.42 | 76.09 +/- 0.00 | 51.59 +/- 0.24 |
| BlenderBot-9b | 52.53 | 54.92 +/- 3.45 | 55.80 +/- 12.90 | 50.90 +/- 2.60 |
| Flan-T5-780m | 62.89 | 56.44 +/- 3.32 | 68.84 +/- 12.90 | 63.44 +/- 6.28 |
| Flan-T5-3b | 52.75 | 55.11 +/- 1.57 | 44.20 +/- 5.98 | 52.41 +/- 3.23 |
| Flan-T5-11b | 57.44 | 53.98 +/- 1.94 | 62.68 +/- 15.85 | 57.28 +/- 4.79 |
| Cohere-command-6b | 65.00 | 60.61 +/- 3.69 | 68.12 +/- 9.53 | 65.25 +/- 1.37 |
| Cohere-command-52b | 72.83 | 67.42 +/- 2.83 | 80.07 +/- 2.92 | 71.36 +/- 1.70 |
| text-ada-001-unknown | 57.36 | 46.97 +/- 2.76 | 74.64 +/- 2.99 | 55.90 +/- 3.11 |
| text-babbage-001-unknown | 63.89 | 58.52 +/- 1.43 | 63.41 +/- 7.26 | 63.70 +/- 1.10 |
| text-curie-001-unknown | 64.39 | 60.98 +/- 2.14 | 69.93 +/- 3.42 | 64.04 +/- 5.79 |
| text-davinci-001-unknown | 72.72 | 62.31 +/- 1.66 | 76.81 +/- 2.71 | 72.83 +/- 1.70 |
| text-davinci-002-unknown | 75.61 | 68.18 +/- 2.86 | 77.54 +/- 2.05 | 75.32 +/- 3.14 |
| text-davinci-003-unknown | 74.31 | 64.20 +/- 1.43 | 80.43 +/- 5.02 | 74.50 +/- 1.29 |
| ChatGPT-unknown | 75.11 | 70.08 +/- 4.38 | 78.99 +/- 7.50 | 74.46 +/- 1.19 |
| GPT-4-unknown | 82.31 | 74.43 +/- 2.60 | 86.96 +/- 3.32 | 81.70 +/- 1.94 |
| Humans | 86.23 | 83.18 | 92.17 | 84.86 |

Table 25: Accuracy per label for 5-shot evaluation.

| Model | Mean | World knowledge | Idiom | Rhetorical question |
|---|---|---|---|---|
| OPT-125m | 50.22 | 44.93 +/- 3.24 | 57.58 +/- 7.73 | 57.58 +/- 4.29 |
| OPT-350m | 51.47 | 53.62 +/- 4.81 | 58.71 +/- 1.56 | 45.45 +/- 0.00 |
| OPT-1.3b | 58.03 | 68.84 +/- 8.48 | 63.26 +/- 6.21 | 30.30 +/- 8.57 |
| OPT-2.7b | 57.33 | 57.97 +/- 4.81 | 66.67 +/- 4.67 | 71.21 +/- 3.39 |
| OPT-6.7b | 63.31 | 66.67 +/- 16.20 | 67.42 +/- 4.08 | 42.42 +/- 16.32 |
| OPT-13b | 67.39 | 80.43 +/- 4.86 | 68.94 +/- 4.29 | 39.39 +/- 6.78 |
| OPT-30b | 65.64 | 84.78 +/- 8.60 | 66.29 +/- 8.13 | 37.88 +/- 6.25 |
| OPT-66b | 61.50 | 75.36 +/- 8.20 | 55.30 +/- 6.90 | 36.36 +/- 7.42 |
| OPT-175b | 63.89 | 78.26 +/- 7.10 | 65.15 +/- 2.83 | 43.94 +/- 3.39 |
| BLOOM-560m | 53.75 | 44.20 +/- 2.99 | 65.91 +/- 3.94 | 54.55 +/- 5.25 |
| BLOOM-1b1 | 57.39 | 49.28 +/- 6.95 | 65.15 +/- 4.85 | 66.67 +/- 6.78 |
| BLOOM-1b7 | 54.44 | 61.59 +/- 5.84 | 56.06 +/- 1.69 | 43.94 +/- 6.25 |
| BLOOM-3b | 57.19 | 50.72 +/- 3.24 | 64.77 +/- 4.87 | 63.64 +/- 12.86 |
| BLOOM-7b1 | 54.50 | 50.00 +/- 2.17 | 62.88 +/- 1.69 | 69.70 +/- 4.29 |
| BLOOM-176b | 65.42 | 76.09 +/- 6.02 | 69.32 +/- 4.87 | 43.94 +/- 3.39 |
| EleutherAI-125m | 49.56 | 50.00 +/- 3.32 | 50.38 +/- 4.43 | 34.85 +/- 3.39 |
| EleutherAI-1.3b | 57.11 | 55.07 +/- 5.98 | 63.64 +/- 4.15 | 37.88 +/- 11.03 |
| EleutherAI-2.7b | 58.03 | 71.74 +/- 4.16 | 59.85 +/- 3.12 | 43.94 +/- 14.29 |
| EleutherAI-6b | 58.39 | 67.39 +/- 6.99 | 56.82 +/- 6.01 | 42.42 +/- 18.68 |
| EleutherAI-20b | 61.14 | 65.22 +/- 8.70 | 64.77 +/- 11.11 | 30.30 +/- 8.57 |
| Cohere-409m | 53.39 | 47.83 +/- 5.61 | 59.47 +/- 5.32 | 31.82 +/- 6.94 |
| Cohere-6b | 60.89 | 65.94 +/- 8.48 | 66.67 +/- 10.05 | 45.45 +/- 9.09 |
| Cohere-13b | 62.47 | 81.88 +/- 8.10 | 62.88 +/- 10.71 | 34.85 +/- 11.03 |
| Cohere-52b | 65.14 | 73.91 +/- 5.61 | 67.80 +/- 3.05 | 51.52 +/- 6.78 |
| GPT-3-350m | 55.72 | 46.38 +/- 3.24 | 65.53 +/- 1.56 | 51.52 +/- 4.29 |
| GPT-3-1.3b | 62.64 | 72.46 +/- 10.55 | 69.70 +/- 4.48 | 37.88 +/- 12.22 |
| GPT-3-6.7b | 62.39 | 76.81 +/- 14.57 | 62.50 +/- 5.53 | 36.36 +/- 7.42 |
| GPT-3-175b | 68.72 | 82.61 +/- 4.35 | 71.59 +/- 2.54 | 60.61 +/- 13.55 |
| T0-3b | 46.67 | 52.17 +/- 0.00 | 38.64 +/- 0.00 | 36.36 +/- 0.00 |
| T0-11b | 47.00 | 52.17 +/- 0.00 | 39.02 +/- 0.85 | 36.36 +/- 0.00 |
| BlenderBot-90m | 46.58 | 52.17 +/- 0.00 | 38.64 +/- 0.00 | 36.36 +/- 0.00 |
| BlenderBot-3b | 53.36 | 47.83 +/- 0.00 | 61.36 +/- 0.00 | 63.64 +/- 0.00 |
| BlenderBot-9b | 52.81 | 47.83 +/- 4.35 | 60.98 +/- 0.85 | 63.64 +/- 0.00 |
| Flan-T5-780m | 61.03 | 61.59 +/- 4.64 | 70.08 +/- 9.59 | 42.42 +/- 4.29 |
| Flan-T5-3b | 54.89 | 62.32 +/- 7.39 | 60.61 +/- 8.26 | 34.85 +/- 3.39 |
| Flan-T5-11b | 61.64 | 68.84 +/- 6.84 | 67.80 +/- 8.03 | 43.94 +/- 8.16 |
| Cohere-command-6b | 68.56 | 77.54 +/- 9.86 | 78.79 +/- 5.36 | 39.39 +/- 4.29 |
| Cohere-command-52b | 75.42 | 87.68 +/- 3.90 | 84.09 +/- 1.31 | 74.24 +/- 9.70 |
| text-ada-001-unknown | 57.61 | 52.17 +/- 3.55 | 64.39 +/- 2.83 | 62.12 +/- 8.16 |
| text-babbage-001-unknown | 66.14 | 71.74 +/- 2.17 | 77.65 +/- 5.15 | 57.58 +/- 12.49 |
| text-curie-001-unknown | 71.33 | 76.09 +/- 2.17 | 70.08 +/- 6.07 | 43.94 +/- 3.39 |
| text-davinci-001-unknown | 74.53 | 88.41 +/- 3.24 | 78.03 +/- 5.97 | 66.67 +/- 12.49 |
| text-davinci-002-unknown | 79.56 | 90.58 +/- 1.62 | 89.02 +/- 2.04 | 69.70 +/- 6.78 |
| text-davinci-003-unknown | 79.67 | 89.13 +/- 2.17 | 86.36 +/- 2.27 | 74.24 +/- 11.03 |
| ChatGPT-unknown | 73.89 | 86.96 +/- 6.15 | 87.88 +/- 4.85 | 75.76 +/- 12.49 |
| GPT-4-unknown | 82.03 | 95.65 +/- 2.51 | 86.74 +/- 2.04 | 87.88 +/- 6.78 |
| Humans | 86.23 | 93.04 | 92.73 | 92.73 |

Table 26: Accuracy per label for 5-shot evaluation.

| Model | Mean | Particularised | Generalised | Other |
|---|---|---|---|---|
| OPT-125m | 50.22 | 47.35 +/- 3.12 | 60.87 +/- 13.28 | 48.84 +/- 3.71 |
| OPT-350m | 51.47 | 39.58 +/- 1.79 | 67.39 +/- 3.07 | 51.38 +/- 0.95 |
| OPT-1.3b | 58.03 | 56.06 +/- 3.63 | 57.61 +/- 4.12 | 58.01 +/- 2.84 |
| OPT-2.7b | 57.33 | 47.35 +/- 3.05 | 72.46 +/- 2.40 | 56.20 +/- 3.60 |
| OPT-6.7b | 63.31 | 56.63 +/- 6.93 | 71.01 +/- 6.48 | 63.74 +/- 3.18 |
| OPT-13b | 67.39 | 60.23 +/- 2.93 | 64.86 +/- 2.92 | 69.12 +/- 2.83 |
| OPT-30b | 65.64 | 59.85 +/- 1.42 | 59.42 +/- 7.70 | 67.27 +/- 4.36 |
| OPT-66b | 61.50 | 56.44 +/- 3.51 | 58.70 +/- 9.64 | 63.65 +/- 3.93 |
| OPT-175b | 63.89 | 61.55 +/- 2.22 | 52.54 +/- 5.53 | 65.33 +/- 2.44 |
| BLOOM-560m | 53.75 | 44.89 +/- 2.05 | 73.19 +/- 6.11 | 52.50 +/- 0.78 |
| BLOOM-1b1 | 57.39 | 48.67 +/- 3.44 | 70.65 +/- 4.12 | 57.02 +/- 1.86 |
| BLOOM-1b7 | 54.44 | 48.86 +/- 4.91 | 60.14 +/- 3.24 | 54.74 +/- 0.96 |
| BLOOM-3b | 57.19 | 50.00 +/- 3.35 | 72.46 +/- 2.40 | 56.24 +/- 0.89 |
| BLOOM-7b1 | 54.50 | 46.02 +/- 2.25 | 72.10 +/- 4.24 | 53.10 +/- 1.04 |
| BLOOM-176b | 65.42 | 65.53 +/- 4.38 | 49.28 +/- 9.69 | 66.88 +/- 3.42 |
| EleutherAI-125m | 49.56 | 44.32 +/- 5.76 | 56.88 +/- 4.05 | 50.04 +/- 2.49 |
| EleutherAI-1.3b | 57.11 | 50.76 +/- 3.26 | 69.93 +/- 4.77 | 56.89 +/- 1.57 |
| EleutherAI-2.7b | 58.03 | 51.52 +/- 2.43 | 61.59 +/- 4.10 | 58.61 +/- 1.39 |
| EleutherAI-6b | 58.39 | 49.62 +/- 1.69 | 63.04 +/- 5.02 | 59.91 +/- 5.04 |
| EleutherAI-20b | 61.14 | 51.52 +/- 2.43 | 61.59 +/- 12.78 | 63.48 +/- 5.29 |
| Cohere-409m | 53.39 | 50.38 +/- 1.69 | 68.48 +/- 4.82 | 52.45 +/- 0.99 |
| Cohere-6b | 60.89 | 52.65 +/- 2.14 | 64.49 +/- 5.98 | 61.80 +/- 4.77 |
| Cohere-13b | 62.47 | 59.66 +/- 6.11 | 68.84 +/- 7.28 | 62.02 +/- 3.56 |
| Cohere-52b | 65.14 | 60.04 +/- 4.07 | 68.12 +/- 7.39 | 65.46 +/- 3.30 |
| GPT-3-350m | 55.72 | 44.70 +/- 1.26 | 74.28 +/- 6.07 | 55.47 +/- 1.59 |
| GPT-3-1.3b | 62.64 | 49.24 +/- 2.51 | 67.39 +/- 4.35 | 64.38 +/- 2.51 |
| GPT-3-6.7b | 62.39 | 51.70 +/- 2.25 | 64.86 +/- 7.68 | 64.38 +/- 2.30 |
| GPT-3-175b | 68.72 | 60.98 +/- 5.74 | 66.67 +/- 8.76 | 69.90 +/- 0.68 |
| T0-3b | 46.67 | 55.68 +/- 0.00 | 23.91 +/- 0.00 | 48.32 +/- 0.15 |
| T0-11b | 47.00 | 55.87 +/- 0.42 | 25.00 +/- 1.66 | 48.62 +/- 0.23 |
| BlenderBot-90m | 46.58 | 55.11 +/- 1.27 | 24.28 +/- 0.81 | 48.28 +/- 0.31 |
| BlenderBot-3b | 53.36 | 44.32 +/- 0.00 | 76.09 +/- 0.00 | 51.72 +/- 0.10 |
| BlenderBot-9b | 52.81 | 44.32 +/- 0.93 | 75.72 +/- 0.81 | 50.95 +/- 1.02 |
| Flan-T5-780m | 61.03 | 54.36 +/- 3.50 | 71.01 +/- 12.96 | 60.77 +/- 6.09 |
| Flan-T5-3b | 54.89 | 57.01 +/- 1.79 | 41.30 +/- 9.64 | 55.47 +/- 4.00 |
| Flan-T5-11b | 61.64 | 56.25 +/- 3.20 | 64.86 +/- 17.04 | 61.76 +/- 5.21 |
| Cohere-command-6b | 68.56 | 60.23 +/- 5.00 | 74.28 +/- 6.57 | 68.91 +/- 1.47 |
| Cohere-command-52b | 75.42 | 70.08 +/- 3.39 | 77.17 +/- 3.26 | 74.68 +/- 2.80 |
| text-ada-001-unknown | 57.61 | 48.86 +/- 2.18 | 72.83 +/- 1.66 | 57.11 +/- 3.74 |
| text-babbage-001-unknown | 66.14 | 57.01 +/- 2.82 | 71.74 +/- 2.81 | 66.06 +/- 0.85 |
| text-curie-001-unknown | 71.33 | 60.04 +/- 0.78 | 69.93 +/- 5.67 | 74.63 +/- 0.98 |
| text-davinci-001-unknown | 74.53 | 60.80 +/- 3.75 | 81.16 +/- 3.69 | 75.80 +/- 1.32 |
| text-davinci-002-unknown | 79.56 | 71.02 +/- 2.76 | 87.68 +/- 1.62 | 79.03 +/- 2.26 |
| text-davinci-003-unknown | 79.67 | 71.59 +/- 1.86 | 87.68 +/- 1.02 | 79.33 +/- 1.12 |
| ChatGPT-unknown | 73.89 | 69.51 +/- 4.80 | 73.91 +/- 11.64 | 72.44 +/- 6.16 |
| GPT-4-unknown | 82.03 | 71.21 +/- 2.91 | 87.32 +/- 3.64 | 82.30 +/- 2.31 |
| Humans | 86.23 | 83.18 | 92.17 | 84.86 |

Table 27: Accuracy per label for 10-shot evaluation.

| Model | Mean | World knowledge | Idiom | Rhetorical question |
|---|---|---|---|---|
| OPT-125m | 52.89 | 55.80 +/- 8.48 | 55.68 +/- 8.78 | 66.67 +/- 6.78 |
| OPT-350m | 56.72 | 57.97 +/- 4.10 | 60.61 +/- 3.86 | 65.15 +/- 16.11 |
| OPT-1.3b | 59.92 | 70.29 +/- 4.64 | 54.17 +/- 3.57 | 34.85 +/- 3.39 |
| OPT-2.7b | 58.03 | 52.17 +/- 2.51 | 65.53 +/- 2.76 | 63.64 +/- 5.25 |
| OPT-6.7b | 63.28 | 71.01 +/- 5.42 | 65.53 +/- 6.07 | 53.03 +/- 17.73 |
| OPT-13b | 65.75 | 77.54 +/- 6.84 | 63.26 +/- 4.23 | 50.00 +/- 10.16 |
| OPT-30b | 63.36 | 78.99 +/- 4.64 | 56.06 +/- 8.57 | 33.33 +/- 4.29 |
| OPT-66b | 60.81 | 71.01 +/- 7.80 | 54.55 +/- 9.46 | 36.36 +/- 0.00 |
| OPT-175b | 60.75 | 76.81 +/- 4.10 | 58.33 +/- 6.52 | 43.94 +/- 11.03 |
| BLOOM-560m | 54.56 | 49.28 +/- 3.24 | 64.39 +/- 3.12 | 63.64 +/- 0.00 |
| BLOOM-1b1 | 57.31 | 55.07 +/- 5.42 | 60.61 +/- 5.36 | 59.09 +/- 8.70 |
| BLOOM-1b7 | 53.14 | 68.12 +/- 6.48 | 45.45 +/- 2.62 | 59.09 +/- 11.44 |
| BLOOM-3b | 59.39 | 54.35 +/- 4.86 | 65.53 +/- 5.32 | 66.67 +/- 4.29 |
| BLOOM-7b1 | 56.11 | 53.62 +/- 7.39 | 65.53 +/- 3.81 | 69.70 +/- 6.78 |
| BLOOM-176b | 63.47 | 75.36 +/- 5.98 | 68.18 +/- 9.00 | 42.42 +/- 4.29 |
| EleutherAI-125m | 54.39 | 58.70 +/- 6.02 | 52.27 +/- 5.41 | 83.33 +/- 19.93 |
| EleutherAI-1.3b | 57.83 | 67.39 +/- 5.47 | 60.61 +/- 5.19 | 62.12 +/- 11.03 |
| EleutherAI-2.7b | 57.03 | 73.19 +/- 2.99 | 55.68 +/- 5.37 | 66.67 +/- 13.55 |
| EleutherAI-6b | 57.64 | 64.49 +/- 3.90 | 51.14 +/- 11.04 | 56.06 +/- 16.94 |
| EleutherAI-20b | 59.33 | 67.39 +/- 5.47 | 62.12 +/- 10.47 | 37.88 +/- 6.25 |
| Cohere-409m | 53.92 | 63.04 +/- 5.47 | 46.21 +/- 6.90 | 51.52 +/- 11.34 |
| Cohere-6b | 58.72 | 66.67 +/- 9.61 | 63.26 +/- 14.46 | 50.00 +/- 12.59 |
| Cohere-13b | 60.36 | 76.81 +/- 6.48 | 56.06 +/- 9.52 | 34.85 +/- 3.39 |
| Cohere-52b | 63.31 | 72.46 +/- 5.98 | 68.18 +/- 4.55 | 51.52 +/- 10.05 |
| GPT-3-350m | 57.72 | 53.62 +/- 3.24 | 63.64 +/- 6.01 | 65.15 +/- 8.16 |
| GPT-3-1.3b | 60.92 | 73.19 +/- 7.28 | 59.47 +/- 5.78 | 48.48 +/- 8.57 |
| GPT-3-6.7b | 63.94 | 71.01 +/- 7.80 | 67.80 +/- 1.56 | 40.91 +/- 8.70 |
| GPT-3-175b | 67.28 | 76.81 +/- 9.61 | 68.56 +/- 5.32 | 81.82 +/- 10.50 |
| T0-3b | 46.67 | 52.17 +/- 0.00 | 38.64 +/- 0.00 | 36.36 +/- 0.00 |
| T0-11b | 46.72 | 52.17 +/- 0.00 | 38.64 +/- 0.00 | 36.36 +/- 0.00 |
| BlenderBot-90m | 46.67 | 52.17 +/- 0.00 | 38.64 +/- 0.00 | 36.36 +/- 0.00 |
| BlenderBot-3b | 53.25 | 47.83 +/- 0.00 | 61.36 +/- 0.00 | 63.64 +/- 0.00 |
| BlenderBot-9b | 53.36 | 42.03 +/- 4.10 | 62.12 +/- 3.12 | 63.64 +/- 0.00 |
| Flan-T5-780m | 60.19 | 63.04 +/- 2.17 | 68.56 +/- 10.77 | 40.91 +/- 4.55 |
| Flan-T5-3b | 55.14 | 61.59 +/- 6.84 | 58.71 +/- 10.37 | 36.36 +/- 0.00 |
| Flan-T5-11b | 60.56 | 67.39 +/- 7.43 | 70.83 +/- 10.45 | 40.91 +/- 4.55 |
| Cohere-command-6b | 68.22 | 78.99 +/- 5.28 | 74.62 +/- 5.32 | 36.36 +/- 0.00 |
| Cohere-command-52b | 75.64 | 88.41 +/- 3.24 | 84.85 +/- 2.51 | 66.67 +/- 8.57 |
| text-ada-001-unknown | 57.36 | 64.49 +/- 7.28 | 56.44 +/- 5.78 | 57.58 +/- 6.78 |
| text-babbage-001-unknown | 63.53 | 67.39 +/- 2.17 | 73.11 +/- 4.62 | 68.18 +/- 4.55 |
| text-curie-001-unknown | 70.17 | 83.33 +/- 1.62 | 76.14 +/- 2.86 | 45.45 +/- 5.25 |
| text-davinci-001-unknown | 74.97 | 89.13 +/- 2.17 | 83.33 +/- 2.83 | 59.09 +/- 6.94 |
| text-davinci-002-unknown | 79.56 | 93.48 +/- 2.17 | 88.26 +/- 0.85 | 66.67 +/- 8.57 |
| text-davinci-003-unknown | 79.00 | 94.93 +/- 3.90 | 85.61 +/- 3.39 | 66.67 +/- 4.29 |
| ChatGPT-unknown | 74.28 | 84.06 +/- 6.48 | 86.36 +/- 4.35 | 62.12 +/- 11.03 |
| GPT-4-unknown | 81.31 | 94.93 +/- 2.99 | 86.74 +/- 4.03 | 89.39 +/- 3.39 |
| Humans | 86.23 | 93.04 | 92.73 | 92.73 |

Table 28: Accuracy per label for 10-shot evaluation.

| Model | Mean | Particularised | Generalised | Other |
|---|---|---|---|---|
| OPT-125m | 52.89 | 51.33 +/- 4.52 | 57.97 +/- 15.30 | 51.68 +/- 1.58 |
| OPT-350m | 56.72 | 55.11 +/- 1.57 | 72.10 +/- 2.32 | 54.39 +/- 1.50 |
| OPT-1.3b | 59.92 | 60.23 +/- 0.93 | 61.23 +/- 2.32 | 60.34 +/- 2.77 |
| OPT-2.7b | 58.03 | 49.62 +/- 3.32 | 74.64 +/- 2.05 | 57.19 +/- 3.77 |
| OPT-6.7b | 63.28 | 58.33 +/- 3.97 | 73.19 +/- 3.48 | 62.70 +/- 3.22 |
| OPT-13b | 65.75 | 60.23 +/- 2.45 | 72.10 +/- 3.18 | 66.37 +/- 2.84 |
| OPT-30b | 63.36 | 62.31 +/- 2.40 | 65.22 +/- 6.28 | 64.17 +/- 3.73 |
| OPT-66b | 60.81 | 57.58 +/- 1.26 | 60.51 +/- 7.98 | 62.36 +/- 3.33 |
| OPT-175b | 60.75 | 60.98 +/- 2.24 | 56.16 +/- 2.92 | 60.94 +/- 3.42 |
| BLOOM-560m | 54.56 | 46.21 +/- 1.26 | 73.19 +/- 2.99 | 53.06 +/- 0.77 |
| BLOOM-1b1 | 57.31 | 47.54 +/- 4.97 | 64.13 +/- 7.40 | 58.31 +/- 3.30 |
| BLOOM-1b7 | 53.14 | 53.03 +/- 3.45 | 53.62 +/- 4.64 | 52.80 +/- 1.74 |
| BLOOM-3b | 59.39 | 55.49 +/- 2.22 | 69.57 +/- 5.89 | 58.35 +/- 0.41 |
| BLOOM-7b1 | 56.11 | 49.62 +/- 5.07 | 71.38 +/- 2.92 | 54.35 +/- 3.34 |
| BLOOM-176b | 63.47 | 68.18 +/- 3.47 | 50.36 +/- 9.00 | 63.35 +/- 4.15 |
| EleutherAI-125m | 54.39 | 55.11 +/- 3.13 | 63.41 +/- 6.20 | 52.20 +/- 2.23 |
| EleutherAI-1.3b | 57.83 | 50.00 +/- 2.78 | 69.20 +/- 2.32 | 57.15 +/- 1.52 |
| EleutherAI-2.7b | 57.03 | 57.20 +/- 2.76 | 57.25 +/- 6.23 | 55.77 +/- 0.96 |
| EleutherAI-6b | 57.64 | 56.25 +/- 2.76 | 59.42 +/- 10.25 | 58.05 +/- 4.71 |
| EleutherAI-20b | 59.33 | 57.39 +/- 1.83 | 63.04 +/- 13.63 | 59.04 +/- 3.30 |
| Cohere-409m | 53.92 | 57.39 +/- 2.60 | 66.30 +/- 4.98 | 51.94 +/- 1.99 |
| Cohere-6b | 58.72 | 51.70 +/- 2.34 | 64.86 +/- 6.33 | 58.74 +/- 5.75 |
| Cohere-13b | 60.36 | 58.52 +/- 1.27 | 70.29 +/- 5.98 | 59.73 +/- 5.24 |
| Cohere-52b | 63.31 | 53.41 +/- 2.93 | 67.75 +/- 7.88 | 64.17 +/- 2.12 |
| GPT-3-350m | 57.72 | 50.95 +/- 2.89 | 73.91 +/- 1.26 | 56.59 +/- 1.79 |
| GPT-3-1.3b | 60.92 | 57.01 +/- 5.01 | 63.77 +/- 2.71 | 61.15 +/- 1.10 |
| GPT-3-6.7b | 63.94 | 58.52 +/- 4.19 | 66.67 +/- 5.28 | 64.56 +/- 1.31 |
| GPT-3-175b | 67.28 | 63.45 +/- 2.66 | 68.84 +/- 4.64 | 66.75 +/- 2.65 |
| T0-3b | 46.67 | 55.68 +/- 0.00 | 24.28 +/- 0.81 | 48.28 +/- 0.10 |
| T0-11b | 46.72 | 55.68 +/- 0.00 | 24.28 +/- 0.81 | 48.36 +/- 0.10 |
| BlenderBot-90m | 46.67 | 55.68 +/- 0.00 | 23.91 +/- 0.00 | 48.32 +/- 0.00 |
| BlenderBot-3b | 53.25 | 44.70 +/- 0.54 | 76.09 +/- 0.00 | 51.46 +/- 0.23 |
| BlenderBot-9b | 53.36 | 45.27 +/- 1.21 | 76.09 +/- 1.77 | 51.77 +/- 0.69 |
| Flan-T5-780m | 60.19 | 54.17 +/- 2.51 | 71.38 +/- 10.75 | 59.60 +/- 4.49 |
| Flan-T5-3b | 55.14 | 54.92 +/- 1.42 | 43.48 +/- 12.10 | 56.29 +/- 3.84 |
| Flan-T5-11b | 60.56 | 59.66 +/- 3.33 | 57.61 +/- 13.09 | 60.03 +/- 5.13 |
| Cohere-command-6b | 68.22 | 63.07 +/- 4.44 | 77.17 +/- 6.73 | 67.79 +/- 2.45 |
| Cohere-command-52b | 75.64 | 70.27 +/- 1.53 | 76.45 +/- 4.24 | 75.15 +/- 1.17 |
| text-ada-001-unknown | 57.36 | 49.24 +/- 3.32 | 61.96 +/- 5.14 | 58.23 +/- 1.53 |
| text-babbage-001-unknown | 63.53 | 56.63 +/- 2.22 | 65.22 +/- 5.47 | 63.35 +/- 1.49 |
| text-curie-001-unknown | 70.17 | 62.69 +/- 2.12 | 67.75 +/- 7.36 | 71.32 +/- 1.01 |
| text-davinci-001-unknown | 74.97 | 63.83 +/- 1.21 | 80.80 +/- 1.95 | 75.41 +/- 1.79 |
| text-davinci-002-unknown | 79.56 | 70.08 +/- 1.56 | 84.78 +/- 2.51 | 79.59 +/- 2.79 |
| text-davinci-003-unknown | 79.00 | 68.18 +/- 1.31 | 87.32 +/- 1.49 | 79.07 +/- 1.38 |
| ChatGPT-unknown | 74.28 | 68.37 +/- 4.37 | 75.36 +/- 11.06 | 73.90 +/- 4.82 |
| GPT-4-unknown | 81.31 | 70.83 +/- 4.29 | 86.96 +/- 2.81 | 81.31 +/- 3.82 |
| Humans | 86.23 | 83.18 | 92.17 | 84.86 |

Table 29: Accuracy per label for 15-shot evaluation.

| Model | Mean | World knowledge | Idiom | Rhetorical question |
|---|---|---|---|---|
| OPT-125m | 51.86 | 44.93 +/- 4.10 | 53.41 +/- 8.48 | 43.94 +/- 19.93 |
| OPT-350m | 55.42 | 48.55 +/- 2.99 | 48.48 +/- 2.51 | 42.42 +/- 6.78 |
| OPT-1.3b | 61.61 | 64.49 +/- 5.28 | 68.94 +/- 4.67 | 42.42 +/- 6.78 |
| OPT-2.7b | 59.53 | 55.80 +/- 5.84 | 62.50 +/- 2.18 | 60.61 +/- 4.29 |
| OPT-6.7b | 64.72 | 55.80 +/- 7.28 | 68.18 +/- 3.47 | 60.61 +/- 16.32 |
| OPT-13b | 65.17 | 64.49 +/- 6.36 | 66.67 +/- 6.11 | 54.55 +/- 5.25 |
| OPT-30b | 64.06 | 68.84 +/- 4.64 | 60.23 +/- 5.98 | 43.94 +/- 8.16 |
| OPT-66b | 61.83 | 65.94 +/- 11.34 | 55.30 +/- 8.26 | 39.39 +/- 4.29 |
| OPT-175b | 64.78 | 76.09 +/- 11.16 | 67.05 +/- 9.44 | 50.00 +/- 6.94 |
| BLOOM-560m | 55.00 | 47.83 +/- 2.51 | 59.09 +/- 2.27 | 62.12 +/- 3.39 |
| BLOOM-1b1 | 57.58 | 50.00 +/- 4.86 | 53.03 +/- 2.51 | 57.58 +/- 4.29 |
| BLOOM-1b7 | 55.14 | 60.14 +/- 12.40 | 50.38 +/- 4.98 | 53.03 +/- 16.94 |
| BLOOM-3b | 58.69 | 44.93 +/- 3.24 | 61.36 +/- 6.94 | 57.58 +/- 6.78 |
| BLOOM-7b1 | 55.67 | 55.07 +/- 7.80 | 61.36 +/- 2.93 | 56.06 +/- 8.16 |
| BLOOM-176b | 61.89 | 77.54 +/- 9.86 | 70.08 +/- 7.59 | 37.88 +/- 3.39 |
| EleutherAI-125m | 56.03 | 60.14 +/- 7.70 | 42.80 +/- 4.62 | 59.09 +/- 13.64 |
| EleutherAI-1.3b | 57.44 | 49.28 +/- 2.05 | 51.52 +/- 6.11 | 39.39 +/- 15.45 |
| EleutherAI-2.7b | 58.08 | 53.62 +/- 4.81 | 57.20 +/- 4.81 | 56.06 +/- 11.03 |
| EleutherAI-6b | 58.81 | 58.70 +/- 10.87 | 56.06 +/- 10.47 | 56.06 +/- 6.25 |
| EleutherAI-20b | 59.86 | 55.80 +/- 2.99 | 63.64 +/- 9.19 | 42.42 +/- 4.29 |
| Cohere-409m | 55.19 | 50.00 +/- 4.16 | 50.76 +/- 4.85 | 42.42 +/- 6.78 |
| Cohere-6b | 60.44 | 65.94 +/- 9.19 | 67.05 +/- 9.88 | 50.00 +/- 17.99 |
| Cohere-13b | 62.83 | 67.39 +/- 13.92 | 64.77 +/- 5.53 | 43.94 +/- 9.70 |
| Cohere-52b | 64.72 | 63.04 +/- 6.52 | 69.32 +/- 7.28 | 63.64 +/- 13.89 |
| GPT-3-350m | 58.83 | 50.00 +/- 7.43 | 53.41 +/- 1.74 | 42.42 +/- 13.55 |
| GPT-3-1.3b | 62.86 | 53.62 +/- 7.39 | 65.91 +/- 3.71 | 50.00 +/- 14.61 |
| GPT-3-6.7b | 65.17 | 62.32 +/- 6.95 | 63.64 +/- 2.27 | 51.52 +/- 10.05 |
| GPT-3-175b | 68.31 | 78.26 +/- 5.02 | 66.67 +/- 4.48 | 56.06 +/- 11.03 |
| T0-3b | 46.67 | 52.17 +/- 0.00 | 38.64 +/- 0.00 | 36.36 +/- 0.00 |
| T0-11b | 46.81 | 52.17 +/- 0.00 | 38.64 +/- 0.00 | 36.36 +/- 0.00 |
| BlenderBot-90m | 46.56 | 52.17 +/- 0.00 | 38.64 +/- 0.00 | 34.85 +/- 3.39 |
| BlenderBot-3b | 53.14 | 47.83 +/- 0.00 | 61.36 +/- 0.00 | 63.64 +/- 0.00 |
| BlenderBot-9b | 53.19 | 45.65 +/- 3.32 | 60.61 +/- 5.03 | 65.15 +/- 3.39 |
| Flan-T5-780m | 61.50 | 65.94 +/- 5.84 | 67.42 +/- 10.71 | 42.42 +/- 4.29 |
| Flan-T5-3b | 55.08 | 66.67 +/- 10.55 | 60.23 +/- 11.64 | 36.36 +/- 0.00 |
| Flan-T5-11b | 60.83 | 65.94 +/- 7.28 | 68.56 +/- 8.65 | 45.45 +/- 7.42 |
| Cohere-command-6b | 70.03 | 80.43 +/- 3.32 | 78.41 +/- 2.18 | 45.45 +/- 10.50 |
| Cohere-command-52b | 75.39 | 89.13 +/- 2.17 | 83.33 +/- 1.69 | 72.73 +/- 5.25 |
| text-ada-001-unknown | 58.28 | 55.07 +/- 5.98 | 56.06 +/- 5.67 | 63.64 +/- 13.89 |
| text-babbage-001-unknown | 65.19 | 63.04 +/- 4.86 | 77.27 +/- 3.71 | 68.18 +/- 6.94 |
| text-curie-001-unknown | 69.92 | 79.71 +/- 2.05 | 73.11 +/- 1.56 | 45.45 +/- 10.50 |
| text-davinci-001-unknown | 75.31 | 88.41 +/- 2.05 | 82.95 +/- 2.86 | 57.58 +/- 8.57 |
| text-davinci-002-unknown | 79.06 | 94.93 +/- 1.62 | 85.23 +/- 2.86 | 72.73 +/- 15.75 |
| text-davinci-003-unknown | 79.03 | 91.30 +/- 0.00 | 85.61 +/- 1.69 | 69.70 +/- 15.45 |
| ChatGPT-unknown | 75.56 | 86.23 +/- 4.64 | 86.74 +/- 4.03 | 60.61 +/- 10.05 |
| GPT-4-unknown | 82.08 | 95.65 +/- 2.51 | 81.44 +/- 2.43 | 90.91 +/- 0.00 |
| Humans | 86.23 | 93.04 | 92.73 | 92.73 |

Table 30: Accuracy per label for 15-shot evaluation.

| Model | Mean | Particularised | Generalised | Other |
|---|---|---|---|---|
| OPT-125m | 51.86 | 50.95 +/- 4.32 | 55.80 +/- 20.34 | 51.94 +/- 0.84 |
| OPT-350m | 55.42 | 55.30 +/- 3.39 | 60.51 +/- 4.24 | 56.29 +/- 0.75 |
| OPT-1.3b | 61.61 | 57.39 +/- 5.90 | 63.77 +/- 6.60 | 61.76 +/- 3.50 |
| OPT-2.7b | 59.53 | 49.24 +/- 2.98 | 75.00 +/- 1.66 | 59.78 +/- 5.03 |
| OPT-6.7b | 64.72 | 58.71 +/- 7.78 | 74.28 +/- 5.24 | 65.12 +/- 2.69 |
| OPT-13b | 65.17 | 64.02 +/- 2.34 | 65.22 +/- 7.32 | 65.50 +/- 1.25 |
| OPT-30b | 64.06 | 62.50 +/- 3.99 | 61.59 +/- 6.84 | 65.42 +/- 3.99 |
| OPT-66b | 61.83 | 58.71 +/- 7.21 | 57.61 +/- 5.14 | 64.25 +/- 3.20 |
| OPT-175b | 64.78 | 64.20 +/- 4.03 | 59.78 +/- 3.92 | 64.90 +/- 3.66 |
| BLOOM-560m | 55.00 | 44.13 +/- 2.58 | 74.64 +/- 4.64 | 54.78 +/- 1.75 |
| BLOOM-1b1 | 57.58 | 45.45 +/- 1.74 | 64.13 +/- 5.58 | 60.42 +/- 2.34 |
| BLOOM-1b7 | 55.14 | 52.27 +/- 5.21 | 57.61 +/- 8.40 | 55.73 +/- 0.84 |
| BLOOM-3b | 58.69 | 49.81 +/- 1.21 | 74.28 +/- 2.92 | 59.30 +/- 0.95 |
| BLOOM-7b1 | 55.67 | 48.67 +/- 3.74 | 70.29 +/- 3.90 | 54.78 +/- 3.23 |
| BLOOM-176b | 61.89 | 64.02 +/- 4.43 | 46.38 +/- 11.20 | 62.02 +/- 4.53 |
| EleutherAI-125m | 56.03 | 56.82 +/- 3.47 | 54.35 +/- 5.89 | 57.11 +/- 0.65 |
| EleutherAI-1.3b | 57.44 | 51.14 +/- 3.21 | 61.59 +/- 9.11 | 59.95 +/- 2.44 |
| EleutherAI-2.7b | 58.08 | 57.58 +/- 2.24 | 61.96 +/- 6.49 | 58.27 +/- 1.50 |
| EleutherAI-6b | 58.81 | 54.17 +/- 6.07 | 66.67 +/- 8.48 | 59.35 +/- 4.11 |
| EleutherAI-20b | 59.86 | 55.11 +/- 3.98 | 62.68 +/- 11.87 | 60.85 +/- 4.53 |
| Cohere-409m | 55.19 | 52.65 +/- 1.69 | 61.23 +/- 8.08 | 56.12 +/- 2.19 |
| Cohere-6b | 60.44 | 49.62 +/- 2.60 | 71.01 +/- 6.48 | 60.85 +/- 4.38 |
| Cohere-13b | 62.83 | 57.77 +/- 3.17 | 71.01 +/- 5.98 | 63.09 +/- 2.62 |
| Cohere-52b | 64.72 | 57.39 +/- 2.69 | 71.01 +/- 4.81 | 65.16 +/- 1.07 |
| GPT-3-350m | 58.83 | 55.68 +/- 2.54 | 65.22 +/- 8.96 | 60.29 +/- 1.99 |
| GPT-3-1.3b | 62.86 | 56.06 +/- 5.67 | 63.04 +/- 7.63 | 64.86 +/- 1.78 |
| GPT-3-6.7b | 65.17 | 58.33 +/- 4.62 | 73.55 +/- 7.88 | 66.37 +/- 2.47 |
| GPT-3-175b | 68.31 | 64.77 +/- 4.10 | 71.38 +/- 6.33 | 68.60 +/- 3.97 |
| T0-3b | 46.67 | 55.68 +/- 0.00 | 23.91 +/- 0.00 | 48.32 +/- 0.00 |
| T0-11b | 46.81 | 55.68 +/- 0.00 | 25.00 +/- 1.09 | 48.41 +/- 0.12 |
| BlenderBot-90m | 46.56 | 55.68 +/- 0.00 | 23.91 +/- 0.00 | 48.19 +/- 0.13 |
| BlenderBot-3b | 53.14 | 44.32 +/- 0.00 | 75.36 +/- 1.02 | 51.46 +/- 0.23 |
| BlenderBot-9b | 53.19 | 44.13 +/- 1.79 | 75.72 +/- 1.49 | 51.72 +/- 0.74 |
| Flan-T5-780m | 61.50 | 56.63 +/- 2.22 | 71.74 +/- 11.30 | 60.90 +/- 4.55 |
| Flan-T5-3b | 55.08 | 56.82 +/- 1.31 | 45.65 +/- 12.92 | 55.04 +/- 3.39 |
| Flan-T5-11b | 60.83 | 57.01 +/- 3.37 | 60.14 +/- 15.81 | 61.02 +/- 5.34 |
| Cohere-command-6b | 70.03 | 60.80 +/- 4.72 | 72.83 +/- 8.30 | 70.84 +/- 1.68 |
| Cohere-command-52b | 75.39 | 69.89 +/- 2.43 | 76.81 +/- 3.90 | 74.76 +/- 1.01 |
| text-ada-001-unknown | 58.28 | 52.08 +/- 3.74 | 69.20 +/- 3.42 | 58.57 +/- 2.04 |
| text-babbage-001-unknown | 65.19 | 58.33 +/- 2.43 | 67.03 +/- 3.42 | 65.12 +/- 2.40 |
| text-curie-001-unknown | 69.92 | 62.50 +/- 1.47 | 68.84 +/- 6.72 | 71.40 +/- 0.93 |
| text-davinci-001-unknown | 75.31 | 64.58 +/- 2.12 | 83.70 +/- 1.66 | 75.54 +/- 0.95 |
| text-davinci-002-unknown | 79.06 | 72.92 +/- 1.02 | 86.96 +/- 3.97 | 77.99 +/- 2.77 |
| text-davinci-003-unknown | 79.03 | 69.32 +/- 2.37 | 87.68 +/- 1.62 | 78.94 +/- 1.12 |
| ChatGPT-unknown | 75.56 | 72.16 +/- 4.81 | 77.54 +/- 7.90 | 74.63 +/- 4.41 |
| GPT-4-unknown | 82.08 | 72.92 +/- 1.02 | 86.23 +/- 2.99 | 82.69 +/- 3.87 |
| Humans | 86.23 | 83.18 | 92.17 | 84.86 |

Table 31: Accuracy per label for 30-shot evaluation.

| Model | Mean | World knowledge | Idiom | Rhetorical question |
|---|---|---|---|---|
| OPT-125m | 51.50 | 55.80 +/- 5.28 | 54.55 +/- 9.28 | 54.55 +/- 9.09 |
| OPT-350m | 54.61 | 49.28 +/- 7.39 | 56.82 +/- 1.86 | 37.88 +/- 9.70 |
| OPT-1.3b | 61.67 | 71.01 +/- 3.24 | 67.80 +/- 5.48 | 28.79 +/- 11.03 |
| OPT-2.7b | 59.86 | 58.70 +/- 10.87 | 71.21 +/- 8.37 | 46.97 +/- 11.03 |
| OPT-6.7b | 63.61 | 62.32 +/- 11.13 | 67.05 +/- 2.86 | 46.97 +/- 19.93 |
| OPT-13b | 63.39 | 60.14 +/- 5.28 | 60.98 +/- 5.93 | 46.97 +/- 6.25 |
| OPT-30b | 65.47 | 71.74 +/- 8.23 | 62.88 +/- 7.38 | 37.88 +/- 3.39 |
| OPT-66b | 60.83 | 60.14 +/- 3.90 | 51.52 +/- 11.79 | 43.94 +/- 6.25 |
| OPT-175b | 62.44 | 65.94 +/- 10.77 | 62.50 +/- 13.43 | 60.61 +/- 4.29 |
| BLOOM-560m | 55.00 | 47.10 +/- 1.62 | 60.98 +/- 2.43 | 62.12 +/- 3.39 |
| BLOOM-1b1 | 56.89 | 49.28 +/- 3.24 | 54.92 +/- 8.65 | 46.97 +/- 8.16 |
| BLOOM-1b7 | 52.28 | 52.90 +/- 7.28 | 47.35 +/- 9.32 | 36.36 +/- 16.60 |
| BLOOM-3b | 58.64 | 50.72 +/- 2.05 | 62.50 +/- 5.68 | 59.09 +/- 6.94 |
| BLOOM-7b1 | 57.61 | 50.72 +/- 5.98 | 61.74 +/- 3.81 | 54.55 +/- 9.09 |
| BLOOM-176b | 61.06 | 73.19 +/- 8.85 | 66.29 +/- 9.41 | 48.48 +/- 4.29 |
| EleutherAI-125m | 53.44 | 47.10 +/- 4.64 | 47.73 +/- 6.43 | 39.39 +/- 15.45 |
| EleutherAI-1.3b | 55.97 | 44.93 +/- 4.81 | 51.89 +/- 6.74 | 37.88 +/- 6.25 |
| EleutherAI-2.7b | 57.36 | 62.32 +/- 5.98 | 53.41 +/- 2.86 | 37.88 +/- 11.03 |
| EleutherAI-6b | 58.75 | 59.42 +/- 12.21 | 52.27 +/- 14.43 | 36.36 +/- 5.25 |
| EleutherAI-20b | 57.36 | 57.97 +/- 5.42 | 60.61 +/- 10.30 | 31.82 +/- 8.70 |
| Cohere-409m | 57.17 | 47.83 +/- 3.55 | 53.41 +/- 3.15 | 53.03 +/- 3.39 |
| Cohere-6b | 60.36 | 58.70 +/- 13.92 | 62.50 +/- 10.23 | 54.55 +/- 10.50 |
| Cohere-13b | 64.81 | 70.29 +/- 21.65 | 65.91 +/- 7.07 | 45.45 +/- 5.25 |
| Cohere-52b | 65.72 | 67.39 +/- 8.60 | 66.29 +/- 1.56 | 53.03 +/- 11.03 |
| GPT-3-350m | 60.25 | 55.07 +/- 2.05 | 57.95 +/- 5.83 | 51.52 +/- 10.05 |
| GPT-3-1.3b | 60.19 | 61.59 +/- 3.90 | 54.92 +/- 6.99 | 43.94 +/- 9.70 |
| GPT-3-6.7b | 62.86 | 56.52 +/- 4.35 | 65.53 +/- 3.32 | 50.00 +/- 6.94 |
| GPT-3-175b | 68.31 | 67.39 +/- 5.47 | 72.73 +/- 2.62 | 75.76 +/- 4.29 |
| T0-3b | 46.67 | 52.17 +/- 0.00 | 38.64 +/- 0.00 | 36.36 +/- 0.00 |
| T0-11b | 46.75 | 52.17 +/- 0.00 | 38.64 +/- 0.00 | 36.36 +/- 0.00 |
| BlenderBot-90m | 46.67 | 51.45 +/- 1.62 | 38.64 +/- 0.00 | 36.36 +/- 0.00 |
| BlenderBot-3b | 53.25 | 47.83 +/- 0.00 | 61.36 +/- 0.00 | 63.64 +/- 0.00 |
| BlenderBot-9b | 53.72 | 46.38 +/- 4.10 | 63.26 +/- 3.32 | 63.64 +/- 0.00 |
| Flan-T5-780m | 61.50 | 67.39 +/- 8.60 | 70.83 +/- 6.35 | 42.42 +/- 4.29 |
| Flan-T5-3b | 56.11 | 65.22 +/- 7.10 | 62.50 +/- 13.04 | 36.36 +/- 0.00 |
| Flan-T5-11b | 62.11 | 67.39 +/- 10.27 | 72.73 +/- 10.66 | 51.52 +/- 15.45 |
| Cohere-command-6b | 70.44 | 81.16 +/- 3.24 | 78.03 +/- 2.83 | 46.97 +/- 8.16 |
| Cohere-command-52b | 75.00 | 85.51 +/- 2.05 | 78.41 +/- 1.14 | 78.79 +/- 6.78 |
| text-ada-001-unknown | 55.58 | 50.72 +/- 7.39 | 57.58 +/- 4.29 | 57.58 +/- 8.57 |
| text-babbage-001-unknown | 66.00 | 67.39 +/- 5.47 | 71.59 +/- 3.15 | 63.64 +/- 5.25 |
| text-curie-001-unknown | 70.33 | 75.36 +/- 3.24 | 76.52 +/- 5.03 | 60.61 +/- 8.57 |
| text-davinci-001-unknown | 75.83 | 85.51 +/- 2.05 | 84.09 +/- 1.86 | 65.15 +/- 8.16 |
| text-davinci-002-unknown | 80.64 | 97.83 +/- 2.17 | 87.50 +/- 2.18 | 83.33 +/- 3.39 |
| text-davinci-003-unknown | 79.53 | 94.93 +/- 1.62 | 84.85 +/- 3.39 | 81.82 +/- 9.09 |
| ChatGPT-unknown | 75.64 | 87.68 +/- 4.64 | 89.02 +/- 4.43 | 83.33 +/- 9.70 |
| GPT-4-unknown | 82.17 | 95.65 +/- 3.55 | 87.12 +/- 3.12 | 90.91 +/- 0.00 |
| Humans | 86.23 | 93.04 | 92.73 | 92.73 |

Table 32: Accuracy per label for 30-shot evaluation.

| Model | Mean | Particularised | Generalised | Other |
|---|---|---|---|---|
| OPT-125m | 51.50 | 50.38 +/- 4.29 | 52.17 +/- 23.95 | 50.99 +/- 2.42 |
| OPT-350m | 54.61 | 55.30 +/- 1.26 | 49.28 +/- 3.48 | 55.51 +/- 1.92 |
| OPT-1.3b | 61.67 | 56.25 +/- 2.76 | 56.16 +/- 4.05 | 63.14 +/- 5.41 |
| OPT-2.7b | 59.86 | 50.19 +/- 2.89 | 69.57 +/- 4.16 | 59.95 +/- 5.23 |
| OPT-6.7b | 63.61 | 58.90 +/- 6.48 | 72.10 +/- 8.08 | 63.74 +/- 4.30 |
| OPT-13b | 63.39 | 59.47 +/- 2.14 | 64.86 +/- 6.07 | 65.07 +/- 2.24 |
| OPT-30b | 65.47 | 63.64 +/- 3.01 | 63.04 +/- 7.94 | 66.93 +/- 5.52 |
| OPT-66b | 60.83 | 63.07 +/- 3.64 | 55.43 +/- 9.29 | 62.62 +/- 4.98 |
| OPT-175b | 62.44 | 64.39 +/- 2.34 | 51.09 +/- 5.72 | 63.18 +/- 2.85 |
| BLOOM-560m | 55.00 | 47.92 +/- 2.31 | 71.01 +/- 6.23 | 54.18 +/- 1.82 |
| BLOOM-1b1 | 56.89 | 49.81 +/- 4.93 | 56.88 +/- 8.46 | 59.35 +/- 3.16 |
| BLOOM-1b7 | 52.28 | 55.49 +/- 1.79 | 40.94 +/- 8.82 | 53.83 +/- 1.25 |
| BLOOM-3b | 58.64 | 52.46 +/- 1.53 | 68.84 +/- 3.69 | 58.74 +/- 1.44 |
| BLOOM-7b1 | 57.61 | 54.36 +/- 8.98 | 70.29 +/- 2.05 | 56.76 +/- 4.88 |
| BLOOM-176b | 61.06 | 67.42 +/- 1.93 | 50.00 +/- 9.05 | 60.08 +/- 4.01 |
| EleutherAI-125m | 53.44 | 59.47 +/- 2.43 | 46.74 +/- 4.65 | 54.18 +/- 1.21 |
| EleutherAI-1.3b | 55.97 | 51.70 +/- 5.20 | 56.52 +/- 10.80 | 58.40 +/- 1.35 |
| EleutherAI-2.7b | 57.36 | 57.39 +/- 2.25 | 51.81 +/- 3.18 | 58.79 +/- 0.89 |
| EleutherAI-6b | 58.75 | 57.77 +/- 4.22 | 58.33 +/- 9.92 | 60.38 +/- 4.52 |
| EleutherAI-20b | 57.36 | 51.52 +/- 4.08 | 59.42 +/- 14.51 | 58.79 +/- 2.39 |
| Cohere-409m | 57.17 | 51.14 +/- 3.99 | 64.49 +/- 8.57 | 58.66 +/- 2.15 |
| Cohere-6b | 60.36 | 51.52 +/- 2.51 | 70.29 +/- 6.11 | 61.20 +/- 5.15 |
| Cohere-13b | 64.81 | 58.14 +/- 2.96 | 68.12 +/- 8.20 | 65.98 +/- 3.57 |
| Cohere-52b | 65.72 | 64.96 +/- 4.17 | 69.93 +/- 2.32 | 65.50 +/- 2.01 |
| GPT-3-350m | 60.25 | 57.58 +/- 2.51 | 65.22 +/- 6.28 | 60.98 +/- 1.63 |
| GPT-3-1.3b | 60.19 | 57.77 +/- 5.66 | 57.61 +/- 9.12 | 62.06 +/- 4.02 |
| GPT-3-6.7b | 62.86 | 61.17 +/- 6.41 | 63.41 +/- 7.98 | 63.65 +/- 2.52 |
| GPT-3-175b | 68.31 | 64.58 +/- 5.06 | 70.29 +/- 6.23 | 68.17 +/- 2.01 |
| T0-3b | 46.67 | 55.68 +/- 0.00 | 23.91 +/- 0.00 | 48.32 +/- 0.00 |
| T0-11b | 46.75 | 55.87 +/- 0.42 | 23.91 +/- 0.00 | 48.41 +/- 0.12 |
| BlenderBot-90m | 46.67 | 55.68 +/- 0.00 | 23.55 +/- 0.81 | 48.41 +/- 0.24 |
| BlenderBot-3b | 53.25 | 44.32 +/- 0.00 | 76.09 +/- 0.00 | 51.55 +/- 0.20 |
| BlenderBot-9b | 53.72 | 45.08 +/- 1.82 | 75.36 +/- 1.02 | 52.11 +/- 0.93 |
| Flan-T5-780m | 61.50 | 57.01 +/- 3.85 | 73.19 +/- 8.67 | 60.16 +/- 3.75 |
| Flan-T5-3b | 56.11 | 56.44 +/- 1.56 | 47.83 +/- 11.71 | 56.29 +/- 4.18 |
| Flan-T5-11b | 62.11 | 61.36 +/- 4.50 | 57.25 +/- 16.68 | 61.58 +/- 6.25 |
| Cohere-command-6b | 70.44 | 64.96 +/- 1.21 | 78.62 +/- 5.53 | 69.81 +/- 1.60 |
| Cohere-command-52b | 75.00 | 71.78 +/- 1.66 | 75.00 +/- 3.49 | 74.55 +/- 0.49 |
| text-ada-001-unknown | 55.58 | 53.41 +/- 1.97 | 56.52 +/- 4.35 | 55.86 +/- 3.03 |
| text-babbage-001-unknown | 66.00 | 60.80 +/- 2.69 | 62.32 +/- 6.11 | 66.88 +/- 2.36 |
| text-curie-001-unknown | 70.33 | 60.42 +/- 5.01 | 74.28 +/- 6.20 | 71.32 +/- 1.42 |
| text-davinci-001-unknown | 75.83 | 67.05 +/- 1.97 | 83.33 +/- 3.48 | 75.67 +/- 1.02 |
| text-davinci-002-unknown | 80.64 | 74.43 +/- 1.83 | 83.70 +/- 2.74 | 79.76 +/- 1.44 |
| text-davinci-003-unknown | 79.53 | 72.92 +/- 2.49 | 86.59 +/- 1.95 | 78.55 +/- 1.20 |
| ChatGPT-unknown | 75.64 | 67.99 +/- 2.74 | 78.26 +/- 6.15 | 74.55 +/- 3.90 |
| GPT-4-unknown | 82.17 | 71.97 +/- 2.83 | 86.23 +/- 3.48 | 82.34 +/- 2.67 |
| Humans | 86.23 | 83.18 | 92.17 | 84.86 |

Table 33: Accuracy per label for model group Example IT for 5-shot chain-of-thought evaluation.

| Model | Mean | World knowledge | Idiom | Rhetorical question |
|---|---|---|---|---|
| Cohere-command-6b | 69.14 | 72.46 +/- 5.98 | 78.03 +/- 3.12 | 62.12 +/- 8.16 |
| Cohere-command-52b | 75.28 | 78.99 +/- 1.62 | 84.47 +/- 3.57 | 51.52 +/- 8.57 |
| text-ada-001-unknown | 15.33 | 11.59 +/- 8.20 | 17.42 +/- 9.43 | 10.61 +/- 9.70 |
| text-babbage-001-unknown | 47.67 | 47.83 +/- 11.50 | 55.30 +/- 16.21 | 42.42 +/- 8.57 |
| text-curie-001-unknown | 68.22 | 69.57 +/- 6.64 | 79.17 +/- 0.85 | 69.70 +/- 10.05 |
| text-davinci-001-unknown | 67.25 | 69.57 +/- 7.10 | 71.59 +/- 3.65 | 60.61 +/- 11.34 |
| text-davinci-002-unknown | 80.06 | 92.03 +/- 1.62 | 88.26 +/- 3.57 | 46.97 +/- 24.29 |
| text-davinci-003-unknown | 83.61 | 93.48 +/- 2.17 | 93.18 +/- 0.00 | 69.70 +/- 10.05 |
| ChatGPT-unknown | 77.19 | 89.86 +/- 4.10 | 87.88 +/- 3.63 | 65.15 +/- 9.70 |
| GPT-4-unknown | 86.47 | 93.48 +/- 3.32 | 93.18 +/- 2.93 | 87.88 +/- 4.29 |
| Humans | 86.23 | 93.04 | 92.73 | 92.73 |

Table 34: Accuracy per label for model group Example IT for 5-shot chain-of-thought evaluation.

| Model | Mean | Particularised | Generalised | Other |
|---|---|---|---|---|
| Cohere-command-6b | 69.14 | 58.33 +/- 1.93 | 81.52 +/- 2.08 | 69.04 +/- 2.04 |
| Cohere-command-52b | 75.28 | 68.94 +/- 3.19 | 77.17 +/- 2.08 | 75.88 +/- 0.53 |
| text-ada-001-unknown | 15.33 | 15.53 +/- 7.73 | 14.86 +/- 9.26 | 15.50 +/- 8.33 |
| text-babbage-001-unknown | 47.67 | 45.27 +/- 11.94 | 40.94 +/- 19.34 | 48.19 +/- 14.11 |
| text-curie-001-unknown | 68.22 | 59.47 +/- 5.15 | 74.28 +/- 7.88 | 68.04 +/- 1.75 |
| text-davinci-001-unknown | 67.25 | 64.58 +/- 3.85 | 64.13 +/- 5.14 | 67.92 +/- 3.30 |
| text-davinci-002-unknown | 80.06 | 75.95 +/- 3.68 | 80.07 +/- 6.69 | 80.23 +/- 1.07 |
| text-davinci-003-unknown | 83.61 | 77.46 +/- 1.02 | 87.32 +/- 3.18 | 83.25 +/- 0.96 |
| ChatGPT-unknown | 77.19 | 72.35 +/- 1.56 | 80.43 +/- 5.47 | 76.23 +/- 1.11 |
| GPT-4-unknown | 86.47 | 81.63 +/- 2.58 | 88.77 +/- 4.05 | 86.05 +/- 1.17 |
| Humans | 86.23 | 83.18 | 92.17 | 84.86 |

Table 35: Accuracy per prompt template for BERT-cased.

| Template | k = 0 | k = 1 | k = 5 | k = 10 | k = 15 | k = 30 |
|---|---|---|---|---|---|---|
| 1 | 47.3 | 48.8 | 50.5 | 49.8 | 46.7 | 46.7 |
| 2 | 46.8 | 50.3 | 45.5 | 50.2 | 46.7 | 46.5 |
| 3 | 57.3 | 51.5 | 50.0 | 50.0 | 47.0 | 46.7 |
| 4 | 48.8 | 51.0 | 49.5 | 48.5 | 46.8 | 46.7 |
| 5 | 46.7 | 50.3 | 44.5 | 47.7 | 46.7 | 46.7 |
| 6 | 46.7 | 50.3 | 45.8 | 47.8 | 46.8 | 46.7 |
| Mean | 48.9 | 50.4 | 47.6 | 49.0 | 46.8 | 46.7 |
| − std | 3.81 | 0.832 | 2.42 | 1.04 | 0.107 | 0.0745 |
| Structured | 51.1 | 50.4 | 50.0 | 49.4 | 46.8 | 46.7 |
| − std | 4.4 | 1.17 | 0.408 | 0.665 | 0.125 | 7.11e-15 |
| Natural | 46.7 | 50.3 | 45.3 | 48.6 | 46.7 | 46.6 |
| − std | 0.0471 | 7.11e-15 | 0.556 | 1.16 | 0.0471 | 0.0943 |

Table 36: Accuracy per prompt template for BERT-uncased.

| Template | k = 0 | k = 1 | k = 5 | k = 10 | k = 15 | k = 30 |
|---|---|---|---|---|---|---|
| 1 | 57.0 | 53.2 | 51.8 | 55.2 | 51.7 | 49.3 |
| 2 | 53.7 | 50.3 | 54.0 | 48.7 | 49.0 | 49.3 |
| 3 | 54.7 | 54.7 | 57.3 | 55.5 | 53.3 | 52.8 |
| 4 | 56.7 | 51.5 | 52.3 | 54.0 | 50.3 | 49.5 |
| 5 | 53.2 | 50.2 | 50.2 | 48.3 | 48.2 | 47.2 |
| 6 | 53.3 | 50.3 | 54.2 | 49.2 | 53.0 | 53.5 |
| Mean | 54.8 | 51.7 | 53.3 | 51.8 | 50.9 | 50.3 |
| – std | 1.55 | 1.71 | 2.24 | 3.13 | 1.92 | 2.19 |
| Structured | 56.1 | 53.1 | 53.8 | 54.9 | 51.8 | 50.5 |
| – std | 1.02 | 1.31 | 2.48 | 0.648 | 1.23 | 1.6 |
| Natural | 53.4 | 50.3 | 52.8 | 48.7 | 50.1 | 50.0 |
| – std | 0.216 | 0.0471 | 1.84 | 0.368 | 2.1 | 2.62 |

Table 37: Accuracy per prompt template for RoBERTa-base.

| Template | k = 0 | k = 1 | k = 5 | k = 10 | k = 15 | k = 30 |
|---|---|---|---|---|---|---|
| 1 | 54.0 | 55.8 | 58.0 | 58.7 | 58.3 | 57.8 |
| 2 | 56.5 | 50.5 | 52.0 | 55.8 | 56.0 | 54.2 |
| 3 | 53.0 | 56.8 | 56.8 | 61.3 | 59.5 | 58.8 |
| 4 | 55.2 | 56.0 | 58.7 | 59.8 | 56.8 | 57.2 |
| 5 | 55.7 | 50.3 | 52.3 | 54.8 | 55.5 | 53.0 |
| 6 | 59.2 | 50.3 | 54.2 | 55.8 | 55.7 | 55.3 |
| Mean | 55.6 | 53.3 | 55.3 | 57.7 | 57.0 | 56.1 |
| – std | 1.97 | 2.93 | 2.65 | 2.38 | 1.47 | 2.05 |
| Structured | 54.1 | 56.2 | 57.8 | 59.9 | 58.2 | 57.9 |
| – std | 0.899 | 0.432 | 0.785 | 1.07 | 1.1 | 0.66 |
| Natural | 57.1 | 50.4 | 52.8 | 55.5 | 55.7 | 54.2 |
| – std | 1.5 | 0.0943 | 0.974 | 0.471 | 0.205 | 0.939 |

Table 38: Accuracy per prompt template for RoBERTa-large.

| Template | k = 0 | k = 1 | k = 5 | k = 10 | k = 15 | k = 30 |
|---|---|---|---|---|---|---|
| 1 | 57.7 | 50.2 | 62.0 | 64.7 | 64.7 | 60.5 |
| 2 | 46.7 | 53.3 | 58.5 | 64.2 | 61.2 | 55.7 |
| 3 | 60.8 | 54.8 | 64.5 | 62.8 | 61.8 | 59.5 |
| 4 | 66.2 | 50.3 | 64.0 | 59.0 | 57.0 | 58.2 |
| 5 | 46.7 | 53.3 | 58.8 | 63.5 | 60.5 | 56.5 |
| 6 | 46.7 | 55.5 | 59.3 | 60.0 | 60.8 | 52.3 |
| Mean | 54.1 | 52.9 | 61.2 | 62.4 | 61.0 | 57.1 |
| – std | 7.84 | 2.03 | 2.45 | 2.13 | 2.26 | 2.7 |
| Structured | 61.6 | 51.8 | 63.5 | 62.2 | 61.2 | 59.4 |
| – std | 3.51 | 2.15 | 1.08 | 2.37 | 3.18 | 0.942 |
| Natural | 46.7 | 54.0 | 58.9 | 62.6 | 60.8 | 54.8 |
| – std | 7.11e-15 | 1.04 | 0.33 | 1.84 | 0.287 | 1.82 |

Table 39: Accuracy per prompt template for GPT-2-medium.

| Template | k = 0 | k = 1 | k = 5 | k = 10 | k = 15 | k = 30 |
|---|---|---|---|---|---|---|
| 1 | 53.2 | 53.7 | 54.0 | 53.8 | 53.8 | 55.0 |
| 2 | 52.8 | 53.7 | 55.8 | 57.2 | 60.3 | 57.2 |
| 3 | 53.7 | 54.0 | 52.5 | 56.5 | 55.8 | 55.3 |
| 4 | 53.5 | 55.7 | 53.3 | 55.8 | 55.5 | 54.3 |
| 5 | 59.2 | 54.3 | 56.7 | 57.7 | 60.7 | 58.8 |
| 6 | 58.3 | 54.8 | 55.7 | 57.7 | 61.7 | 57.8 |
| Mean | 55.1 | 54.4 | 54.7 | 56.4 | 58.0 | 56.4 |
| – std | 2.6 | 0.706 | 1.5 | 1.36 | 3.03 | 1.63 |
| Structured | 53.5 | 54.5 | 53.3 | 55.4 | 55.0 | 54.9 |
| – std | 0.205 | 0.881 | 0.613 | 1.14 | 0.881 | 0.419 |
| Natural | 56.8 | 54.3 | 56.1 | 57.5 | 60.9 | 57.9 |
| – std | 2.83 | 0.45 | 0.45 | 0.236 | 0.589 | 0.66 |

Table 40: Accuracy per prompt template for GPT-2-large.

| Template | k = 0 | k = 1 | k = 5 | k = 10 | k = 15 | k = 30 |
|---|---|---|---|---|---|---|
| 1 | 53.3 | 53.3 | 54.5 | 53.5 | 55.3 | 56.2 |
| 2 | 47.5 | 56.7 | 57.5 | 57.8 | 60.8 | 61.0 |
| 3 | 55.0 | 53.8 | 55.7 | 54.0 | 54.8 | 56.0 |
| 4 | 54.0 | 53.7 | 56.2 | 53.5 | 54.8 | 56.7 |
| 5 | 47.2 | 54.5 | 56.7 | 58.8 | 61.2 | 60.8 |
| 6 | 47.0 | 53.3 | 57.2 | 59.5 | 60.3 | 60.8 |
| Mean | 50.7 | 54.2 | 56.3 | 56.2 | 57.9 | 58.6 |
| – std | 3.47 | 1.18 | 1.0 | 2.57 | 2.92 | 2.29 |
| Structured | 54.1 | 53.6 | 55.5 | 53.7 | 55.0 | 56.3 |
| – std | 0.698 | 0.216 | 0.713 | 0.236 | 0.236 | 0.294 |
| Natural | 47.2 | 54.8 | 57.1 | 58.7 | 60.8 | 60.9 |
| – std | 0.205 | 1.41 | 0.33 | 0.698 | 0.368 | 0.0943 |

Table 41: Accuracy per prompt template for GPT-2-xl.

| Template | k = 0 | k = 1 | k = 5 | k = 10 | k = 15 | k = 30 |
|---|---|---|---|---|---|---|
| 1 | 53.2 | 53.3 | 57.0 | 54.5 | 54.7 | 56.2 |
| 2 | 48.7 | 61.3 | 57.3 | 63.7 | 62.0 | 60.5 |
| 3 | 55.0 | 55.2 | 59.5 | 59.0 | 58.0 | 60.7 |
| 4 | 54.2 | 54.3 | 56.0 | 54.5 | 54.3 | 56.3 |
| 5 | 48.0 | 59.7 | 58.3 | 60.8 | 62.7 | 61.7 |
| 6 | 48.5 | 60.8 | 58.0 | 61.8 | 61.5 | 61.5 |
| Mean | 51.3 | 57.4 | 57.7 | 59.1 | 58.9 | 59.5 |
| – std | 2.92 | 3.25 | 1.1 | 3.5 | 3.43 | 2.32 |
| Structured | 54.1 | 54.3 | 57.5 | 56.0 | 55.7 | 57.7 |
| – std | 0.736 | 0.776 | 1.47 | 2.12 | 1.66 | 2.1 |
| Natural | 48.4 | 60.6 | 57.9 | 62.1 | 62.1 | 61.2 |
| – std | 0.294 | 0.668 | 0.419 | 1.2 | 0.492 | 0.525 |

Table 42: Accuracy per prompt template for EleutherAI-125M.

| Template | k = 0 | k = 1 | k = 5 | k = 10 | k = 15 | k = 30 |
|---|---|---|---|---|---|---|
| 1 | 53.3 | 53.7 | 52.7 | 56.2 | 56.2 | 54.0 |
| 2 | 52.2 | 50.0 | 47.5 | 53.5 | 55.7 | 53.3 |
| 3 | 53.3 | 53.8 | 51.2 | 55.8 | 54.8 | 52.8 |
| 4 | 53.7 | 52.5 | 51.2 | 53.8 | 55.8 | 53.2 |
| 5 | 50.7 | 50.2 | 47.3 | 53.8 | 56.2 | 53.8 |
| 6 | 48.2 | 49.8 | 47.5 | 53.2 | 57.5 | 53.5 |
| Mean | 51.9 | 51.7 | 49.6 | 54.4 | 56.0 | 53.4 |
| – std | 1.93 | 1.72 | 2.19 | 1.17 | 0.806 | 0.394 |
| Structured | 53.4 | 53.3 | 51.7 | 55.3 | 55.6 | 53.3 |
| – std | 0.189 | 0.591 | 0.707 | 1.05 | 0.589 | 0.499 |
| Natural | 50.4 | 50.0 | 47.4 | 53.5 | 56.5 | 53.5 |
| – std | 1.65 | 0.163 | 0.0943 | 0.245 | 0.759 | 0.205 |

Table 43: Accuracy per prompt template for EleutherAI-1.3B.

| Template | k = 0 | k = 1 | k = 5 | k = 10 | k = 15 | k = 30 |
|---|---|---|---|---|---|---|
| 1 | 54.3 | 53.7 | 54.8 | 57.5 | 57.2 | 56.2 |
| 2 | 51.8 | 56.8 | 57.5 | 59.0 | 55.8 | 54.7 |
| 3 | 58.0 | 55.5 | 59.5 | 58.0 | 61.5 | 57.5 |
| 4 | 53.2 | 57.5 | 56.8 | 55.2 | 56.5 | 54.7 |
| 5 | 49.7 | 55.2 | 57.5 | 58.7 | 57.2 | 56.7 |
| 6 | 51.8 | 55.7 | 56.5 | 58.7 | 56.5 | 56.2 |
| Mean | 53.1 | 55.7 | 57.1 | 57.8 | 57.4 | 56.0 |
| – std | 2.59 | 1.21 | 1.4 | 1.29 | 1.87 | 1.02 |
| Structured | 55.2 | 55.6 | 57.0 | 56.9 | 58.4 | 56.1 |
| – std | 2.05 | 1.55 | 1.93 | 1.22 | 2.21 | 1.14 |
| Natural | 51.1 | 55.9 | 57.2 | 58.8 | 56.5 | 55.9 |
| – std | 0.99 | 0.668 | 0.471 | 0.141 | 0.572 | 0.85 |

Table 44: Accuracy per prompt template for EleutherAI-2.7B.

| Template | k = 0 | k = 1 | k = 5 | k = 10 | k = 15 | k = 30 |
|---|---|---|---|---|---|---|
| 1 | 54.0 | 52.8 | 58.2 | 57.8 | 59.5 | 56.7 |
| 2 | 62.0 | 56.2 | 57.7 | 55.8 | 57.8 | 57.7 |
| 3 | 58.7 | 60.0 | 58.8 | 59.2 | 57.8 | 57.8 |
| 4 | 56.5 | 54.2 | 57.5 | 56.2 | 57.5 | 55.5 |
| 5 | 62.7 | 54.7 | 58.7 | 55.7 | 57.3 | 57.8 |
| 6 | 61.2 | 55.2 | 57.3 | 57.5 | 58.5 | 58.7 |
| Mean | 59.2 | 55.5 | 58.0 | 57.0 | 58.1 | 57.4 |
| – std | 3.13 | 2.25 | 0.576 | 1.26 | 0.741 | 1.02 |
| Structured | 56.4 | 55.7 | 58.2 | 57.7 | 58.3 | 56.7 |
| – std | 1.92 | 3.12 | 0.531 | 1.23 | 0.881 | 0.939 |
| Natural | 62.0 | 55.4 | 57.9 | 56.3 | 57.9 | 58.1 |
| – std | 0.613 | 0.624 | 0.589 | 0.826 | 0.492 | 0.45 |

Table 45: Accuracy per prompt template for EleutherAI-6B.

| Template | k = 0 | k = 1 | k = 5 | k = 10 | k = 15 | k = 30 |
|---|---|---|---|---|---|---|
| 1 | 57.5 | 58.8 | 52.7 | 53.0 | 52.5 | 51.3 |
| 2 | 57.7 | 51.8 | 63.2 | 62.7 | 64.3 | 65.3 |
| 3 | 56.2 | 58.2 | 57.2 | 53.0 | 54.7 | 54.5 |
| 4 | 52.8 | 55.5 | 53.3 | 52.2 | 54.0 | 53.8 |
| 5 | 56.8 | 52.7 | 62.7 | 63.2 | 65.2 | 64.2 |
| 6 | 57.2 | 52.8 | 61.3 | 61.8 | 62.2 | 63.3 |
| Mean | 56.4 | 55.0 | 58.4 | 57.6 | 58.8 | 58.7 |
| − std | 1.67 | 2.75 | 4.28 | 4.94 | 5.2 | 5.65 |
| Structured | 55.5 | 57.5 | 54.4 | 52.7 | 53.7 | 53.2 |
| − std | 1.98 | 1.44 | 1.99 | 0.377 | 0.918 | 1.37 |
| Natural | 57.2 | 52.4 | 62.4 | 62.6 | 63.9 | 64.3 |
| − std | 0.368 | 0.45 | 0.804 | 0.579 | 1.26 | 0.818 |

Table 46: Accuracy per prompt template for EleutherAI-20B.

| Template | k = 0 | k = 1 | k = 5 | k = 10 | k = 15 | k = 30 |
|---|---|---|---|---|---|---|
| 1 | 53.0 | 58.0 | 55.3 | 54.3 | 52.8 | 54.3 |
| 2 | 61.3 | 54.2 | 65.8 | 63.3 | 65.0 | 60.3 |
| 3 | 54.3 | 58.3 | 58.5 | 56.7 | 55.3 | 52.0 |
| 4 | 56.2 | 58.2 | 55.3 | 57.2 | 57.0 | 58.7 |
| 5 | 59.0 | 53.0 | 66.7 | 62.8 | 65.0 | 59.2 |
| 6 | 61.3 | 53.5 | 65.2 | 61.7 | 64.0 | 59.7 |
| Mean | 57.5 | 55.9 | 61.1 | 59.3 | 59.9 | 57.4 |
| − std | 3.25 | 2.33 | 4.9 | 3.42 | 4.98 | 3.09 |
| Structured | 54.5 | 58.2 | 56.4 | 56.1 | 55.0 | 55.0 |
| − std | 1.31 | 0.125 | 1.51 | 1.27 | 1.72 | 2.78 |
| Natural | 60.5 | 53.6 | 65.9 | 62.6 | 64.7 | 59.7 |
| − std | 1.08 | 0.492 | 0.616 | 0.668 | 0.471 | 0.45 |

Table 47: Accuracy per prompt template for BLOOM-560M.

| Template | k = 0 | k = 1 | k = 5 | k = 10 | k = 15 | k = 30 |
|---|---|---|---|---|---|---|
| 1 | 54.3 | 54.2 | 53.5 | 53.8 | 53.8 | 53.5 |
| 2 | 46.7 | 56.3 | 54.0 | 54.8 | 56.0 | 55.3 |
| 3 | 58.8 | 53.3 | 53.8 | 53.3 | 54.5 | 54.0 |
| 4 | 56.3 | 54.8 | 53.5 | 54.8 | 52.7 | 56.7 |
| 5 | 46.7 | 54.3 | 53.7 | 55.3 | 56.3 | 55.5 |
| 6 | 46.7 | 56.0 | 54.0 | 55.2 | 56.7 | 55.0 |
| Mean | 51.6 | 54.8 | 53.8 | 54.5 | 55.0 | 55.0 |
| − std | 5.05 | 1.04 | 0.206 | 0.734 | 1.45 | 1.04 |
| Structured | 56.5 | 54.1 | 53.6 | 54.0 | 53.7 | 54.7 |
| − std | 1.84 | 0.616 | 0.141 | 0.624 | 0.741 | 1.41 |
| Natural | 46.7 | 55.5 | 53.9 | 55.1 | 56.3 | 55.3 |
| − std | 7.11e-15 | 0.881 | 0.141 | 0.216 | 0.287 | 0.205 |

Table 48: Accuracy per prompt template for BLOOM-1B1.

| Template | k = 0 | k = 1 | k = 5 | k = 10 | k = 15 | k = 30 |
|---|---|---|---|---|---|---|
| 1 | 53.3 | 53.5 | 56.2 | 54.2 | 55.2 | 54.5 |
| 2 | 49.0 | 51.5 | 58.2 | 59.8 | 58.8 | 60.8 |
| 3 | 57.2 | 54.2 | 55.8 | 54.0 | 55.5 | 50.8 |
| 4 | 53.3 | 54.0 | 54.2 | 53.3 | 55.7 | 55.8 |
| 5 | 47.3 | 51.2 | 59.8 | 61.3 | 60.2 | 60.0 |
| 6 | 46.8 | 51.0 | 60.2 | 61.2 | 60.2 | 59.3 |
| Mean | 51.2 | 52.6 | 57.4 | 57.3 | 57.6 | 56.9 |
| – std | 3.75 | 1.36 | 2.18 | 3.51 | 2.19 | 3.53 |
| Structured | 54.6 | 53.9 | 55.4 | 53.8 | 55.5 | 53.7 |
| – std | 1.84 | 0.294 | 0.864 | 0.386 | 0.205 | 2.12 |
| Natural | 47.7 | 51.2 | 59.4 | 60.8 | 59.7 | 60.0 |
| – std | 0.942 | 0.205 | 0.864 | 0.685 | 0.66 | 0.613 |

Table 49: Accuracy per prompt template for BLOOM-1B7.

| Template | k = 0 | k = 1 | k = 5 | k = 10 | k = 15 | k = 30 |
|---|---|---|---|---|---|---|
| 1 | 53.5 | 54.7 | 53.8 | 54.0 | 55.7 | 56.5 |
| 2 | 57.7 | 52.2 | 56.3 | 55.5 | 55.8 | 52.0 |
| 3 | 54.7 | 53.2 | 53.8 | 51.0 | 54.5 | 54.0 |
| 4 | 54.5 | 53.8 | 54.5 | 51.2 | 55.5 | 50.3 |
| 5 | 50.0 | 51.2 | 54.3 | 53.2 | 54.7 | 50.0 |
| 6 | 51.3 | 51.8 | 53.8 | 54.0 | 54.7 | 50.8 |
| Mean | 53.6 | 52.8 | 54.4 | 53.1 | 55.1 | 52.3 |
| – std | 2.49 | 1.2 | 0.886 | 1.6 | 0.528 | 2.31 |
| Structured | 54.2 | 53.9 | 54.0 | 52.1 | 55.2 | 53.6 |
| – std | 0.525 | 0.616 | 0.33 | 1.37 | 0.525 | 2.55 |
| Natural | 53.0 | 51.7 | 54.8 | 54.2 | 55.1 | 50.9 |
| – std | 3.37 | 0.411 | 1.08 | 0.953 | 0.519 | 0.822 |

Table 50: Accuracy per prompt template for BLOOM-3B.

| Template | k = 0 | k = 1 | k = 5 | k = 10 | k = 15 | k = 30 |
|---|---|---|---|---|---|---|
| 1 | 53.0 | 54.0 | 56.8 | 59.5 | 60.0 | 58.2 |
| 2 | 62.5 | 58.0 | 58.2 | 59.7 | 57.5 | 60.0 |
| 3 | 53.5 | 54.0 | 57.2 | 58.7 | 59.2 | 58.2 |
| 4 | 54.8 | 55.3 | 55.7 | 59.0 | 58.2 | 55.8 |
| 5 | 58.5 | 57.5 | 58.0 | 59.7 | 58.8 | 60.2 |
| 6 | 59.0 | 56.8 | 57.3 | 59.8 | 58.5 | 59.5 |
| Mean | 56.9 | 55.9 | 57.2 | 59.4 | 58.7 | 58.6 |
| – std | 3.4 | 1.6 | 0.823 | 0.408 | 0.783 | 1.5 |
| Structured | 53.8 | 54.4 | 56.6 | 59.1 | 59.1 | 57.4 |
| – std | 0.759 | 0.613 | 0.634 | 0.33 | 0.736 | 1.13 |
| Natural | 60.0 | 57.4 | 57.8 | 59.7 | 58.3 | 59.9 |
| – std | 1.78 | 0.492 | 0.386 | 0.0471 | 0.556 | 0.294 |

Table 51: Accuracy per prompt template for BLOOM-7B1.

| Template | k = 0 | k = 1 | k = 5 | k = 10 | k = 15 | k = 30 |
|---|---|---|---|---|---|---|
| 1 | 53.2 | 55.2 | 55.2 | 52.0 | 53.0 | 52.7 |
| 2 | 61.2 | 59.0 | 53.7 | 58.3 | 58.8 | 61.7 |
| 3 | 58.7 | 53.3 | 53.0 | 53.3 | 53.0 | 52.8 |
| 4 | 53.5 | 53.5 | 55.2 | 52.8 | 54.3 | 53.5 |
| 5 | 62.0 | 61.0 | 55.3 | 60.3 | 58.5 | 62.5 |
| 6 | 63.5 | 60.0 | 54.7 | 59.8 | 56.3 | 62.5 |
| Mean | 58.7 | 57.0 | 54.5 | 56.1 | 55.7 | 57.6 |
| – std | 4.03 | 3.11 | 0.871 | 3.46 | 2.39 | 4.63 |
| Structured | 55.1 | 54.0 | 54.5 | 52.7 | 53.4 | 53.0 |
| – std | 2.52 | 0.852 | 1.04 | 0.535 | 0.613 | 0.356 |
| Natural | 62.2 | 60.0 | 54.6 | 59.5 | 57.9 | 62.2 |
| – std | 0.953 | 0.816 | 0.66 | 0.85 | 1.11 | 0.377 |

Table 52: Accuracy per prompt template for BLOOM-176B.

| Template | k = 0 | k = 1 | k = 5 | k = 10 | k = 15 | k = 30 |
|---|---|---|---|---|---|---|
| 1 | 53.8 | 58.8 | 58.5 | 57.7 | 55.7 | 56.7 |
| 2 | 55.8 | 60.8 | 68.0 | 65.7 | 64.2 | 62.7 |
| 3 | 53.5 | 66.7 | 69.3 | 71.8 | 71.7 | 69.8 |
| 4 | 54.3 | 59.8 | 64.8 | 62.2 | 60.7 | 61.3 |
| 5 | 52.3 | 61.3 | 66.2 | 61.8 | 58.8 | 57.5 |
| 6 | 55.5 | 59.2 | 65.7 | 61.7 | 60.3 | 58.3 |
| Mean | 54.2 | 61.1 | 65.4 | 63.5 | 61.9 | 61.1 |
| – std | 1.19 | 2.65 | 3.43 | 4.38 | 5.06 | 4.44 |
| Structured | 53.9 | 61.8 | 64.2 | 63.9 | 62.7 | 62.6 |
| – std | 0.33 | 3.51 | 4.43 | 5.88 | 6.68 | 5.43 |
| Natural | 54.5 | 60.4 | 66.6 | 63.1 | 61.1 | 59.5 |
| – std | 1.58 | 0.896 | 0.988 | 1.86 | 2.28 | 2.29 |

Table 53: Accuracy per prompt template for OPT-125M.

| Template | k = 0 | k = 1 | k = 5 | k = 10 | k = 15 | k = 30 |
|---|---|---|---|---|---|---|
| 1 | 53.3 | 55.2 | 54.0 | 55.2 | 54.2 | 55.0 |
| 2 | 49.5 | 50.5 | 47.5 | 52.7 | 50.5 | 48.2 |
| 3 | 53.5 | 55.5 | 53.0 | 55.0 | 53.7 | 56.0 |
| 4 | 53.3 | 54.5 | 54.2 | 53.8 | 54.3 | 53.8 |
| 5 | 48.5 | 50.5 | 46.3 | 50.7 | 49.5 | 48.0 |
| 6 | 47.3 | 50.2 | 46.3 | 50.0 | 49.0 | 48.0 |
| Mean | 50.9 | 52.7 | 50.2 | 52.9 | 51.9 | 51.5 |
| – std | 2.55 | 2.35 | 3.56 | 1.99 | 2.25 | 3.49 |
| Structured | 53.4 | 55.1 | 53.7 | 54.7 | 54.1 | 54.9 |
| – std | 0.0943 | 0.419 | 0.525 | 0.618 | 0.262 | 0.899 |
| Natural | 48.4 | 50.4 | 46.7 | 51.1 | 49.7 | 48.1 |
| – std | 0.899 | 0.141 | 0.566 | 1.14 | 0.624 | 0.0943 |

Table 54: Accuracy per prompt template for OPT-350M.

| Template | k = 0 | k = 1 | k = 5 | k = 10 | k = 15 | k = 30 |
|---|---|---|---|---|---|---|
| 1 | 53.3 | 53.8 | 51.5 | 56.5 | 54.2 | 54.7 |
| 2 | 60.5 | 50.3 | 50.8 | 56.5 | 55.2 | 54.0 |
| 3 | 53.3 | 56.3 | 52.8 | 58.7 | 55.0 | 56.2 |
| 4 | 53.7 | 56.3 | 52.0 | 55.2 | 55.2 | 56.3 |
| 5 | 62.3 | 50.3 | 50.8 | 57.0 | 56.5 | 53.5 |
| 6 | 59.7 | 50.3 | 50.8 | 56.5 | 56.5 | 53.0 |
| Mean | 57.1 | 52.9 | 51.4 | 56.7 | 55.4 | 54.6 |
| – std | 3.78 | 2.71 | 0.752 | 1.04 | 0.826 | 1.26 |
| Structured | 53.4 | 55.5 | 52.1 | 56.8 | 54.8 | 55.7 |
| – std | 0.189 | 1.18 | 0.535 | 1.44 | 0.432 | 0.732 |
| Natural | 60.8 | 50.3 | 50.8 | 56.7 | 56.1 | 53.5 |
| – std | 1.09 | 7.11e-15 | 7.11e-15 | 0.236 | 0.613 | 0.408 |

Table 55: Accuracy per prompt template for OPT-1.3B.

| Template | k = 0 | k = 1 | k = 5 | k = 10 | k = 15 | k = 30 |
|---|---|---|---|---|---|---|
| 1 | 57.8 | 56.2 | 55.5 | 60.2 | 59.8 | 62.7 |
| 2 | 62.2 | 57.0 | 61.2 | 61.8 | 64.8 | 67.2 |
| 3 | 60.8 | 59.5 | 57.2 | 59.7 | 60.3 | 58.2 |
| 4 | 54.8 | 55.8 | 59.2 | 56.5 | 57.0 | 54.7 |
| 5 | 62.5 | 56.2 | 59.3 | 61.7 | 65.0 | 64.5 |
| 6 | 64.0 | 53.2 | 55.8 | 59.7 | 62.7 | 62.8 |
| Mean | 60.4 | 56.3 | 58.0 | 59.9 | 61.6 | 61.7 |
| – std | 3.13 | 1.85 | 2.05 | 1.76 | 2.86 | 4.11 |
| Structured | 57.8 | 57.2 | 57.3 | 58.8 | 59.0 | 58.5 |
| – std | 2.45 | 1.66 | 1.51 | 1.64 | 1.45 | 3.27 |
| Natural | 62.9 | 55.5 | 58.8 | 61.1 | 64.2 | 64.8 |
| – std | 0.787 | 1.64 | 2.24 | 0.967 | 1.04 | 1.81 |

Table 56: Accuracy per prompt template for OPT-2.7B.

| Template | k = 0 | k = 1 | k = 5 | k = 10 | k = 15 | k = 30 |
|---|---|---|---|---|---|---|
| 1 | 54.7 | 53.0 | 53.2 | 53.8 | 54.3 | 53.7 |
| 2 | 64.0 | 60.3 | 60.2 | 60.3 | 61.3 | 64.5 |
| 3 | 55.8 | 53.3 | 55.2 | 55.8 | 57.0 | 56.5 |
| 4 | 54.5 | 53.3 | 54.8 | 55.5 | 56.8 | 57.0 |
| 5 | 64.8 | 60.7 | 60.7 | 62.2 | 64.3 | 64.3 |
| 6 | 63.5 | 60.3 | 60.0 | 60.5 | 63.3 | 63.2 |
| Mean | 59.6 | 56.8 | 57.4 | 58.0 | 59.5 | 59.9 |
| – std | 4.58 | 3.62 | 3.02 | 3.11 | 3.68 | 4.28 |
| Structured | 55.0 | 53.2 | 54.4 | 55.0 | 56.0 | 55.7 |
| – std | 0.572 | 0.141 | 0.864 | 0.881 | 1.23 | 1.45 |
| Natural | 64.1 | 60.4 | 60.3 | 61.0 | 63.0 | 64.0 |
| – std | 0.535 | 0.189 | 0.294 | 0.852 | 1.25 | 0.572 |

Table 57: Accuracy per prompt template for OPT-6.7B.

| Template | k = 0 | k = 1 | k = 5 | k = 10 | k = 15 | k = 30 |
|----------|-------|-------|-------|--------|--------|--------|
| 1 | 55.7 | 54.3 | 60.8 | 61.2 | 61.2 | 58.5 |
| 2 | 64.2 | 68.0 | 66.8 | 65.7 | 66.3 | 66.3 |
| 3 | 54.2 | 53.5 | 59.5 | 61.2 | 63.3 | 60.5 |
| 4 | 58.8 | 56.3 | 61.8 | 62.2 | 63.5 | 63.2 |
| 5 | 64.2 | 65.2 | 66.0 | 65.2 | 67.7 | 67.5 |
| 6 | 65.0 | 63.2 | 64.8 | 64.3 | 66.3 | 65.7 |
| Mean | 60.4 | 60.1 | 63.3 | 63.3 | 64.7 | 63.6 |
| − std | 4.34 | 5.62 | 2.73 | 1.84 | 2.23 | 3.23 |
| Structured | 56.2 | 54.7 | 60.7 | 61.5 | 62.7 | 60.7 |
| − std | 1.92 | 1.18 | 0.942 | 0.471 | 1.04 | 1.93 |
| Natural | 64.5 | 65.5 | 65.9 | 65.1 | 66.8 | 66.5 |
| − std | 0.377 | 1.97 | 0.822 | 0.579 | 0.66 | 0.748 |

Table 58: Accuracy per prompt template for OPT-13B.

| Template | k = 0 | k = 1 | k = 5 | k = 10 | k = 15 | k = 30 |
|----------|-------|-------|-------|--------|--------|--------|
| 1 | 54.7 | 64.0 | 69.8 | 68.2 | 67.8 | 62.2 |
| 2 | 68.2 | 57.8 | 69.5 | 68.0 | 66.8 | 63.7 |
| 3 | 54.3 | 62.2 | 65.2 | 63.2 | 64.3 | 66.3 |
| 4 | 58.3 | 63.3 | 64.3 | 63.7 | 63.5 | 64.0 |
| 5 | 66.0 | 58.5 | 67.2 | 65.3 | 63.7 | 62.7 |
| 6 | 64.7 | 57.5 | 68.3 | 66.2 | 64.8 | 61.5 |
| Mean | 61.0 | 60.6 | 67.4 | 65.8 | 65.1 | 63.4 |
| − std | 5.51 | 2.68 | 2.06 | 1.92 | 1.6 | 1.55 |
| Structured | 55.8 | 63.2 | 66.4 | 65.0 | 65.2 | 64.2 |
| − std | 1.8 | 0.741 | 2.41 | 2.25 | 1.87 | 1.68 |
| Natural | 66.3 | 57.9 | 68.3 | 66.5 | 65.1 | 62.6 |
| − std | 1.44 | 0.419 | 0.939 | 1.12 | 1.28 | 0.899 |

Table 59: Accuracy per prompt template for OPT-30B.

| Template | k = 0 | k = 1 | k = 5 | k = 10 | k = 15 | k = 30 |
|----------|-------|-------|-------|--------|--------|--------|
| 1 | 62.2 | 62.7 | 66.0 | 65.2 | 65.5 | 65.0 |
| 2 | 62.0 | 58.7 | 69.0 | 65.7 | 66.3 | 69.0 |
| 3 | 60.3 | 63.5 | 62.7 | 60.8 | 60.5 | 61.5 |
| 4 | 65.0 | 66.8 | 57.8 | 57.2 | 57.2 | 56.2 |
| 5 | 60.3 | 55.8 | 70.0 | 66.0 | 67.2 | 71.0 |
| 6 | 59.0 | 54.5 | 68.3 | 65.3 | 67.7 | 70.2 |
| Mean | 61.5 | 60.3 | 65.6 | 63.4 | 64.1 | 65.5 |
| − std | 1.92 | 4.37 | 4.24 | 3.27 | 3.87 | 5.28 |
| Structured | 62.5 | 64.3 | 62.2 | 61.1 | 61.1 | 60.9 |
| − std | 1.93 | 1.77 | 3.37 | 3.27 | 3.41 | 3.62 |
| Natural | 60.4 | 56.3 | 69.1 | 65.7 | 67.1 | 70.1 |
| − std | 1.23 | 1.76 | 0.698 | 0.287 | 0.579 | 0.822 |

Table 60: Accuracy per prompt template for OPT-66B.

| Template | k = 0 | k = 1 | k = 5 | k = 10 | k = 15 | k = 30 |
|---|---|---|---|---|---|---|
| 1 | 59.3 | 56.2 | 56.7 | 56.5 | 55.7 | 54.3 |
| 2 | 66.5 | 67.3 | 65.3 | 64.2 | 67.2 | 65.2 |
| 3 | 56.5 | 64.3 | 55.5 | 55.0 | 56.2 | 52.2 |
| 4 | 62.0 | 61.5 | 66.5 | 63.0 | 61.7 | 63.7 |
| 5 | 62.5 | 66.0 | 64.8 | 63.7 | 65.7 | 65.0 |
| 6 | 61.2 | 63.8 | 60.2 | 62.5 | 64.7 | 64.7 |
| Mean | 61.3 | 63.2 | 61.5 | 60.8 | 61.9 | 60.8 |
| – std | 3.06 | 3.61 | 4.3 | 3.65 | 4.5 | 5.43 |
| Structured | 59.3 | 60.7 | 59.6 | 58.2 | 57.9 | 56.7 |
| – std | 2.25 | 3.36 | 4.93 | 3.47 | 2.72 | 5.0 |
| Natural | 63.4 | 65.7 | 63.4 | 63.5 | 65.9 | 65.0 |
| – std | 2.26 | 1.44 | 2.3 | 0.713 | 1.03 | 0.205 |

Table 61: Accuracy per prompt template for OPT-175B.

| Template | k = 0 | k = 1 | k = 5 | k = 10 | k = 15 | k = 30 |
|---|---|---|---|---|---|---|
| 1 | 56.7 | 58.0 | 64.8 | 61.0 | 65.0 | 62.3 |
| 2 | 52.7 | 53.3 | 67.3 | 63.2 | 68.0 | 65.8 |
| 3 | 54.5 | 68.5 | 60.0 | 55.3 | 57.8 | 56.7 |
| 4 | 64.0 | 66.7 | 61.5 | 58.0 | 62.0 | 58.7 |
| 5 | 52.0 | 52.0 | 65.0 | 63.8 | 67.8 | 65.2 |
| 6 | 52.2 | 51.7 | 64.7 | 63.2 | 68.0 | 66.0 |
| Mean | 55.3 | 58.4 | 63.9 | 60.8 | 64.8 | 62.4 |
| – std | 4.19 | 6.87 | 2.42 | 3.13 | 3.79 | 3.62 |
| Structured | 58.4 | 64.4 | 62.1 | 58.1 | 61.6 | 59.2 |
| – std | 4.06 | 4.58 | 2.0 | 2.33 | 2.95 | 2.32 |
| Natural | 52.3 | 52.3 | 65.7 | 63.4 | 67.9 | 65.7 |
| – std | 0.294 | 0.694 | 1.16 | 0.283 | 0.0943 | 0.34 |

Table 62: Accuracy per prompt template for Cohere-409.3M (Cohere-small).

| Template | k = 0 | k = 1 | k = 5 | k = 10 | k = 15 | k = 30 |
|---|---|---|---|---|---|---|
| 1 | 54.2 | 49.7 | 52.7 | 51.7 | 53.5 | 56.0 |
| 2 | 47.5 | 50.7 | 52.7 | 53.2 | 55.8 | 57.8 |
| 3 | 57.2 | 55.5 | 55.2 | 55.5 | 55.7 | 57.0 |
| 4 | 54.8 | 53.8 | 54.5 | 56.8 | 54.8 | 54.5 |
| 5 | 48.5 | 50.7 | 52.8 | 52.7 | 56.0 | 58.8 |
| 6 | 47.5 | 51.0 | 52.5 | 53.7 | 55.3 | 58.8 |
| Mean | 51.6 | 51.9 | 53.4 | 53.9 | 55.2 | 57.2 |
| – std | 3.91 | 2.05 | 1.05 | 1.72 | 0.847 | 1.54 |
| Structured | 55.4 | 53.0 | 54.1 | 54.7 | 54.7 | 55.8 |
| – std | 1.3 | 2.43 | 1.05 | 2.16 | 0.903 | 1.03 |
| Natural | 47.8 | 50.8 | 52.7 | 53.2 | 55.7 | 58.5 |
| – std | 0.471 | 0.141 | 0.125 | 0.408 | 0.294 | 0.471 |

Table 63: Accuracy per prompt template for Cohere-6.067B (Cohere-medium).

| Template | k = 0 | k = 1 | k = 5 | k = 10 | k = 15 | k = 30 |
|---|---|---|---|---|---|---|
| 1 | 54.7 | 54.2 | 55.3 | 51.8 | 56.3 | 55.3 |
| 2 | 61.8 | 62.8 | 64.3 | 63.8 | 65.2 | 64.7 |
| 3 | 57.2 | 53.3 | 58.5 | 55.3 | 57.8 | 55.3 |
| 4 | 56.0 | 53.3 | 57.0 | 53.2 | 55.8 | 56.7 |
| 5 | 57.8 | 60.7 | 64.0 | 64.2 | 64.7 | 64.2 |
| 6 | 56.2 | 62.8 | 66.2 | 64.0 | 62.8 | 66.0 |
| Mean | 57.3 | 57.9 | 60.9 | 58.7 | 60.4 | 60.4 |
| − std | 2.24 | 4.32 | 4.11 | 5.38 | 3.92 | 4.65 |
| Structured | 56.0 | 53.6 | 56.9 | 53.4 | 56.6 | 55.8 |
| − std | 1.02 | 0.424 | 1.31 | 1.44 | 0.85 | 0.66 |
| Natural | 58.6 | 62.1 | 64.8 | 64.0 | 64.2 | 65.0 |
| − std | 2.36 | 0.99 | 0.974 | 0.163 | 1.03 | 0.759 |

Table 64: Accuracy per prompt template for Cohere-13.12B (Cohere-large).

| Template | k = 0 | k = 1 | k = 5 | k = 10 | k = 15 | k = 30 |
|---|---|---|---|---|---|---|
| 1 | 55.3 | 57.3 | 56.3 | 55.0 | 58.5 | 59.0 |
| 2 | 59.2 | 64.2 | 68.0 | 66.3 | 64.7 | 69.5 |
| 3 | 57.2 | 62.8 | 61.0 | 59.0 | 64.2 | 62.3 |
| 4 | 55.5 | 61.3 | 56.3 | 54.0 | 59.0 | 59.8 |
| 5 | 56.8 | 64.3 | 66.7 | 64.2 | 65.7 | 69.8 |
| 6 | 59.2 | 60.7 | 66.5 | 63.7 | 65.0 | 68.3 |
| Mean | 57.2 | 61.8 | 62.5 | 60.4 | 62.9 | 64.8 |
| − std | 1.56 | 2.41 | 4.88 | 4.69 | 2.94 | 4.55 |
| Structured | 56.0 | 60.5 | 57.9 | 56.0 | 60.6 | 60.4 |
| − std | 0.852 | 2.32 | 2.22 | 2.16 | 2.58 | 1.41 |
| Natural | 58.4 | 63.1 | 67.1 | 64.7 | 65.1 | 69.2 |
| − std | 1.13 | 1.67 | 0.665 | 1.13 | 0.419 | 0.648 |

Table 65: Accuracy per prompt template for Cohere-52B (Cohere-xl).

| Template | k = 0 | k = 1 | k = 5 | k = 10 | k = 15 | k = 30 |
|---|---|---|---|---|---|---|
| 1 | 56.0 | 60.7 | 70.3 | 65.3 | 66.3 | 68.7 |
| 2 | 62.8 | 65.0 | 64.3 | 64.2 | 65.0 | 64.3 |
| 3 | 54.0 | 65.3 | 62.8 | 60.2 | 64.0 | 63.5 |
| 4 | 53.8 | 55.5 | 61.8 | 64.8 | 64.3 | 64.7 |
| 5 | 62.2 | 65.7 | 67.3 | 63.0 | 63.7 | 65.3 |
| 6 | 62.2 | 65.7 | 64.2 | 62.3 | 65.0 | 67.8 |
| Mean | 58.5 | 63.0 | 65.1 | 63.3 | 64.7 | 65.7 |
| − std | 3.97 | 3.77 | 2.87 | 1.72 | 0.855 | 1.89 |
| Structured | 54.6 | 60.5 | 65.0 | 63.4 | 64.9 | 65.6 |
| − std | 0.993 | 4.0 | 3.79 | 2.3 | 1.02 | 2.22 |
| Natural | 62.4 | 65.5 | 65.3 | 63.2 | 64.6 | 65.8 |
| − std | 0.283 | 0.33 | 1.44 | 0.785 | 0.613 | 1.47 |

Table 66: Accuracy per prompt template for GPT-3-350M (ada).

| Template | k = 0 | k = 1 | k = 5 | k = 10 | k = 15 | k = 30 |
|---|---|---|---|---|---|---|
| 1 | 55.3 | 57.2 | 58.3 | 57.5 | 58.2 | 60.5 |
| 2 | 46.7 | 56.8 | 56.3 | 59.5 | 59.2 | 61.7 |
| 3 | 54.0 | 54.5 | 53.3 | 54.0 | 56.5 | 56.7 |
| 4 | 53.5 | 52.8 | 54.7 | 56.7 | 58.8 | 59.7 |
| 5 | 49.8 | 57.3 | 55.3 | 58.5 | 58.8 | 61.8 |
| 6 | 49.5 | 57.2 | 56.3 | 60.2 | 61.5 | 61.2 |
| Mean | 51.5 | 56.0 | 55.7 | 57.7 | 58.8 | 60.3 |
| − std | 3.02 | 1.72 | 1.55 | 2.04 | 1.48 | 1.75 |
| Structured | 54.3 | 54.8 | 55.4 | 56.1 | 57.8 | 59.0 |
| − std | 0.759 | 1.81 | 2.11 | 1.5 | 0.974 | 1.64 |
| Natural | 48.7 | 57.1 | 56.0 | 59.4 | 59.8 | 61.6 |
| − std | 1.4 | 0.216 | 0.471 | 0.698 | 1.19 | 0.262 |

Table 67: Accuracy per prompt template for GPT-3-1.3B (babbage).

| Template | k = 0 | k = 1 | k = 5 | k = 10 | k = 15 | k = 30 |
|---|---|---|---|---|---|---|
| 1 | 55.7 | 60.7 | 61.0 | 59.0 | 60.7 | 57.8 |
| 2 | 63.0 | 62.5 | 65.7 | 61.7 | 63.0 | 59.3 |
| 3 | 56.2 | 59.0 | 60.5 | 59.3 | 64.8 | 61.0 |
| 4 | 53.3 | 59.7 | 60.7 | 62.5 | 65.0 | 66.7 |
| 5 | 59.2 | 62.5 | 63.7 | 61.8 | 61.5 | 58.7 |
| 6 | 59.0 | 60.2 | 64.3 | 61.2 | 62.2 | 57.7 |
| Mean | 57.7 | 60.8 | 62.6 | 60.9 | 62.9 | 60.2 |
| − std | 3.1 | 1.33 | 2.01 | 1.31 | 1.6 | 3.11 |
| Structured | 55.1 | 59.8 | 60.7 | 60.3 | 63.5 | 61.8 |
| − std | 1.27 | 0.698 | 0.205 | 1.58 | 1.98 | 3.68 |
| Natural | 60.4 | 61.7 | 64.6 | 61.6 | 62.2 | 58.6 |
| − std | 1.84 | 1.08 | 0.838 | 0.262 | 0.613 | 0.66 |

Table 68: Accuracy per prompt template for GPT-3-6.7B (curie).

| Template | k = 0 | k = 1 | k = 5 | k = 10 | k = 15 | k = 30 |
|---|---|---|---|---|---|---|
| 1 | 53.3 | 58.3 | 63.0 | 64.8 | 67.7 | 64.0 |
| 2 | 57.5 | 65.2 | 63.2 | 65.3 | 65.8 | 65.2 |
| 3 | 57.0 | 54.2 | 59.2 | 61.2 | 60.8 | 59.3 |
| 4 | 53.3 | 61.7 | 62.8 | 63.8 | 64.7 | 60.7 |
| 5 | 55.3 | 64.2 | 62.5 | 64.5 | 65.8 | 63.7 |
| 6 | 52.5 | 63.5 | 63.7 | 64.0 | 66.2 | 64.3 |
| Mean | 54.8 | 61.2 | 62.4 | 63.9 | 65.2 | 62.9 |
| − std | 1.92 | 3.83 | 1.48 | 1.32 | 2.14 | 2.12 |
| Structured | 54.5 | 58.1 | 61.7 | 63.3 | 64.4 | 61.3 |
| − std | 1.74 | 3.07 | 1.75 | 1.52 | 2.82 | 1.97 |
| Natural | 55.1 | 64.3 | 63.1 | 64.6 | 65.9 | 64.4 |
| − std | 2.05 | 0.698 | 0.492 | 0.535 | 0.189 | 0.616 |

Table 69: Accuracy per prompt template for GPT-3-175B (davinci).

| Template | k = 0 | k = 1 | k = 5 | k = 10 | k = 15 | k = 30 |
|---|---|---|---|---|---|---|
| 1 | 61.2 | 67.3 | 66.3 | 62.7 | 66.7 | 66.2 |
| 2 | 53.7 | 65.3 | 68.8 | 69.3 | 71.0 | 69.7 |
| 3 | 58.7 | 65.8 | 68.2 | 64.7 | 65.0 | 65.3 |
| 4 | 64.0 | 62.8 | 71.3 | 68.7 | 66.2 | 67.8 |
| 5 | 54.2 | 66.3 | 69.0 | 70.0 | 70.0 | 70.8 |
| 6 | 51.7 | 66.7 | 68.7 | 68.3 | 71.0 | 70.0 |
| Mean | 57.2 | 65.7 | 68.7 | 67.3 | 68.3 | 68.3 |
| − std | 4.4 | 1.44 | 1.46 | 2.65 | 2.43 | 2.03 |
| Structured | 61.3 | 65.3 | 68.6 | 65.4 | 66.0 | 66.4 |
| − std | 2.16 | 1.87 | 2.06 | 2.49 | 0.713 | 1.03 |
| Natural | 53.2 | 66.1 | 68.8 | 69.2 | 70.7 | 70.2 |
| − std | 1.08 | 0.589 | 0.125 | 0.698 | 0.471 | 0.464 |

Table 70: Accuracy per prompt template for BlenderBot-90M.

| Template | k = 0 | k = 1 | k = 5 | k = 10 | k = 15 | k = 30 |
|---|---|---|---|---|---|---|
| 1 | 46.7 | 51.5 | 46.7 | 46.7 | 46.5 | 46.5 |
| 2 | 46.7 | 51.3 | 46.5 | 46.7 | 46.7 | 46.7 |
| 3 | 46.7 | 46.7 | 46.7 | 46.7 | 46.3 | 46.8 |
| 4 | 46.7 | 46.7 | 46.7 | 46.7 | 46.5 | 46.7 |
| 5 | 46.7 | 50.0 | 46.7 | 46.7 | 46.7 | 46.7 |
| 6 | 46.5 | 53.5 | 46.3 | 46.7 | 46.7 | 46.7 |
| Mean | 46.7 | 49.9 | 46.6 | 46.7 | 46.6 | 46.7 |
| − std | 0.0745 | 2.52 | 0.153 | 7.11e-15 | 0.149 | 0.0898 |
| Structured | 46.7 | 48.3 | 46.7 | 46.7 | 46.4 | 46.7 |
| − std | 7.11e-15 | 2.26 | 7.11e-15 | 7.11e-15 | 0.0943 | 0.125 |
| Natural | 46.6 | 51.6 | 46.5 | 46.7 | 46.7 | 46.7 |
| − std | 0.0943 | 1.44 | 0.163 | 7.11e-15 | 7.11e-15 | 7.11e-15 |

Table 71: Accuracy per prompt template for BlenderBot-2.7B.

| Template | k = 0 | k = 1 | k = 5 | k = 10 | k = 15 | k = 30 |
|---|---|---|---|---|---|---|
| 1 | 54.0 | 53.2 | 53.3 | 53.0 | 52.8 | 53.3 |
| 2 | 53.3 | 53.3 | 53.3 | 53.3 | 53.3 | 53.3 |
| 3 | 53.2 | 53.2 | 53.3 | 53.2 | 53.2 | 53.2 |
| 4 | 53.5 | 53.5 | 53.5 | 53.3 | 52.8 | 53.0 |
| 5 | 53.3 | 53.3 | 53.3 | 53.3 | 53.3 | 53.3 |
| 6 | 53.3 | 53.3 | 53.3 | 53.3 | 53.3 | 53.3 |
| Mean | 53.4 | 53.3 | 53.3 | 53.2 | 53.1 | 53.2 |
| − std | 0.269 | 0.1 | 0.0745 | 0.111 | 0.227 | 0.111 |
| Structured | 53.6 | 53.3 | 53.4 | 53.2 | 52.9 | 53.2 |
| − std | 0.33 | 0.141 | 0.0943 | 0.125 | 0.189 | 0.125 |
| Natural | 53.3 | 53.3 | 53.3 | 53.3 | 53.3 | 53.3 |
| − std | 7.11e-15 | 7.11e-15 | 7.11e-15 | 7.11e-15 | 7.11e-15 | 7.11e-15 |

Table 72: Accuracy per prompt template for BlenderBot-9.4B.

| Template | k = 0 | k = 1 | k = 5 | k = 10 | k = 15 | k = 30 |
|---|---|---|---|---|---|---|
| 1 | 53.7 | 51.5 | 53.0 | 53.0 | 53.0 | 54.0 |
| 2 | 53.2 | 53.8 | 54.2 | 52.5 | 52.2 | 52.2 |
| 3 | 53.3 | 49.7 | 52.0 | 54.0 | 54.2 | 55.5 |
| 4 | 54.0 | 55.3 | 52.5 | 54.0 | 53.5 | 53.7 |
| 5 | 53.3 | 52.8 | 53.5 | 53.2 | 53.5 | 53.3 |
| 6 | 52.7 | 52.0 | 51.7 | 53.5 | 52.8 | 53.7 |
| Mean | 53.4 | 52.5 | 52.8 | 53.4 | 53.2 | 53.7 |
| – std | 0.407 | 1.77 | 0.859 | 0.537 | 0.63 | 0.978 |
| Structured | 53.7 | 52.2 | 52.5 | 53.7 | 53.6 | 54.4 |
| – std | 0.287 | 2.33 | 0.408 | 0.471 | 0.492 | 0.787 |
| Natural | 53.1 | 52.9 | 53.1 | 53.1 | 52.8 | 53.1 |
| – std | 0.262 | 0.736 | 1.05 | 0.419 | 0.531 | 0.634 |

Table 73: Accuracy per prompt template for T0-3B.

| Template | k = 0 | k = 1 | k = 5 | k = 10 | k = 15 | k = 30 |
|---|---|---|---|---|---|---|
| 1 | 48.7 | 49.5 | 46.5 | 46.7 | 46.7 | 46.7 |
| 2 | 46.7 | 47.5 | 46.7 | 46.7 | 46.7 | 46.7 |
| 3 | 49.2 | 48.3 | 46.7 | 46.7 | 46.7 | 46.7 |
| 4 | 51.7 | 49.0 | 46.7 | 46.7 | 46.7 | 46.7 |
| 5 | 46.7 | 49.2 | 46.7 | 46.7 | 46.7 | 46.7 |
| 6 | 46.7 | 49.8 | 46.8 | 46.7 | 46.7 | 46.7 |
| Mean | 48.3 | 48.9 | 46.7 | 46.7 | 46.7 | 46.7 |
| – std | 1.84 | 0.773 | 0.0898 | 7.11e-15 | 7.11e-15 | 7.11e-15 |
| Structured | 49.9 | 48.9 | 46.6 | 46.7 | 46.7 | 46.7 |
| – std | 1.31 | 0.492 | 0.0943 | 7.11e-15 | 7.11e-15 | 7.11e-15 |
| Natural | 46.7 | 48.8 | 46.7 | 46.7 | 46.7 | 46.7 |
| – std | 7.11e-15 | 0.974 | 0.0471 | 7.11e-15 | 7.11e-15 | 7.11e-15 |

Table 74: Accuracy per prompt template for T0-11B.

| Template | k = 0 | k = 1 | k = 5 | k = 10 | k = 15 | k = 30 |
|---|---|---|---|---|---|---|
| 1 | 57.5 | 47.7 | 47.3 | 46.8 | 46.7 | 46.7 |
| 2 | 49.3 | 47.5 | 46.7 | 46.7 | 46.8 | 46.7 |
| 3 | 65.3 | 48.8 | 47.3 | 46.7 | 46.7 | 46.7 |
| 4 | 63.8 | 48.0 | 47.0 | 46.7 | 46.7 | 46.7 |
| 5 | 48.0 | 47.2 | 46.7 | 46.7 | 47.0 | 46.8 |
| 6 | 49.7 | 47.5 | 47.0 | 46.8 | 47.0 | 47.0 |
| Mean | 55.6 | 47.8 | 47.0 | 46.7 | 46.8 | 46.8 |
| – std | 7.04 | 0.515 | 0.245 | 0.0471 | 0.134 | 0.111 |
| Structured | 62.2 | 48.2 | 47.2 | 46.7 | 46.7 | 46.7 |
| – std | 3.38 | 0.464 | 0.141 | 0.0471 | 7.11e-15 | 7.11e-15 |
| Natural | 49.0 | 47.4 | 46.8 | 46.7 | 46.9 | 46.8 |
| – std | 0.726 | 0.141 | 0.141 | 0.0471 | 0.0943 | 0.125 |

Table 75: Accuracy per prompt template for Flan-T5-780M.

| Template | k = 0 | k = 1 | k = 5 | k = 10 | k = 15 | k = 30 |
|---|---|---|---|---|---|---|
| 1 | 64.5 | 63.3 | 62.2 | 60.7 | 61.5 | 60.2 |
| 2 | 66.5 | 65.8 | 65.3 | 62.8 | 65.5 | 65.0 |
| 3 | 61.7 | 60.2 | 58.8 | 60.8 | 59.8 | 59.7 |
| 4 | 58.0 | 50.2 | 50.7 | 51.3 | 52.3 | 54.8 |
| 5 | 63.8 | 69.0 | 64.3 | 63.2 | 65.2 | 65.5 |
| 6 | 65.3 | 68.8 | 64.8 | 62.3 | 64.7 | 63.8 |
| Mean | 63.3 | 62.9 | 61.0 | 60.2 | 61.5 | 61.5 |
| – std | 2.79 | 6.44 | 5.1 | 4.08 | 4.61 | 3.73 |
| Structured | 61.4 | 57.9 | 57.2 | 57.6 | 57.9 | 58.2 |
| – std | 2.66 | 5.59 | 4.82 | 4.45 | 4.0 | 2.44 |
| Natural | 65.2 | 67.9 | 64.8 | 62.8 | 65.1 | 64.8 |
| – std | 1.1 | 1.46 | 0.408 | 0.368 | 0.33 | 0.713 |

Table 76: Accuracy per prompt template for Flan-T5-3B.

| Template | k = 0 | k = 1 | k = 5 | k = 10 | k = 15 | k = 30 |
|---|---|---|---|---|---|---|
| 1 | 54.7 | 58.8 | 56.8 | 56.7 | 57.5 | 60.0 |
| 2 | 51.2 | 50.8 | 59.0 | 59.2 | 59.0 | 59.7 |
| 3 | 54.8 | 51.3 | 49.7 | 49.0 | 48.7 | 48.5 |
| 4 | 55.3 | 50.0 | 48.0 | 49.0 | 49.3 | 50.8 |
| 5 | 51.0 | 54.3 | 57.2 | 58.0 | 58.0 | 57.8 |
| 6 | 48.0 | 51.2 | 58.7 | 59.0 | 58.0 | 59.8 |
| Mean | 52.5 | 52.7 | 54.9 | 55.1 | 55.1 | 56.1 |
| – std | 2.65 | 3.02 | 4.37 | 4.42 | 4.33 | 4.67 |
| Structured | 54.9 | 53.4 | 51.5 | 51.6 | 51.8 | 53.1 |
| – std | 0.262 | 3.88 | 3.81 | 3.63 | 4.01 | 4.97 |
| Natural | 50.1 | 52.1 | 58.3 | 58.7 | 58.3 | 59.1 |
| – std | 1.46 | 1.56 | 0.787 | 0.525 | 0.471 | 0.92 |

Table 77: Accuracy per prompt template for Flan-T5-11B.

| Template | k = 0 | k = 1 | k = 5 | k = 10 | k = 15 | k = 30 |
|---|---|---|---|---|---|---|
| 1 | 64.3 | 61.0 | 63.7 | 65.0 | 62.5 | 64.3 |
| 2 | 61.5 | 59.7 | 63.2 | 62.3 | 64.0 | 68.0 |
| 3 | 56.5 | 63.0 | 60.2 | 57.3 | 56.7 | 56.8 |
| 4 | 61.7 | 47.7 | 51.7 | 50.3 | 50.3 | 49.5 |
| 5 | 61.5 | 55.8 | 64.8 | 64.7 | 65.5 | 66.3 |
| 6 | 59.2 | 57.5 | 66.3 | 63.7 | 66.0 | 67.7 |
| Mean | 60.8 | 57.4 | 61.7 | 60.5 | 60.8 | 62.1 |
| – std | 2.42 | 4.94 | 4.82 | 5.25 | 5.62 | 6.78 |
| Structured | 60.8 | 57.2 | 58.5 | 57.5 | 56.5 | 56.9 |
| – std | 3.24 | 6.79 | 5.04 | 6.0 | 4.98 | 6.04 |
| Natural | 60.7 | 57.7 | 64.8 | 63.6 | 65.2 | 67.3 |
| – std | 1.08 | 1.6 | 1.27 | 0.984 | 0.85 | 0.741 |

Table 78: Accuracy per prompt template for Cohere-command-6b.

| Template | k = 0 | k = 1 | k = 5 | k = 10 | k = 15 | k = 30 |
|---|---|---|---|---|---|---|
| 1 | 65.0 | 63.2 | 71.7 | 70.2 | 71.3 | 70.3 |
| 2 | 64.8 | 64.2 | 66.8 | 67.5 | 69.8 | 71.7 |
| 3 | 68.0 | 65.5 | 69.2 | 65.5 | 66.8 | 68.2 |
| 4 | 70.0 | 68.5 | 69.2 | 71.2 | 71.7 | 73.2 |
| 5 | 66.3 | 65.0 | 66.8 | 67.5 | 70.5 | 69.8 |
| 6 | 63.7 | 63.7 | 67.7 | 67.5 | 70.0 | 69.5 |
| Mean | 66.3 | 65.0 | 68.6 | 68.2 | 70.0 | 70.5 |
| − std | 2.13 | 1.74 | 1.71 | 1.9 | 1.59 | 1.61 |
| Structured | 67.7 | 65.7 | 70.0 | 69.0 | 69.9 | 70.6 |
| − std | 2.05 | 2.17 | 1.18 | 2.49 | 2.22 | 2.05 |
| Natural | 64.9 | 64.3 | 67.1 | 67.5 | 70.1 | 70.3 |
| − std | 1.07 | 0.535 | 0.424 | 0.0 | 0.294 | 0.974 |

Table 79: Accuracy per prompt template for Cohere-command-52b.

| Template | k = 0 | k = 1 | k = 5 | k = 10 | k = 15 | k = 30 |
|---|---|---|---|---|---|---|
| 1 | 65.2 | 74.2 | 77.8 | 75.8 | 75.5 | 76.0 |
| 2 | 61.7 | 72.0 | 73.5 | 75.3 | 75.2 | 74.5 |
| 3 | 56.7 | 74.5 | 77.3 | 77.2 | 76.5 | 75.0 |
| 4 | 68.2 | 70.7 | 76.0 | 74.8 | 75.3 | 75.3 |
| 5 | 54.8 | 72.7 | 74.8 | 76.2 | 74.8 | 74.7 |
| 6 | 54.8 | 73.0 | 73.0 | 74.5 | 75.0 | 74.5 |
| Mean | 60.2 | 72.8 | 75.4 | 75.6 | 75.4 | 75.0 |
| − std | 5.19 | 1.29 | 1.8 | 0.903 | 0.546 | 0.529 |
| Structured | 63.4 | 73.1 | 77.0 | 75.9 | 75.8 | 75.4 |
| − std | 4.87 | 1.72 | 0.759 | 0.984 | 0.525 | 0.419 |
| Natural | 57.1 | 72.6 | 73.8 | 75.3 | 75.0 | 74.6 |
| − std | 3.25 | 0.419 | 0.759 | 0.694 | 0.163 | 0.0943 |

Table 80: Accuracy per prompt template for text-ada-001-unknown.

| Template | k = 0 | k = 1 | k = 5 | k = 10 | k = 15 | k = 30 |
|---|---|---|---|---|---|---|
| 1 | 60.8 | 62.8 | 60.8 | 59.0 | 58.7 | 58.8 |
| 2 | 50.7 | 56.3 | 54.8 | 56.0 | 57.7 | 52.7 |
| 3 | 63.7 | 58.5 | 60.8 | 59.0 | 56.7 | 57.5 |
| 4 | 61.8 | 56.3 | 59.3 | 58.3 | 61.0 | 56.7 |
| 5 | 53.3 | 55.5 | 55.2 | 55.7 | 58.0 | 54.3 |
| 6 | 48.7 | 54.7 | 54.7 | 56.2 | 57.7 | 53.5 |
| Mean | 56.5 | 57.3 | 57.6 | 57.4 | 58.3 | 55.6 |
| − std | 5.82 | 2.7 | 2.75 | 1.43 | 1.34 | 2.22 |
| Structured | 62.1 | 59.2 | 60.3 | 58.8 | 58.8 | 57.7 |
| − std | 1.2 | 2.7 | 0.707 | 0.33 | 1.76 | 0.865 |
| Natural | 50.9 | 55.5 | 54.9 | 56.0 | 57.8 | 53.5 |
| − std | 1.88 | 0.653 | 0.216 | 0.205 | 0.141 | 0.653 |

Table 81: Accuracy per prompt template for text-babbage-001-unknown.

| Template | k = 0 | k = 1 | k = 5 | k = 10 | k = 15 | k = 30 |
|----------|-------|-------|-------|--------|--------|--------|
| 1 | 67.5 | 64.0 | 66.3 | 63.0 | 64.0 | 64.7 |
| 2 | 63.0 | 62.5 | 66.2 | 64.2 | 66.5 | 68.2 |
| 3 | 65.3 | 65.2 | 66.0 | 63.2 | 64.7 | 64.5 |
| 4 | 65.2 | 63.5 | 65.7 | 62.7 | 63.0 | 64.8 |
| 5 | 61.8 | 64.3 | 66.5 | 64.0 | 66.3 | 67.8 |
| 6 | 64.0 | 63.8 | 66.2 | 64.2 | 66.7 | 66.0 |
| Mean | 64.5 | 63.9 | 66.1 | 63.6 | 65.2 | 66.0 |
| − std | 1.82 | 0.815 | 0.25 | 0.605 | 1.4 | 1.5 |
| Structured | 66.0 | 64.2 | 66.0 | 63.0 | 63.9 | 64.7 |
| − std | 1.06 | 0.713 | 0.245 | 0.205 | 0.698 | 0.125 |
| Natural | 62.9 | 63.5 | 66.3 | 64.1 | 66.5 | 67.3 |
| − std | 0.899 | 0.759 | 0.141 | 0.0943 | 0.163 | 0.957 |

Table 82: Accuracy per prompt template for text-curie-001-unknown.

| Template | k = 0 | k = 1 | k = 5 | k = 10 | k = 15 | k = 30 |
|----------|-------|-------|-------|--------|--------|--------|
| 1 | 70.7 | 70.2 | 72.5 | 70.8 | 70.8 | 70.7 |
| 2 | 66.5 | 59.3 | 70.3 | 69.7 | 68.3 | 71.2 |
| 3 | 73.2 | 70.2 | 73.5 | 69.7 | 71.8 | 69.7 |
| 4 | 71.3 | 68.0 | 71.0 | 69.8 | 71.0 | 69.0 |
| 5 | 65.5 | 58.8 | 70.0 | 70.2 | 68.5 | 70.7 |
| 6 | 66.5 | 59.8 | 70.7 | 70.8 | 69.0 | 70.8 |
| Mean | 69.0 | 64.4 | 71.3 | 70.2 | 69.9 | 70.4 |
| − std | 2.9 | 5.14 | 1.25 | 0.478 | 1.35 | 0.754 |
| Structured | 71.7 | 69.5 | 72.3 | 70.1 | 71.2 | 69.8 |
| − std | 1.07 | 1.04 | 1.03 | 0.497 | 0.432 | 0.698 |
| Natural | 66.2 | 59.3 | 70.3 | 70.2 | 68.6 | 70.9 |
| − std | 0.471 | 0.408 | 0.287 | 0.45 | 0.294 | 0.216 |

Table 83: Accuracy per prompt template for text-davinci-001-unknown.

| Template | k = 0 | k = 1 | k = 5 | k = 10 | k = 15 | k = 30 |
|----------|-------|-------|-------|--------|--------|--------|
| 1 | 76.5 | 73.7 | 75.7 | 75.7 | 76.3 | 76.8 |
| 2 | 72.0 | 72.5 | 74.3 | 75.2 | 76.0 | 75.3 |
| 3 | 74.8 | 74.2 | 75.7 | 77.2 | 75.8 | 76.8 |
| 4 | 68.0 | 70.2 | 72.8 | 72.8 | 73.3 | 75.0 |
| 5 | 72.5 | 73.2 | 74.3 | 74.3 | 75.3 | 75.7 |
| 6 | 70.0 | 72.7 | 74.3 | 74.7 | 75.0 | 75.3 |
| Mean | 72.3 | 72.7 | 74.5 | 75.0 | 75.3 | 75.8 |
| − std | 2.82 | 1.28 | 0.991 | 1.34 | 0.986 | 0.724 |
| Structured | 73.1 | 72.7 | 74.7 | 75.2 | 75.1 | 76.2 |
| − std | 3.67 | 1.78 | 1.37 | 1.83 | 1.31 | 0.849 |
| Natural | 71.5 | 72.8 | 74.3 | 74.7 | 75.4 | 75.4 |
| − std | 1.08 | 0.294 | 0.0 | 0.368 | 0.419 | 0.189 |

Table 84: Accuracy per prompt template for text-davinci-002-unknown.

| Template | k = 0 | k = 1 | k = 5 | k = 10 | k = 15 | k = 30 |
|---|---|---|---|---|---|---|
| 1 | 73.7 | 76.2 | 80.2 | 79.5 | 79.8 | 80.7 |
| 2 | 69.5 | 73.5 | 78.2 | 78.5 | 76.7 | 79.8 |
| 3 | 73.0 | 78.7 | 82.8 | 82.8 | 82.7 | 82.8 |
| 4 | 71.3 | 79.7 | 80.5 | 80.8 | 82.0 | 81.5 |
| 5 | 67.5 | 72.5 | 79.2 | 79.2 | 77.0 | 79.8 |
| 6 | 68.5 | 73.2 | 76.5 | 76.5 | 76.2 | 79.2 |
| Mean | 70.6 | 75.6 | 79.6 | 79.5 | 79.1 | 80.6 |
| − std | 2.28 | 2.79 | 1.96 | 1.94 | 2.6 | 1.22 |
| Structured | 72.7 | 78.2 | 81.2 | 81.0 | 81.5 | 81.7 |
| − std | 1.01 | 1.47 | 1.16 | 1.36 | 1.24 | 0.865 |
| Natural | 68.5 | 73.1 | 78.0 | 78.1 | 76.6 | 79.6 |
| − std | 0.816 | 0.419 | 1.11 | 1.14 | 0.33 | 0.283 |

Table 85: Accuracy per prompt template for text-davinci-003-unknown.

| Template | k = 0 | k = 1 | k = 5 | k = 10 | k = 15 | k = 30 |
|---|---|---|---|---|---|---|
| 1 | 74.3 | 71.7 | 79.8 | 80.2 | 80.7 | 80.3 |
| 2 | 71.8 | 75.0 | 80.2 | 78.8 | 78.2 | 78.3 |
| 3 | 71.8 | 73.7 | 79.7 | 79.5 | 79.2 | 81.2 |
| 4 | 65.2 | 74.2 | 78.5 | 78.2 | 79.7 | 79.5 |
| 5 | 72.2 | 75.3 | 80.2 | 78.5 | 78.2 | 78.8 |
| 6 | 72.2 | 76.0 | 79.7 | 78.8 | 78.3 | 79.0 |
| Mean | 71.2 | 74.3 | 79.7 | 79.0 | 79.0 | 79.5 |
| − std | 2.84 | 1.38 | 0.57 | 0.666 | 0.929 | 0.975 |
| Structured | 70.4 | 73.2 | 79.3 | 79.3 | 79.9 | 80.3 |
| − std | 3.84 | 1.08 | 0.591 | 0.829 | 0.624 | 0.694 |
| Natural | 72.1 | 75.4 | 80.0 | 78.7 | 78.2 | 78.7 |
| − std | 0.189 | 0.419 | 0.236 | 0.141 | 0.0471 | 0.294 |

Table 86: Accuracy per prompt template for ChatGPT-unknown.

| Template | k = 0 | k = 1 | k = 5 | k = 10 | k = 15 | k = 30 |
|---|---|---|---|---|---|---|
| 1 | 77.8 | 73.3 | 72.7 | 72.7 | 74.2 | 74.5 |
| 2 | 73.2 | 76.2 | 78.7 | 78.2 | 79.7 | 79.2 |
| 3 | 72.7 | 74.0 | 74.3 | 74.7 | 75.0 | 74.8 |
| 4 | 59.3 | 73.7 | 60.8 | 63.5 | 66.0 | 68.0 |
| 5 | 74.7 | 76.8 | 77.8 | 77.7 | 79.3 | 78.8 |
| 6 | 74.8 | 76.7 | 79.0 | 79.0 | 79.2 | 78.5 |
| Mean | 72.1 | 75.1 | 73.9 | 74.3 | 75.6 | 75.6 |
| − std | 5.94 | 1.48 | 6.29 | 5.29 | 4.79 | 3.9 |
| Structured | 69.9 | 73.7 | 69.3 | 70.3 | 71.7 | 72.4 |
| − std | 7.8 | 0.287 | 6.02 | 4.88 | 4.07 | 3.14 |
| Natural | 74.2 | 76.6 | 78.5 | 78.3 | 79.4 | 78.8 |
| − std | 0.732 | 0.262 | 0.51 | 0.535 | 0.216 | 0.287 |

Table 87: Accuracy per prompt template for GPT-4-unknown.

| Template | k = 0 | k = 1 | k = 5 | k = 10 | k = 15 | k = 30 |
|---|---|---|---|---|---|---|
| 1 | 83.3 | 82.7 | 84.0 | 84.2 | 85.5 | 84.5 |
| 2 | 81.8 | 80.8 | 80.8 | 78.0 | 79.3 | 79.7 |
| 3 | 84.7 | 83.7 | 84.2 | 85.3 | 85.5 | 85.3 |
| 4 | 80.5 | 84.3 | 82.5 | 84.3 | 83.3 | 83.7 |
| 5 | 79.5 | 81.0 | 80.8 | 77.0 | 79.0 | 79.0 |
| 6 | 80.8 | 81.3 | 79.8 | 79.0 | 79.8 | 80.8 |
| Mean | 81.8 | 82.3 | 82.0 | 81.3 | 82.1 | 82.2 |
| – std | 1.76 | 1.36 | 1.67 | 3.37 | 2.81 | 2.44 |
| Structured | 82.8 | 83.6 | 83.6 | 84.6 | 84.8 | 84.5 |
| – std | 1.75 | 0.66 | 0.759 | 0.497 | 1.04 | 0.653 |
| Natural | 80.7 | 81.0 | 80.5 | 78.0 | 79.4 | 79.8 |
| – std | 0.942 | 0.205 | 0.471 | 0.816 | 0.33 | 0.741 |

## L   Timestamps API calls

For reproducibility purposes, Table 88, 89, and 90 contain the dates and times the APIs from OpenAI and Cohere were queried for the results.

Table 88: Timestamp each was evaluated through OpenAI's API (1/2).

| model | timestamp |
|---|---|
| GPT-3-ada/0-shot | 2022-09-22 13:13:29 |
| GPT-3-ada/1-shot | 2022-09-22 15:11:13 |
| GPT-3-ada/5-shot | 2022-09-22 15:40:12 |
| GPT-3-ada/10-shot | 2022-09-22 18:14:18 |
| GPT-3-ada/15-shot | 2022-09-22 19:15:29 |
| GPT-3-ada/30-shot | 2022-09-22 22:47:58 |
| GPT-3-babbage/0-shot | 2022-09-22 23:19:05 |
| GPT-3-babbage/1-shot | 2022-09-22 23:39:53 |
| GPT-3-babbage/5-shot | 2022-09-23 00:01:32 |
| GPT-3-babbage/10-shot | 2022-09-23 00:24:27 |
| GPT-3-babbage/15-shot | 2022-09-23 00:49:13 |
| GPT-3-babbage/30-shot | 2022-09-23 01:15:44 |
| GPT-3-curie/0-shot | 2022-09-22 14:04:32 |
| GPT-3-curie/1-shot | 2022-09-23 02:09:14 |
| GPT-3-curie/5-shot | 2022-09-23 02:32:20 |
| GPT-3-curie/10-shot | 2022-09-23 02:56:43 |
| GPT-3-curie/15-shot | 2022-09-23 03:23:19 |
| GPT-3-curie/30-shot | 2022-09-23 03:52:30 |
| GPT-3-davinci/0-shot | 2022-09-22 12:21:48 |
| GPT-3-davinci/1-shot | 2022-09-23 14:27:15 |
| GPT-3-davinci/5-shot | 2022-09-23 15:10:40 |
| GPT-3-davinci/10-shot | 2022-09-23 16:04:53 |
| GPT-3-davinci/15-shot | 2022-09-23 17:17:04 |
| GPT-3-davinci/30-shot | 2022-09-23 18:36:38 |
| OpenAI-text-ada-001/0-shot | 2022-08-17 16:59:45 |
| OpenAI-text-ada-001/1-shot | 2022-08-17 18:23:12 |
| OpenAI-text-ada-001/5-shot | 2022-08-17 19:16:48 |
| OpenAI-text-ada-001/10-shot | 2022-08-17 20:24:16 |
| OpenAI-text-ada-001/15-shot | 2022-08-17 21:21:46 |
| OpenAI-text-ada-001/30-shot | 2022-08-17 22:44:47 |
| OpenAI-text-babbage-001/0-shot | 2022-08-17 11:50:44 |
| OpenAI-text-babbage-001/1-shot | 2022-08-17 12:22:08 |
| OpenAI-text-babbage-001/5-shot | 2022-08-17 12:50:59 |
| OpenAI-text-babbage-001/10-shot | 2022-08-17 13:27:52 |
| OpenAI-text-babbage-001/15-shot | 2022-08-17 14:57:43 |
| OpenAI-text-babbage-001/30-shot | 2022-08-17 15:45:16 |
| OpenAI-text-curie-001/0-shot | 2022-08-18 04:39:55 |
| OpenAI-text-curie-001/1-shot | 2022-08-18 05:10:17 |
| OpenAI-text-curie-001/5-shot | 2022-08-18 05:40:56 |
| OpenAI-text-curie-001/10-shot | 2022-08-18 06:15:28 |
| OpenAI-text-curie-001/15-shot | 2022-08-18 06:53:09 |
| OpenAI-text-curie-001/30-shot | 2022-08-18 07:35:40 |
| OpenAI-text-davinci-001/0-shot | 2022-08-26 20:26:21 |
| OpenAI-text-davinci-001/1-shot | 2022-08-26 21:02:31 |
| OpenAI-text-davinci-001/5-shot | 2022-08-26 21:35:19 |
| OpenAI-text-davinci-001/10-shot | 2022-08-27 07:14:02 |
| OpenAI-text-davinci-001/15-shot | 2022-08-27 07:58:25 |
| OpenAI-text-davinci-001/30-shot | 2022-08-27 08:44:42 |

Table 89: Timestamp each was evaluated through OpenAI's API - continued (2/2).

| model | timestamp |
|---|---|
| OpenAI-text-davinci-002/0-shot | 2022-08-10 21:41:50 |
| OpenAI-text-davinci-002/1-shot | 2022-08-11 10:04:17 |
| OpenAI-text-davinci-002/5-shot | 2022-08-12 15:41:45 |
| OpenAI-text-davinci-002/10-shot | 2022-08-12 16:41:14 |
| OpenAI-text-davinci-002/15-shot | 2022-08-16 12:11:43 |
| OpenAI-text-davinci-002/30-shot | 2022-08-16 14:35:38 |
| OpenAI-text-davinci-003/0-shot | 2023-03-15 11:35:23 |
| OpenAI-text-davinci-003/1-shot | 2023-04-04 13:12:05 |
| OpenAI-text-davinci-003/5-shot | 2023-03-15 12:30:39 |
| OpenAI-text-davinci-003/10-shot | 2023-04-04 14:01:03 |
| OpenAI-text-davinci-003/15-shot | 2023-04-04 15:23:29 |
| OpenAI-text-davinci-003/30-shot | 2023-04-06 15:08:38 |
| OpenAI-gpt-3.5.turbo/0-shot | 2023-04-05 13:33:09 |
| OpenAI-gpt-3.5.turbo/1-shot | 2023-04-05 16:36:45 |
| OpenAI-gpt-3.5.turbo/5-shot | 2023-04-06 08:46:09 |
| OpenAI-gpt-3.5.turbo/10-shot | 2023-04-06 09:54:07 |
| OpenAI-gpt-3.5.turbo/15-shot | 2023-04-06 10:57:18 |
| OpenAI-gpt-3.5.turbo/30-shot | 2023-04-06 12:03:59 |
| OpenAI-gpt-4/0-shot | 2023-04-06 17:38:16 |
| OpenAI-gpt-4/1-shot | 2023-04-06 19:41:59 |
| OpenAI-gpt-4/5-shot | 2023-04-06 22:56:31 |
| OpenAI-gpt-4/10-shot | 2023-04-08 12:06:03 |
| OpenAI-gpt-4/15-shot | 2023-04-08 17:32:04 |
| OpenAI-gpt-4/30-shot | 2023-04-08 19:56:26 |

Table 90: Timestamp each model was evaluated through Cohere's API.

| model | timestamp |
|---|---|
| Cohere-small/0-shot | 2022-08-16 22:22:17 |
| Cohere-small/1-shot | 2022-08-17 08:22:43 |
| Cohere-small/5-shot | 2022-08-17 09:19:57 |
| Cohere-small/10-shot | 2022-08-17 10:43:53 |
| Cohere-small/15-shot | 2022-08-17 12:53:02 |
| Cohere-small/30-shot | 2022-08-17 13:46:08 |
| Cohere-medium/0-shot | 2022-08-17 15:14:02 |
| Cohere-medium/1-shot | 2022-08-17 16:00:21 |
| Cohere-medium/5-shot | 2022-08-17 18:23:38 |
| Cohere-medium/10-shot | 2022-08-17 19:16:00 |
| Cohere-medium/15-shot | 2022-08-17 20:24:12 |
| Cohere-medium/30-shot | 2022-08-17 21:20:28 |
| Cohere-large/0-shot | 2022-08-17 22:47:49 |
| Cohere-large/1-shot | 2022-08-17 23:27:00 |
| Cohere-large/5-shot | 2022-08-18 00:10:08 |
| Cohere-large/10-shot | 2022-08-18 00:56:55 |
| Cohere-large/15-shot | 2022-08-18 01:48:30 |
| Cohere-large/30-shot | 2022-08-18 02:47:14 |
| Cohere-xl/0-shot | 2022-07-29 |
| Cohere-xl/1-shot | 2022-07-31 |
| Cohere-xl/5-shot | 2022-08-02 |
| Cohere-xl/10-shot | 2022-08-02 15:16:45 |
| Cohere-xl/15-shot | 2022-08-07 13:55:44 |
| Cohere-xl/30-shot | 2022-08-16 19:51:08 |
| Cohere-command-medium/0-shot | 2023-04-04 09:54:27 |
| Cohere-command-medium/1-shot | 2023-04-04 11:51:07 |
| Cohere-command-medium/5-shot | 2023-04-04 13:03:07 |
| Cohere-command-medium/10-shot | 2023-04-04 13:31:47 |
| Cohere-command-medium/15-shot | 2023-04-04 14:06:10 |
| Cohere-command-medium/30-shot | 2023-04-04 14:42:13 |
| Cohere-command-xl/0-shot | 2023-04-04 10:25:30 |
| Cohere-command-xl/1-shot | 2023-04-04 15:27:01 |
| Cohere-command-xl/5-shot | 2023-04-04 15:59:47 |
| Cohere-command-xl/10-shot | 2023-04-04 16:36:22 |
| Cohere-command-xl/15-shot | 2023-04-04 17:22:58 |
| Cohere-command-xl/30-shot | 2023-04-04 18:16:54 |

Table 91: Timestamp, duration, and emissions per experiment with non-API models. (1/4)

| model | timestamp | duration |
|---|---|---|
| EleutherAI-125m-0-shot | 2022-09-01T21:33:14 | 8549.649220 |
| EleutherAI-125m-1-shot | 2022-09-02T00:10:03 | 640.861120 |
| EleutherAI-125m-5-shot | 2022-09-02T00:26:27 | 982.369876 |
| EleutherAI-125m-10-shot | 2022-09-02T00:51:24 | 1495.525381 |
| EleutherAI-125m-15-shot | 2022-09-02T01:29:03 | 2257.290708 |
| EleutherAI-125m-30-shot | 2022-09-02T09:04:03 | 27298.375266 |
| EleutherAI-2.7b-0-shot | 2022-09-03T00:36:14 | 3752.897449 |
| EleutherAI-2.7b-1-shot | 2022-09-03T02:04:16 | 5279.884696 |
| EleutherAI-2.7b-5-shot | 2022-09-03T04:28:19 | 8641.654516 |
| EleutherAI-2.7b-10-shot | 2022-09-03T08:18:13 | 13792.592126 |
| EleutherAI-2.7b-15-shot | 2022-09-03T13:33:25 | 18909.551123 |
| EleutherAI-2.7b-30-shot | 2022-09-03T22:47:06 | 33219.682098 |
| EleutherAI-20b-0-shot | 2022-08-25T07:40:55 | 1378.197924 |
| EleutherAI-20b-1-shot | 2022-08-25T08:15:23 | 807.702344 |
| EleutherAI-20b-5-shot | 2022-08-25T15:39:51 | 859.585535 |
| EleutherAI-20b-10-shot | 2022-08-25T16:18:50 | 1175.128651 |
| EleutherAI-20b-15-shot | 2022-08-25T16:47:30 | 1713.266182 |
| EleutherAI-20b-30-shot | 2022-08-25T17:45:28 | 3469.811664 |
| EleutherAI-6b-0-shot | 2022-08-24T22:29:30 | 1287.627453 |
| EleutherAI-6b-1-shot | 2022-08-24T23:22:30 | 1831.554774 |
| EleutherAI-6b-5-shot | 2022-08-25T00:16:57 | 3255.128955 |
| EleutherAI-6b-10-shot | 2022-08-25T01:23:21 | 3971.650578 |
| EleutherAI-6b-15-shot | 2022-08-25T02:26:23 | 3772.113814 |
| EleutherAI-6b-30-shot | 2022-08-25T04:18:30 | 6719.419030 |
| EleutherAI-1.3b-0-shot | 2022-09-02T09:54:06 | 3000.666020 |
| EleutherAI-1.3b-1-shot | 2022-09-02T10:46:30 | 3142.207699 |
| EleutherAI-1.3b-5-shot | 2022-09-02T12:25:25 | 5933.046596 |
| EleutherAI-1.3b-10-shot | 2022-09-02T12:39:00 | 8509.257493 |
| EleutherAI-1.3b-15-shot | 2022-09-02T18:00:39 | 11615.289366 |
| EleutherAI-1.3b-30-shot | 2022-09-02T23:33:39 | 19978.306457 |

## M   Compute and Emissions

Find below in Table 91 until Table 94 the timestamps, durations, and emissions per experiment (calculated with the CodeCarbon library in Python). Find below in Table 95 until Table 98 the cpu-type and count and gpu-type and count per experiment. In terms of compute the following GPU hours can be estimated if we assume each run is entirely done on the GPU (which is not true in reality, but worst case):

NVIDIA A100-SXM4-40GB used for 926.4291392151515 hours.

Tesla V100-PCIE-32GB used for 29.282544113265143 hours.

Tesla V100-PCIE-16GB used for 11.462701331244574 hours.

Table 92: Timestamp, duration, and emissions per experiment with non-API models. (2/4)

| model | timestamp | duration |
|---|---|---|
| BLOOM-3b-0-shot | 2022-08-31T12:54:37 | 5178.369790 |
| BLOOM-3b-1-shot | 2022-08-31T14:39:32 | 6292.560350 |
| BLOOM-3b-5-shot | 2022-08-31T17:37:29 | 10675.230701 |
| BLOOM-3b-10-shot | 2022-08-31T21:59:27 | 15715.744792 |
| BLOOM-3b-15-shot | 2022-09-01T03:41:02 | 20492.823278 |
| BLOOM-3b-30-shot | 2022-09-01T15:47:21 | 43577.882397 |
| BLOOM-7b1-0-shot | 2022-08-25T04:56:35 | 625.931470 |
| BLOOM-7b1-1-shot | 2022-08-25T05:07:13 | 630.628939 |
| BLOOM-7b1-5-shot | 2022-08-25T05:24:22 | 1022.138932 |
| BLOOM-7b1-10-shot | 2022-08-25T05:49:00 | 1471.008220 |
| BLOOM-7b1-15-shot | 2022-08-25T06:23:26 | 2058.455127 |
| BLOOM-7b1-30-shot | 2022-08-25T07:29:46 | 3972.772039 |
| BLOOM-560m-0-shot | 2022-08-29T15:35:52 | 2541.248956 |
| BLOOM-560m-1-shot | 2022-08-29T18:52:16 | 2532.794568 |
| BLOOM-560m-5-shot | 2022-08-29T20:16:16 | 5038.547060 |
| BLOOM-560m-10-shot | 2022-08-29T22:17:43 | 7285.239875 |
| BLOOM-560m-15-shot | 2022-08-30T00:38:23 | 8438.096533 |
| BLOOM-560m-30-shot | 2022-08-30T04:38:44 | 14419.447170 |
| BLOOM-1b1-0-shot | 2022-08-30T05:18:44 | 2398.828856 |
| BLOOM-1b1-1-shot | 2022-08-30T06:06:45 | 2879.435828 |
| BLOOM-1b1-5-shot | 2022-08-30T07:35:59 | 5352.607075 |
| BLOOM-1b1-10-shot | 2022-08-30T10:15:02 | 9541.535419 |
| BLOOM-1b1-15-shot | 2022-08-30T13:22:42 | 11257.077128 |
| BLOOM-1b1-30-shot | 2022-08-30T18:08:15 | 17131.797610 |
| BLOOM-176b-0-shot | 2022-10-14T12:51:11 | 3015.240235 |
| BLOOM-176b-1-shot | 2022-10-14T13:57:53 | 3906.461752 |
| BLOOM-176b-5-shot | 2022-10-14T20:41:10 | 7411.725385 |
| BLOOM-176b-10-shot | 2022-10-23T21:43:21 | 14462.201855 |
| BLOOM-176b-15-shot | 2022-10-24T01:14:10 | 12609.026736 |
| BLOOM-176b-30-shot | 2022-10-14T20:47:02 | 33159.499966 |

Table 93: Timestamp, duration, and emissions per experiment with non-API models. (3/4)

| model | timestamp | duration |
|---|---|---|
| OPT-13b-0-shot | 2022-08-25T07:07:08 | 878.202579 |
| OPT-13b-1-shot | 2022-08-25T07:31:30 | 458.133617 |
| OPT-13b-5-shot | 2022-08-25T07:37:39 | 578.308507 |
| OPT-13b-10-shot | 2022-08-25T08:01:50 | 821.158826 |
| OPT-13b-15-shot | 2022-08-25T08:20:49 | 1131.479665 |
| OPT-13b-30-shot | 2022-08-25T16:05:27 | 2235.869414 |
| OPT-350m-0-shot | 2022-09-16T17:26:28 | 389.173905 |
| OPT-350m-1-shot | 2022-09-16T17:33:42 | 424.832551 |
| OPT-350m-5-shot | 2022-09-16T18:00:14 | 1583.824094 |
| OPT-350m-10-shot | 2022-09-16T18:32:12 | 1908.822462 |
| OPT-350m-15-shot | 2022-09-16T19:03:23 | 1863.625027 |
| OPT-350m-30-shot | 2022-09-16T19:47:29 | 2637.811867 |
| OPT-125m-0-shot | 2022-09-16T15:15:56 | 273.178967 |
| OPT-125m-1-shot | 2022-09-16T15:20:28 | 259.680856 |
| OPT-125m-5-shot | 2022-09-16T15:41:37 | 1259.801105 |
| OPT-125m-10-shot | 2022-09-16T16:09:59 | 1693.598805 |
| OPT-125m-15-shot | 2022-09-16T16:41:46 | 1899.415318 |
| OPT-125m-30-shot | 2022-09-16T17:19:51 | 2276.441314 |
| OPT-6.7b-0-shot | 2022-08-24T23:03:07 | 1140.485014 |
| OPT-6.7b-1-shot | 2022-08-24T23:17:51 | 872.225225 |
| OPT-6.7b-5-shot | 2022-08-24T23:34:40 | 995.894396 |
| OPT-6.7b-10-shot | 2022-08-24T23:55:44 | 1252.956499 |
| OPT-6.7b-15-shot | 2022-08-25T00:23:04 | 1627.749039 |
| OPT-6.7b-30-shot | 2022-08-25T01:05:49 | 2553.054289 |
| OPT-2.7b-0-shot | 2022-09-18T16:35:05 | 686.197892 |
| OPT-2.7b-1-shot | 2022-09-18T16:45:11 | 593.508211 |
| OPT-2.7b-5-shot | 2022-09-18T17:12:11 | 1613.313387 |
| OPT-2.7b-10-shot | 2022-09-18T17:44:48 | 1949.808232 |
| OPT-2.7b-15-shot | 2022-09-18T18:22:02 | 2225.927837 |
| OPT-2.7b-30-shot | 2022-09-18T19:09:05 | 2815.327871 |
| OPT-30b-0-shot | 2022-08-25T19:03:37 | 591.665447 |
| OPT-30b-1-shot | 2022-08-25T19:14:32 | 645.923823 |
| OPT-30b-5-shot | 2022-08-25T16:44:22 | 1825.821606 |
| OPT-30b-10-shot | 2022-08-25T17:07:22 | 1372.752916 |
| OPT-30b-15-shot | 2022-08-25T17:41:05 | 2015.006104 |
| OPT-30b-30-shot | 2022-08-25T18:10:39 | 3859.078056 |
| OPT-1.3b-0-shot | 2022-09-17T17:53:50 | 595.193443 |
| OPT-1.3b-1-shot | 2022-09-17T18:03:45 | 579.367790 |
| OPT-1.3b-5-shot | 2022-09-17T18:33:18 | 1759.103432 |
| OPT-1.3b-10-shot | 2022-09-17T19:12:19 | 2327.300123 |
| OPT-1.3b-15-shot | 2022-09-17T19:48:32 | 2161.637401 |
| OPT-1.3b-30-shot | 2022-09-17T20:37:00 | 2893.829010 |
| OPT-175b-0-shot | 2022-10-19T15:02:56 | 2387.104187 |
| OPT-175b-1-shot | 2022-10-19T16:34:06 | 1589.972279 |
| OPT-175b-5-shot | 2022-10-19T17:25:58 | 3072.591171 |
| OPT-175b-10-shot | 2022-10-19T17:33:15 | 6211.692086 |
| OPT-175b-15-shot | 2022-10-19T21:29:16 | 8019.585246 |
| OPT-175b-30-shot | 2022-10-19T21:36:53 | 19901.470347 |
| OPT-66b-0-shot | 2022-08-25T18:58:11 | 2834.901372 |
| OPT-66b-1-shot | 2022-08-25T19:22:09 | 1427.806986 |
| OPT-66b-5-shot | 2022-08-25T19:47:39 | 1521.168440 |
| OPT-66b-10-shot | 2022-08-25T20:24:56 | 2228.407874 |
| OPT-66b-15-shot | 2022-08-25T21:41:21 | 3370.689256 |
| OPT-66b-30-shot | 2022-08-26T00:31:36 | 6816.312183 |

Table 94: Timestamp, duration, and emissions per experiment with non-API models. (4/4)

| model | timestamp | duration |
|---|---|---|
| BlenderBot-2.7b-0-shot | 2022-09-04T08:09:56 | 3656.381540 |
| BlenderBot-2.7b-1-shot | 2022-09-12T15:58:01 | 4051.858183 |
| BlenderBot-2.7b-5-shot | 2022-09-12T17:16:20 | 4696.628979 |
| BlenderBot-2.7b-10-shot | 2022-09-12T18:35:53 | 4772.083818 |
| BlenderBot-2.7b-15-shot | 2022-09-12T19:54:13 | 4698.638356 |
| BlenderBot-2.7b-30-shot | 2022-09-12T21:10:34 | 4579.460884 |
| BlenderBot-9.4b-0-shot | 2022-10-22T04:04:24 | 614.201131 |
| BlenderBot-9.4b-1-shot | 2022-10-22T17:17:21 | 659.975971 |
| BlenderBot-9.4b-5-shot | 2022-10-22T17:31:48 | 839.336277 |
| BlenderBot-9.4b-10-shot | 2022-10-22T17:46:18 | 843.852691 |
| BlenderBot-9.4b-15-shot | 2022-10-22T17:53:41 | 1262.038660 |
| BlenderBot-9.4b-30-shot | 2022-10-22T18:23:25 | 853.334728 |
| BlenderBot-90m-0-shot | 2022-09-14T15:11:44 | 273.134700 |
| BlenderBot-90m-1-shot | 2022-09-14T15:17:38 | 351.542638 |
| BlenderBot-90m-5-shot | 2022-09-14T15:29:50 | 730.774348 |
| BlenderBot-90m-10-shot | 2022-09-14T15:47:22 | 1050.647882 |
| BlenderBot-90m-15-shot | 2022-09-14T16:07:27 | 1204.079804 |
| BlenderBot-90m-30-shot | 2022-09-14T16:28:55 | 1285.913686 |
| T0-3b-0-shot | 2022-10-21T17:33:36 | 348.245298 |
| T0-3b-1-shot | 2022-10-24T23:20:57 | 350.730799 |
| T0-3b-5-shot | 2022-10-24T23:29:21 | 474.378557 |
| T0-3b-10-shot | 2022-10-25T15:56:54 | 676.111759 |
| T0-3b-15-shot | 2022-10-25T16:12:55 | 928.215524 |
| T0-3b-30-shot | 2022-10-24T23:30:17 | 1961.897054 |
| T0-11b-0-shot | 2022-10-21T15:38:13 | 2289.815276 |
| T0-11b-1-shot | 2022-10-22T19:18:25 | 814.872760 |
| T0-11b-5-shot | 2022-10-22T19:41:45 | 1368.644314 |
| T0-11b-10-shot | 2022-10-22T20:17:30 | 2112.628515 |
| T0-11b-15-shot | 2022-10-22T21:06:30 | 2904.655213 |
| T0-11b-30-shot | 2022-10-22T22:41:16 | 5648.105648 |
| Flan-T5-3b-0-shot | 2022-10-24T11:20:36 | 617.820384 |
| Flan-T5-3b-1-shot | 2022-10-25T12:29:59 | 348.405589 |
| Flan-T5-3b-5-shot | 2022-10-25T12:38:24 | 474.872964 |
| Flan-T5-3b-10-shot | 2022-10-25T12:50:00 | 665.592482 |
| Flan-T5-3b-15-shot | 2022-10-25T13:05:34 | 902.197151 |
| Flan-T5-3b-30-shot | 2022-10-25T13:37:14 | 1864.885266 |
| Flan-T5-780m-0-shot | 2022-10-24T11:54:09 | 160.503411 |
| Flan-T5-780m-1-shot | 2022-10-25T14:41:28 | 3816.321305 |
| Flan-T5-780m-5-shot | 2022-10-25T14:46:09 | 251.699700 |
| Flan-T5-780m-10-shot | 2022-10-25T14:52:09 | 331.340966 |
| Flan-T5-780m-15-shot | 2022-10-25T14:59:00 | 381.107934 |
| Flan-T5-780m-30-shot | 2022-10-25T15:11:18 | 705.711192 |
| Flan-T5-11b-0-shot | 2022-10-24T10:25:09 | 1111.283857 |
| Flan-T5-11b-1-shot | 2022-10-24T10:56:52 | 654.411412 |
| Flan-T5-11b-5-shot | 2022-10-25T17:26:50 | 1403.159768 |
| Flan-T5-11b-10-shot | 2022-10-25T18:29:59 | 3756.529085 |
| Flan-T5-11b-15-shot | 2022-10-25T19:21:15 | 3042.271478 |
| Flan-T5-11b-30-shot | 2022-10-25T20:57:13 | 5722.244579 |

Table 95: Compute used per experiment with non-API models. (1/4)

| model | cpus | cpu model | gpu model |
|---|---|---|---|
| EleutherAI-125m-0-shot | 10 | Apple M1 Max | |
| EleutherAI-125m-1-shot | 10 | Apple M1 Max | |
| EleutherAI-125m-5-shot | 10 | Apple M1 Max | |
| EleutherAI-125m-10-shot | 10 | Apple M1 Max | |
| EleutherAI-125m-15-shot | 10 | Apple M1 Max | |
| EleutherAI-125m-30-shot | 10 | Apple M1 Max | |
| EleutherAI-2.7b-0-shot | 10 | Apple M1 Max | |
| EleutherAI-2.7b-1-shot | 10 | Apple M1 Max | |
| EleutherAI-2.7b-5-shot | 10 | Apple M1 Max | |
| EleutherAI-2.7b-10-shot | 10 | Apple M1 Max | |
| EleutherAI-2.7b-15-shot | 10 | Apple M1 Max | |
| EleutherAI-2.7b-30-shot | 10 | Apple M1 Max | |
| EleutherAI-20b-0-shot | 48 | Intel(R) Xeon(R) Platinum 8275CL CPU @ 3.00GHz | 8 x NVIDIA A100-40GB |
| EleutherAI-20b-1-shot | 1 | Intel(R) Xeon(R) Platinum 8275CL CPU @ 3.00GHz | 8 x NVIDIA A100-40GB |
| EleutherAI-20b-5-shot | 48 | Intel(R) Xeon(R) Platinum 8275CL CPU @ 3.00GHz | 8 x NVIDIA A100-40GB |
| EleutherAI-20b-10-shot | 48 | Intel(R) Xeon(R) Platinum 8275CL CPU @ 3.00GHz | 8 x NVIDIA A100-40GB |
| EleutherAI-20b-15-shot | 48 | Intel(R) Xeon(R) Platinum 8275CL CPU @ 3.00GHz | 8 x NVIDIA A100-40GB |
| EleutherAI-20b-30-shot | 48 | Intel(R) Xeon(R) Platinum 8275CL CPU @ 3.00GHz | 8 x NVIDIA A100-40GB |
| EleutherAI-6b-0-shot | 1 | Intel(R) Xeon(R) Platinum 8275CL CPU @ 3.00GHz | 8 x NVIDIA A100-40GB |
| EleutherAI-6b-1-shot | 1 | Intel(R) Xeon(R) Platinum 8275CL CPU @ 3.00GHz | 8 x NVIDIA A100-40GB |
| EleutherAI-6b-5-shot | 1 | Intel(R) Xeon(R) Platinum 8275CL CPU @ 3.00GHz | 8 x NVIDIA A100-40GB |
| EleutherAI-6b-10-shot | 1 | Intel(R) Xeon(R) Platinum 8275CL CPU @ 3.00GHz | 8 x NVIDIA A100-40GB |
| EleutherAI-6b-15-shot | 1 | Intel(R) Xeon(R) Platinum 8275CL CPU @ 3.00GHz | 8 x NVIDIA A100-40GB |
| EleutherAI-6b-30-shot | 1 | Intel(R) Xeon(R) Platinum 8275CL CPU @ 3.00GHz | 8 x NVIDIA A100-40GB |
| EleutherAI-1.3b-0-shot | 10 | Apple M1 Max | |
| EleutherAI-1.3b-1-shot | 10 | Apple M1 Max | |
| EleutherAI-1.3b-5-shot | 10 | Apple M1 Max | |
| EleutherAI-1.3b-10-shot | 10 | Apple M1 Max | |
| EleutherAI-1.3b-15-shot | 10 | Apple M1 Max | |
| EleutherAI-1.3b-30-shot | 10 | Apple M1 Max | |

Table 96: Compute used per experiment with non-API models. (2/4)

| model | cpus | cpu model | gpu model |
|---|---|---|---|
| BLOOM-3b-0-shot | 10 | Apple M1 Max | |
| BLOOM-3b-1-shot | 10 | Apple M1 Max | |
| BLOOM-3b-5-shot | 10 | Apple M1 Max | |
| BLOOM-3b-10-shot | 10 | Apple M1 Max | |
| BLOOM-3b-15-shot | 10 | Apple M1 Max | |
| BLOOM-3b-30-shot | 10 | Apple M1 Max | |
| BLOOM-7b1-0-shot | 1 | Intel(R) Xeon(R) Platinum 8275CL CPU @ 3.00GHz | 8 x NVIDIA A100-40GB |
| BLOOM-7b1-1-shot | 1 | Intel(R) Xeon(R) Platinum 8275CL CPU @ 3.00GHz | 8 x NVIDIA A100-40GB |
| BLOOM-7b1-5-shot | 1 | Intel(R) Xeon(R) Platinum 8275CL CPU @ 3.00GHz | 8 x NVIDIA A100-40GB |
| BLOOM-7b1-10-shot | 1 | Intel(R) Xeon(R) Platinum 8275CL CPU @ 3.00GHz | 8 x NVIDIA A100-40GB |
| BLOOM-7b1-15-shot | 1 | Intel(R) Xeon(R) Platinum 8275CL CPU @ 3.00GHz | 8 x NVIDIA A100-40GB |
| BLOOM-7b1-30-shot | 1 | Intel(R) Xeon(R) Platinum 8275CL CPU @ 3.00GHz | 8 x NVIDIA A100-40GB |
| BLOOM-560m-0-shot | 10 | Apple M1 Max | |
| BLOOM-560m-1-shot | 10 | Apple M1 Max | |
| BLOOM-560m-5-shot | 10 | Apple M1 Max | |
| BLOOM-560m-10-shot | 10 | Apple M1 Max | |
| BLOOM-560m-15-shot | 10 | Apple M1 Max | |
| BLOOM-560m-30-shot | 10 | Apple M1 Max | |
| BLOOM-1b1-0-shot | 10 | Apple M1 Max | |
| BLOOM-1b1-1-shot | 10 | Apple M1 Max | |
| BLOOM-1b1-5-shot | 10 | Apple M1 Max | |
| BLOOM-1b1-10-shot | 10 | Apple M1 Max | |
| BLOOM-1b1-15-shot | 10 | Apple M1 Max | |
| BLOOM-1b1-30-shot | 10 | Apple M1 Max | |
| BLOOM-176b-0-shot | 96 | Intel(R) Xeon(R) CPU @ 2.20GHz | 16 x NVIDIA A100-40GB |
| BLOOM-176b-1-shot | 96 | Intel(R) Xeon(R) CPU @ 2.20GHz | 16 x NVIDIA A100-40GB |
| BLOOM-176b-5-shot | 96 | Intel(R) Xeon(R) CPU @ 2.20GHz | 16 x NVIDIA A100-40GB |
| BLOOM-176b-10-shot | 96 | Intel(R) Xeon(R) CPU @ 2.20GHz | 16 x NVIDIA A100-40GB |
| BLOOM-176b-15-shot | 96 | Intel(R) Xeon(R) CPU @ 2.20GHz | 16 x NVIDIA A100-40GB |
| BLOOM-176b-30-shot | 96 | Intel(R) Xeon(R) CPU @ 2.20GHz | 16 x NVIDIA A100-40GB |

Table 97: Compute used per experiment with non-API models. (3/4)

| model | cpus | cpu model | gpu model |
|---|---|---|---|
| OPT-13b-0-shot | 48 | Intel(R) Xeon(R) Platinum 8275CL CPU @ 3.00GHz | 8 x NVIDIA A100-40GB |
| OPT-13b-1-shot | 48 | Intel(R) Xeon(R) Platinum 8275CL CPU @ 3.00GHz | 8 x NVIDIA A100-40GB |
| OPT-13b-5-shot | 1 | Intel(R) Xeon(R) Platinum 8275CL CPU @ 3.00GHz | 8 x NVIDIA A100-40GB |
| OPT-13b-10-shot | 48 | Intel(R) Xeon(R) Platinum 8275CL CPU @ 3.00GHz | 8 x NVIDIA A100-40GB |
| OPT-13b-15-shot | 48 | Intel(R) Xeon(R) Platinum 8275CL CPU @ 3.00GHz | 8 x NVIDIA A100-40GB |
| OPT-13b-30-shot | 48 | Intel(R) Xeon(R) Platinum 8275CL CPU @ 3.00GHz | 8 x NVIDIA A100-40GB |
| OPT-350m-0-shot | 24 | Intel(R) Xeon(R) Gold 5118 CPU @ 2.30GHz | 4 x Tesla V100-PCIE-32GB |
| OPT-350m-1-shot | 24 | Intel(R) Xeon(R) Gold 5118 CPU @ 2.30GHz | 4 x Tesla V100-PCIE-32GB |
| OPT-350m-5-shot | 24 | Intel(R) Xeon(R) Gold 5118 CPU @ 2.30GHz | 4 x Tesla V100-PCIE-32GB |
| OPT-350m-10-shot | 24 | Intel(R) Xeon(R) Gold 5118 CPU @ 2.30GHz | 4 x Tesla V100-PCIE-32GB |
| OPT-350m-15-shot | 24 | Intel(R) Xeon(R) Gold 5118 CPU @ 2.30GHz | 4 x Tesla V100-PCIE-32GB |
| OPT-350m-30-shot | 24 | Intel(R) Xeon(R) Gold 5118 CPU @ 2.30GHz | 4 x Tesla V100-PCIE-32GB |
| OPT-125m-0-shot | 24 | Intel(R) Xeon(R) Gold 5118 CPU @ 2.30GHz | 4 x Tesla V100-PCIE-32GB |
| OPT-125m-1-shot | 24 | Intel(R) Xeon(R) Gold 5118 CPU @ 2.30GHz | 4 x Tesla V100-PCIE-32GB |
| OPT-125m-5-shot | 24 | Intel(R) Xeon(R) Gold 5118 CPU @ 2.30GHz | 4 x Tesla V100-PCIE-32GB |
| OPT-125m-10-shot | 24 | Intel(R) Xeon(R) Gold 5118 CPU @ 2.30GHz | 4 x Tesla V100-PCIE-32GB |
| OPT-125m-15-shot | 24 | Intel(R) Xeon(R) Gold 5118 CPU @ 2.30GHz | 4 x Tesla V100-PCIE-32GB |
| OPT-125m-30-shot | 24 | Intel(R) Xeon(R) Gold 5118 CPU @ 2.30GHz | 4 x Tesla V100-PCIE-32GB |
| OPT-6.7b-0-shot | 1 | Intel(R) Xeon(R) Platinum 8275CL CPU @ 3.00GHz | 8 x NVIDIA A100-40GB |
| OPT-6.7b-1-shot | 1 | Intel(R) Xeon(R) Platinum 8275CL CPU @ 3.00GHz | 8 x NVIDIA A100-40GB |
| OPT-6.7b-5-shot | 1 | Intel(R) Xeon(R) Platinum 8275CL CPU @ 3.00GHz | 8 x NVIDIA A100-40GB |
| OPT-6.7b-10-shot | 1 | Intel(R) Xeon(R) Platinum 8275CL CPU @ 3.00GHz | 8 x NVIDIA A100-40GB |
| OPT-6.7b-15-shot | 1 | Intel(R) Xeon(R) Platinum 8275CL CPU @ 3.00GHz | 8 x NVIDIA A100-40GB |
| OPT-6.7b-30-shot | 1 | Intel(R) Xeon(R) Platinum 8275CL CPU @ 3.00GHz | 8 x NVIDIA A100-40GB |
| OPT-2.7b-0-shot | 24 | Intel(R) Xeon(R) Gold 5118 CPU @ 2.30GHz | 4 x Tesla V100-PCIE-32GB |
| OPT-2.7b-1-shot | 24 | Intel(R) Xeon(R) Gold 5118 CPU @ 2.30GHz | 4 x Tesla V100-PCIE-32GB |
| OPT-2.7b-5-shot | 24 | Intel(R) Xeon(R) Gold 5118 CPU @ 2.30GHz | 4 x Tesla V100-PCIE-32GB |
| OPT-2.7b-10-shot | 24 | Intel(R) Xeon(R) Gold 5118 CPU @ 2.30GHz | 4 x Tesla V100-PCIE-32GB |
| OPT-2.7b-15-shot | 24 | Intel(R) Xeon(R) Gold 5118 CPU @ 2.30GHz | 4 x Tesla V100-PCIE-32GB |
| OPT-2.7b-30-shot | 24 | Intel(R) Xeon(R) Gold 5118 CPU @ 2.30GHz | 4 x Tesla V100-PCIE-32GB |
| OPT-30b-0-shot | 48 | Intel(R) Xeon(R) Platinum 8275CL CPU @ 3.00GHz | 8 x NVIDIA A100-40GB |
| OPT-30b-1-shot | 48 | Intel(R) Xeon(R) Platinum 8275CL CPU @ 3.00GHz | 8 x NVIDIA A100-40GB |
| OPT-30b-5-shot | 48 | Intel(R) Xeon(R) Platinum 8275CL CPU @ 3.00GHz | 8 x NVIDIA A100-40GB |
| OPT-30b-10-shot | 48 | Intel(R) Xeon(R) Platinum 8275CL CPU @ 3.00GHz | 8 x NVIDIA A100-40GB |
| OPT-30b-15-shot | 48 | Intel(R) Xeon(R) Platinum 8275CL CPU @ 3.00GHz | 8 x NVIDIA A100-40GB |
| OPT-30b-30-shot | 48 | Intel(R) Xeon(R) Platinum 8275CL CPU @ 3.00GHz | 8 x NVIDIA A100-40GB |
| OPT-1.3b-0-shot | 40 | Intel(R) Xeon(R) Silver 4114 CPU @ 2.20GHz | 4 x Tesla V100-PCIE-16GB |
| OPT-1.3b-1-shot | 40 | Intel(R) Xeon(R) Silver 4114 CPU @ 2.20GHz | 4 x Tesla V100-PCIE-16GB |
| OPT-1.3b-5-shot | 40 | Intel(R) Xeon(R) Silver 4114 CPU @ 2.20GHz | 4 x Tesla V100-PCIE-16GB |
| OPT-1.3b-10-shot | 40 | Intel(R) Xeon(R) Silver 4114 CPU @ 2.20GHz | 4 x Tesla V100-PCIE-16GB |
| OPT-1.3b-15-shot | 40 | Intel(R) Xeon(R) Silver 4114 CPU @ 2.20GHz | 4 x Tesla V100-PCIE-16GB |
| OPT-1.3b-30-shot | 40 | Intel(R) Xeon(R) Silver 4114 CPU @ 2.20GHz | 4 x Tesla V100-PCIE-16GB |
| OPT-175b-0-shot | 96 | Intel(R) Xeon(R) CPU @ 2.20GHz | 16 x NVIDIA A100-40GB |
| OPT-175b-1-shot | 96 | Intel(R) Xeon(R) CPU @ 2.20GHz | 16 x NVIDIA A100-40GB |
| OPT-175b-5-shot | 96 | Intel(R) Xeon(R) CPU @ 2.20GHz | 16 x NVIDIA A100-40GB |
| OPT-175b-10-shot | 96 | Intel(R) Xeon(R) CPU @ 2.20GHz | 16 x NVIDIA A100-40GB |
| OPT-175b-15-shot | 96 | Intel(R) Xeon(R) CPU @ 2.20GHz | 16 x NVIDIA A100-40GB |
| OPT-175b-30-shot | 96 | Intel(R) Xeon(R) CPU @ 2.20GHz | 16 x NVIDIA A100-40GB |
| OPT-66b-0-shot | 48 | Intel(R) Xeon(R) Platinum 8275CL CPU @ 3.00GHz | 8 x NVIDIA A100-40GB |
| OPT-66b-1-shot | 48 | Intel(R) Xeon(R) Platinum 8275CL CPU @ 3.00GHz | 8 x NVIDIA A100-40GB |
| OPT-66b-5-shot | 48 | Intel(R) Xeon(R) Platinum 8275CL CPU @ 3.00GHz | 8 x NVIDIA A100-40GB |
| OPT-66b-10-shot | 48 | Intel(R) Xeon(R) Platinum 8275CL CPU @ 3.00GHz | 8 x NVIDIA A100-40GB |
| OPT-66b-15-shot | 48 | Intel(R) Xeon(R) Platinum 8275CL CPU @ 3.00GHz | 8 x NVIDIA A100-40GB |
| OPT-66b-30-shot | 48 | Intel(R) Xeon(R) Platinum 8275CL CPU @ 3.00GHz | 8 x NVIDIA A100-40GB |

Table 98: Compute used per experiment with non-API models. (4/4)

| model | cpus | cpu model | gpu model |
|-------|------|-----------|-----------|
| BlenderBot-2.7b-0-shot | 10 | Apple M1 Max | |
| BlenderBot-2.7b-1-shot | 10 | Apple M1 Max | |
| BlenderBot-2.7b-5-shot | 10 | Apple M1 Max | |
| BlenderBot-2.7b-10-shot | 10 | Apple M1 Max | |
| BlenderBot-2.7b-15-shot | 10 | Apple M1 Max | |
| BlenderBot-2.7b-30-shot | 10 | Apple M1 Max | |
| BlenderBot-9.4b-0-shot | 96 | Intel(R) Xeon(R) CPU @ 2.20GHz | 16 x NVIDIA A100-40GB |
| BlenderBot-9.4b-1-shot | 96 | Intel(R) Xeon(R) CPU @ 2.20GHz | 16 x NVIDIA A100-40GB |
| BlenderBot-9.4b-5-shot | 96 | Intel(R) Xeon(R) CPU @ 2.20GHz | 16 x NVIDIA A100-40GB |
| BlenderBot-9.4b-10-shot | 96 | Intel(R) Xeon(R) CPU @ 2.20GHz | 16 x NVIDIA A100-40GB |
| BlenderBot-9.4b-15-shot | 96 | Intel(R) Xeon(R) CPU @ 2.20GHz | 16 x NVIDIA A100-40GB |
| BlenderBot-9.4b-30-shot | 96 | Intel(R) Xeon(R) CPU @ 2.20GHz | 16 x NVIDIA A100-40GB |
| BlenderBot-90m-0-shot | 10 | Apple M1 Max | |
| BlenderBot-90m-1-shot | 10 | Apple M1 Max | |
| BlenderBot-90m-5-shot | 10 | Apple M1 Max | |
| BlenderBot-90m-10-shot | 10 | Apple M1 Max | |
| BlenderBot-90m-15-shot | 10 | Apple M1 Max | |
| BlenderBot-90m-30-shot | 10 | Apple M1 Max | |
| T0-3b-0-shot | 96 | Intel(R) Xeon(R) CPU @ 2.20GHz | 16 x NVIDIA A100-40GB |
| T0-3b-1-shot | 96 | Intel(R) Xeon(R) CPU @ 2.20GHz | 16 x NVIDIA A100-40GB |
| T0-3b-5-shot | 96 | Intel(R) Xeon(R) CPU @ 2.20GHz | 16 x NVIDIA A100-40GB |
| T0-3b-10-shot | 96 | Intel(R) Xeon(R) CPU @ 2.20GHz | 16 x NVIDIA A100-40GB |
| T0-3b-15-shot | 96 | Intel(R) Xeon(R) CPU @ 2.20GHz | 16 x NVIDIA A100-40GB |
| T0-3b-30-shot | 96 | Intel(R) Xeon(R) CPU @ 2.20GHz | 16 x NVIDIA A100-40GB |
| T0-11b-0-shot | 96 | Intel(R) Xeon(R) CPU @ 2.20GHz | 16 x NVIDIA A100-40GB |
| T0-11b-1-shot | 96 | Intel(R) Xeon(R) CPU @ 2.20GHz | 16 x NVIDIA A100-40GB |
| T0-11b-5-shot | 96 | Intel(R) Xeon(R) CPU @ 2.20GHz | 16 x NVIDIA A100-40GB |
| T0-11b-10-shot | 96 | Intel(R) Xeon(R) CPU @ 2.20GHz | 16 x NVIDIA A100-40GB |
| T0-11b-15-shot | 96 | Intel(R) Xeon(R) CPU @ 2.20GHz | 16 x NVIDIA A100-40GB |
| T0-11b-30-shot | 96 | Intel(R) Xeon(R) CPU @ 2.20GHz | 16 x NVIDIA A100-40GB |
| Flan-T5-3b-0-shot | 96 | Intel(R) Xeon(R) CPU @ 2.20GHz | 16 x NVIDIA A100-40GB |
| Flan-T5-3b-1-shot | 96 | Intel(R) Xeon(R) CPU @ 2.20GHz | 16 x NVIDIA A100-40GB |
| Flan-T5-3b-5-shot | 96 | Intel(R) Xeon(R) CPU @ 2.20GHz | 16 x NVIDIA A100-40GB |
| Flan-T5-3b-10-shot | 96 | Intel(R) Xeon(R) CPU @ 2.20GHz | 16 x NVIDIA A100-40GB |
| Flan-T5-3b-15-shot | 96 | Intel(R) Xeon(R) CPU @ 2.20GHz | 16 x NVIDIA A100-40GB |
| Flan-T5-3b-30-shot | 96 | Intel(R) Xeon(R) CPU @ 2.20GHz | 16 x NVIDIA A100-40GB |
| Flan-T5-780m-0-shot | 96 | Intel(R) Xeon(R) CPU @ 2.20GHz | 16 x NVIDIA A100-40GB |
| Flan-T5-780m-1-shot | 96 | Intel(R) Xeon(R) CPU @ 2.20GHz | 16 x NVIDIA A100-40GB |
| Flan-T5-780m-5-shot | 96 | Intel(R) Xeon(R) CPU @ 2.20GHz | 16 x NVIDIA A100-40GB |
| Flan-T5-780m-10-shot | 96 | Intel(R) Xeon(R) CPU @ 2.20GHz | 16 x NVIDIA A100-40GB |
| Flan-T5-780m-15-shot | 96 | Intel(R) Xeon(R) CPU @ 2.20GHz | 16 x NVIDIA A100-40GB |
| Flan-T5-780m-30-shot | 96 | Intel(R) Xeon(R) CPU @ 2.20GHz | 16 x NVIDIA A100-40GB |
| Flan-T5-11b-0-shot | 96 | Intel(R) Xeon(R) CPU @ 2.20GHz | 16 x NVIDIA A100-40GB |
| Flan-T5-11b-1-shot | 96 | Intel(R) Xeon(R) CPU @ 2.20GHz | 16 x NVIDIA A100-40GB |
| Flan-T5-11b-5-shot | 96 | Intel(R) Xeon(R) CPU @ 2.20GHz | 16 x NVIDIA A100-40GB |
| Flan-T5-11b-10-shot | 96 | Intel(R) Xeon(R) CPU @ 2.20GHz | 16 x NVIDIA A100-40GB |
| Flan-T5-11b-15-shot | 96 | Intel(R) Xeon(R) CPU @ 2.20GHz | 16 x NVIDIA A100-40GB |
| Flan-T5-11b-30-shot | 96 | Intel(R) Xeon(R) CPU @ 2.20GHz | 16 x NVIDIA A100-40GB |

