# OpenReview forum: "The Goldilocks of Pragmatic Understanding: Fine-Tuning Strategy Matters for Implicature Resolution by LLMs"
_NeurIPS.cc/2023/Conference — NeurIPS 2023 spotlight_

### Official Review · Reviewer_VB9K · 2023-07-04

**Soundness:** 3 good
**Presentation:** 3 good
**Contribution:** 2 fair
**Rating:** 5
**Confidence:** 3

**Summary:**

The paper introduces a task to evaluate the ability of large language models (LLMs) to resolve conversational implicatures and go beyond the literal interpretation of the meaning of the language. The evaluation is based on a dataset of naturally occurring implicatures that are converted into a binary classification task using a set of templates. For instance, the dialogue _"Can you come to my party on Friday?”_ , _"I have to work"_ is converted into two sentences: _Esther asked “Can you come to my party on Friday?” and Juan responded “I have to work” which means no_ and _Esther asked “Can you come to my party on Friday?” and Juan responded “I have to work”, which means yes._. The two sentences are inputed in the different models, considering the one maximizing the likelihood as the model choice.

Four categories of state of the art models are evaluated on the proposed task. The evaluation protocol takes into account potential variance induced by prompt templates. Different methodologies are also benchmarked to try to improve the performance of the models on the task, this includes:
- Fine tuning on the task.
- Instruction tuning-;
- Chain of thought prompting

Results demonstrate that the task is challenging for base models (BERT, RoBERTa, GPT2, GPT3, ...) with a performance slightly above random. Without any fine tuning, some models already reach a performance above 70% (text-davinci-00x, ChatGPT), with GPT4 only 5 points behind human performance. The improvement induced by instruction tuning and chain of thought prompting is further demonstrated for some models with GPT4 reaching human performance.

**Strengths:**

The paper is well written with all the material provided to reproduce the results. The paper introduces an original evaluation protocol for the task of resolving implicatures and demonstrates that human performance can be reached by combining state of the art models (GPT-4) and the recently proposed chain-of-thought prompting.

**Weaknesses:**

The task proposed is not new and the dataset proposed is the same as the one introduced in [BigBench](https://arxiv.org/abs/2206.04615). The results, in the mentioned paper, already demonstrated the ability of PaLM with k-shot-prompting to [perform](https://github.com/google/BIG-bench/tree/main/bigbench/benchmark_tasks/implicatures) above average human performance. The main differences between the two papers include the choice to keep ambiguous cases for the task and adding other models along with a chain of thoughts prompting to the benchmark.
In the benchmark, the variance in the performance of the models, displayed in Table 2, only takes into account the variability induced by different prompt wording. To compare the models, it would have been helpful to perform bootstrap on the test set to estimate to support the claim on the comparison of the performance of the different models.

**Questions:**

Why has no fine tuning on the base models been performed?

Do you have the increase in performance induced by chain of thoughts prompting for GPT4 for particularized (context heavy) examples?

l107 _"Our analysis is novel in its approach; using ambiguous data that humans can easily resolve, and scope"_ Maybe reformulate, not very clear

GPT4 doesn't seem to benefit from in context example. How do you explain that?

Did you try to isolate the effect of data contamination? Especially for GPT-4, the model could have been trained on the data used in your evaluation protocol given that the original dataset was published in 2020

**Limitations:**

The proposed evaluation framework focuses on conversational implicature. If implicature is key for natural language understanding, evaluating models on conventional implicature is also important. If outside of the scope of this well conducted evaluation, it could be of interest for future work.

---

> ### Author Rebuttal · Authors · 2023-08-09
>
> We thank the reviewer for taking the time to thoroughly review, rating the soundness and presentation as good. We are happy the reviewer believes our evaluation protocol is original and all the material is provided to reproduce it. We justify below why _our contribution is especially important in light of the BIG bench result the reviewer mentions_, which we believe is not a sound nor reproducible empirical result.
>
> To summarise our argument, further detailed below;
>
> 1. The method of the implicature task contributors in BIG bench raises serious questions as to the validity of their claims
> 2. The result is based on base LLMs; no models with SotA fine-tuning methods are evaluated
> 3. It is not discussed in a peer-reviewed (or any) paper
>
> Additionally, we discuss the bootstrap estimates of the std error, why data contamination is unlikely, and answers to other questions. We hope that this will lead to strengthening your recommendation or letting us know what still stands in the way, so that we may further improve our paper.
>
> ### Clarifying misunderstandings in the reviewer’s summary of the submission
>
> Before we address the reviewer’s comments in detail, we’d like to clarify what we believe is a misunderstanding in the reviewer's summary. We don't fine-tune models on the task. We instead look at _emergent_ pragmatic understanding from different domain-general SotA LLM training and fine-tuning methods. Further, the reviewer says we perform “_Instruction tuning: adding k examples from the training set in the prompt;”._ We apologise for the confusion, but instruction tuning is a different method from adding k examples to the prompt. The latter is a prompting technique requiring no weight updates to the model, whereas the former is a SotA fine-tuning method. This is mentioned in line 47-49 and line 179-182. We will make sure to update these explanations to prevent future confusion by other readers.
>
> ### The BIG bench result is not an empirically sound or reproducible result
>
> We respectfully disagree with the reviewer's assertion that our work lacks contribution due to the existence of the BIG bench task using the same dataset and presenting similar results, although we can understand that at first glance this might seem to be the case. For a detailed argument, please refer to the [global author rebuttal](https://openreview.net/forum?id=5bWW9Eop7l&noteId=RgN3e2U0c5).
>
> ### Bootstrap on the test set to compare different models
>
> Bootstrap estimates of standard error is actually a feature of the evaluation library that we used (see line 207-233 on GitHub in the file `lm-evaluation-harness/lm_eval/metrics.py`, function `bootstrap_stderr()`), so we do have these numbers. We didn't include them because the variation was always much lower than the variation across prompts, which we felt would be more interesting to most NLP researchers. Once we upload all result files, the bootstrap estimates will also be included.
>
> ### Conventional implicature is out of scope for this work
>
> The reviewer says _“evaluating models on conventional implicature is also important. If outside of the scope of this well conducted evaluation, it could be of interest for future work.”_ We agree that studying whether conventional implicature is interesting, but indeed, it is out of scope of what is being studied here. We look at how models can resolve implicatures that require context to be resolved, whereas conventional implicatures are resolved by the conventional meaning of the word. In fact, linguists sometimes argue conventional implicatures are part of semantics, not pragmatics (see appendix D).
>
> ### Answers to questions
>
> _Why no fine tuning on the base models?_
>
> We believe the reviewer suggest an interesting avenue for future work by this question that is out of scope for our current work (we do not have the computational resources), and hope someone with the available computes picks up this question.
>
> _Increase in performance from CoT for GPT4 for particularized examples?_
>
> On particularised examples GPT-4 achieves 81.6\% accuracy with CoT prompting (from Table 28), and 71.2\% 5-shot (from Table 20), so the improvement is ~10\%.
>
> _Why does GPT4 not benefit from in context examples_
>
> It’s a good question but it’s hard to say because we don’t know details of GPT-4. We think this is due to more extensive instruction tuning allowing it to better respond to instructions zero-shot. This is partly corroborated by our results in appendix I.6 that show that the models mostly benefit from in-context examples because of the formatting.
>
> _Effect of data contamination_
>
> We believe data contamination is not an issue for this benchmark for multiple reasons. When we evaluate GPT-4 with only the question, the performance is close to random (e.g. question-only 0-shot is ~49\% and 5-shot is ~54%). We can add these results to the appendix to clarify this in the paper. Additionally, when we search for the dataset in the large-scale internet-scraped datasets, we do not find any matches. This can be verified with the Gaia search tool on HuggingFace. Finally, the pattern of performance that GPT-4 achieves maps closely onto human performance. If it would’ve memorised the dataset it should be higher than human performance, as our human evals are not public yet.
>
> ### Concluding remarks
>
> We hope this adequately addresses the reviewer’s concerns. We will make the following changes to the manuscript to clarify the arguments laid out above:
>
> 1. adjust the related work to summarise the argument about BIG bench.
> 2. rewrite Appendix H to get across the above arguments in more detail.
>
> We would like to thank the reviewer for raising these issues, as we believe clarifying the presentation issues above will make the paper’s contribution clearer. We hope you consider raising your score, and if there are any outstanding concerns that would inhibit you from doing so, we will gladly address them.

---

> > ### Comment · Reviewer_VB9K · 2023-08-16
> >
> > Thanks a lot for answering all the concerns mentioned. Once again, I think it's a well conducted paper with a comprehensive evaluation providing interesting insights on the ability of LLMs to resolve conversational implicatures.
> > - About instruction tuning, you are completely right and sorry for the confusion. And you nicely demonstrate the benefit on this methodology to resolve implicatures.
> > - About the BIG Bench task, I understand the two limitations you are mentioning: overestimation of the human performance and the other one about the choice made to keep only the subset of unambiguous examples in the BIG Bench benchmark. About the human performance, you performed a thorough evaluation, involving 5 annotators for each subsets of 150 examples with metrics on annotator agreement. This is way more detailed that the BIG BENCH paper. About the choice, made in Big Bench of keeping the subset of *unambiguous* cases, it seems that the annotators, in your paper, are able to perform significantly above chance level. So, indeed, they might have been too conservative when selecting the subset but I don't think it makes the task more interesting. You also mention that the *BIG bench task uses only base LLMs and no state-of-the-art fine-tuning methods*. It's important to note that they reach above human performance relying on it.
> > - About the effect of data contamination, thanks a lot for providing the performance with the question only for GPT-4. With those numbers, it seems indeed that data contamination is unlikely. I think adding this in the Appendix could indeed be helpful. Could you also specify how you evaluated GPT-4 with question only?
> >
> > Finally, I want to thanks the authors for addressing all the comments. The paper is well written with a well conducted benchmark, ensuring reproducibility with all the details provided. My main concerns were about the novelty of the work done and the potential moderate impact given the BIG BENCH paper. To be more specific, the task seemed already to be [solved](https://github.com/google/BIG-bench/tree/main/bigbench/benchmark_tasks/implicatures) with the big bench paper, where they demonstrated that relying on PaLM with k-shot-prompting performed above average human. Given that the Big Bench paper is not peered reviewed and that evaluation is way more detailed in your paper,  I have raised my score.

---

> > > ### Author Response · Authors · 2023-08-17
> > >
> > > We thank the reviewer for thoughtfully responding to our rebuttal, and revising their score. We are curious as to what would hold the reviewer back from stronger support for the paper. We believe it might be down to some outstanding misunderstandings, and would appreciate working with you both to ensure that the writing can be amended where needed such that other readers do not make them, and in the hope that you will consider bringing your level of support for the paper in line with the other reviewers.
> > >
> > > **BIG bench**
> > >
> > > We believe (correct us if we’re wrong) that what is standing between the reviewer and stronger support for the paper is related to a perceived limited impact due to the BIG bench result, as the reviewer also mentions they think the paper is *"well conducted paper with a comprehensive evaluation providing interesting insights on the ability of LLMs"* and say that we are *"ensuring reproducibility with all the details provided"*. Below, we aim to clarify that the difference with the BIG bench is not *just* that our work was peer reviewed and theirs wasn’t, but that—beyond the novel analysis and model comparisons in our paper—**our version of the task is also harder**, with differences in the data distribution and experimental protocol, which we believe makes the evaluation **fairer and more informative**.
> > > We expect PaLM's few-shot score to be lower on our dataset due to the difficult additional examples in our benchmark comprising 30% of the total data. We expect BIG bench human performance to be underestimated due to low-quality human evaluation, and actual human performance to be higher, like on our benchmark. Therefore we cannot be sure that PaLM would reach above human level performance on our benchmark, and with it, solving the task -- in fact it seems quite unlikely.
> > >
> > > Taking all this into account, we would argue that you simply cannot say much about implicature performance of contemporary LLMs based on the BIG bench result at all, and in turn our paper is **the first to actually provide this contribution**.
> > >
> > > **Spurious correlation experiment**.
> > > We re-ran the benchmark fully but removing the responses from the examples. We did this 0-shot and 5-shot, where we did the latter to give 5 examples of how to potentially utilise spurious correlations in the benchmark. The results are as follows:
> > >
> > > ```
> > > | Question-only | 0-shot         | 5-shot         |
> > > |---------------|----------------|----------------|
> > > | ChatGPT       | 54.3% +/- 3.3  | 41.7% +/- 12.4 |
> > > | GPT-4         | 48.9% +/- 10.5 | 53.7% +/- 0.5  |
> > > ```
> > >
> > > The result for chatGPT and GPT-4 are mostly random, meaning that it's unlikely these models will utilise spurious correlations in our benchmark in a few-shot setup.
> > >
> > > **********************Conclusions**********************
> > >
> > > We hope that clarifies the fact that the contribution of this paper is indeed ******************************************substantive and novel******************************************. To further clarify the writing regarding this, we will revise the related work section to get the above across more clearly. Specifically, we will focus more on the claim that we still don’t know LLM’s performance compared to humans from the BIG bench result because it likely overestimates LLM implicature understanding and underestimates human performance. We would be thankful if you had any further suggestions regarding where we could make this distinction clearer, and hope you will be in a position to strengthen your support for the paper.

---

> > > > ### Comment · Reviewer_VB9K · 2023-08-17
> > > >
> > > > Thanks for the answer to my questions. As explained in my first comment, I acknowledge the quality of your work but I am still questioning the novelty. As apparently I made *some outstanding misunderstandings*, I am going to try to rephrase why I think the paper is interesting but with limited novelty.
> > > > - One of the main arguments you state is that your version of the task is harder. More specifically, you kept *the difficult additional examples in our benchmark comprising 30% of the total data*. I want to emphasize that some of the filtering performed for the BIG BENCH task seems legitimate. For instance, they provide an example of factual mistake in the original dataset:
> > > >
> > > > *Speaker 1: Are you busy?*
> > > >
> > > > *Speaker 2: I’m drowning in work.*
> > > >
> > > > *Original answer: No.*
> > > >
> > > > To me, removing this example and others that are too ambiguous to reach human agreement mostly changes the expected human performance but doesn't make the task more fair or informative.
> > > > - One important strength of your evaluation protocol compared to the BIG BENCH Benchmark is the addition of the prompt sensitivity analysis over the prompt templates. One again this demonstrates the soundness of your approach and I want to highlight once again the quality of your work. However, based on the results, we can observe that the variability induced by the prompt template is quite low. Therefore, I disagree with your statement: *"you simply cannot say much about implicature performance of contemporary LLMs based on the BIG bench result"*.
> > > >
> > > > I want to reiterate that the paper is very well written, reproducible with experiments that are well conducted. Your work is clearly more detailed than the one from BIG BENCH. However I disagree with the statement that you are the first to assess implicature performance of LLMs. As I understand it, it mostly come from our disagreement on the importance of including what you call the *the difficult additional examples*. I hope it clarifies my view on the novelty of the paper and happy to discuss if you need further clarification.

---

> > > > > ### Author Response · Authors · 2023-08-18
> > > > >
> > > > > We thank the reviewer for the swift response and engaging so actively with our rebuttals. We genuinely appreciate this. We do however still (respectfully!) disagree with the reviewer’s comments about the BIG bench result, and elaborate further below.
> > > > >
> > > > > **BIG bench**
> > > > >
> > > > > While it is true that our protocol keeps all the data of the original dataset collectors, and therefore has some annotation errors, the fact remains that on this entire subset of 30% of the data humans achieve 72%. Even if there are some annotation errors, humans still get significantly above random accuracy on the examples. So before we can say PaLM achieves human accuracy, we should verify the accuracy it gets on these more challenging examples. Two examples that BIG bench also discards:
> > > > >
> > > > > Utterance: *"Can you lend me hundred dollars?"*.
> > > > > Response: *"Is this supposed to be some kind of a joke?"*.
> > > > > Implicature: *"No"*.
> > > > >
> > > > > Utterance: *"Do you know, how long is Uncle Arthur staying with us?"*.
> > > > > Response: *"Ask your father."*.
> > > > > Implicature: *"No"*
> > > > >
> > > > > These are valid implicatures that humans get correct. They are indeed ambiguous, but that’s the point of implicatures. As the reviewer says, the annotation errors don’t make the task more fair or informative, but the ambiguous examples do.
> > > > >
> > > > > **Prompt sensitivity**
> > > > >
> > > > > The reviewer also mentions a strength of our protocol is the estimation of prompt sensitivity and that based on our result we can say variability is low. Now, we might guess that for PaLM prompt sensitivity also won’t be an issue (although we do not know this for sure), but without our result we wouldn’t know this.
> > > > >
> > > > > To summarise; the reason we say we can’t say much about LLM implicature performance from the BIG bench result is because it overestimates LLM performance by also discarding valid examples, likely underestimates human performance, doesn’t estimate prompt sensitivity, doesn’t evaluate SotA fine-tuning techniques, and is not reproducible or analysed in any paper.

---

> > > > > > ### Author Response · Authors · 2023-08-21
> > > > > >
> > > > > > Dear reviewer,
> > > > > >
> > > > > > Thank you again for your responses and the engagement with our rebuttal. In light of the nearing end of the rebuttal period, we’d like to kindly remind you of the above response to your clarification on the novelty of our work.

---

### Official Review · Reviewer_FPTW · 2023-07-06

**Soundness:** 3 good
**Presentation:** 3 good
**Contribution:** 2 fair
**Rating:** 7
**Confidence:** 4

**Summary:**

This paper evaluates several popular LLMs on a benchmark for evaluating pragmatic reasoning via resolving question-answer conversation snippets into the implied binary answer. The study explores the influence of different factors in LLM design, including training method (next-token prediction vs. instruction tuning), number of few-shot examples, model size, inference method, etc. Several conclusions are made about the results: instruction tuning seems to improve pragmatic reasoning, models are mostly robust to perturbations in the input prompts, scaling model size seems to have an effect in improving performance, CoT prompting is useful for some models, and remaining failures of the model appear to center on more difficult (particularised) implicatures.

**Strengths:**

* The paper offers a very comprehensive / thorough / careful evaluation of the LLMs being evaluated. Particularly, the try a number of different prompt variations, try different numbers of few-shot examples (as well as having a randomization process for choosing these examples, to avoid confounding results with some fixed set of examples)
* The finding that few-shot examples are useful mostly for conveying format of the response, rather than as a way of defining the task itself, is very interesting. I would be interested to see a more in-depth study of this (perhaps another paper) as it is somewhat surprising (but perhaps less surprising for instruction-tuned models).

**Weaknesses:**

In general, I wanted to see a bit more discussion and analysis, since this paper is focused on analysis.
* Moving the analysis for other types of implicatures into the main paper
* More examples of what the model is generating, particularly for CoT examples
* More discussion / analysis of human "errors" here -- why might there be errors? Genuine noise in the annotation process, or different interpretations of the same example?
* Discussion on why instruction tuning might be influential in improving pragmatic reasoning
* Discussion / analysis on why there is still a gap for particularised implicatures, and what we can do to address it

Nits:
* The first example (GPT's answer to the user's question about the phone) actually doesn't necessarily make sense, as it's definitely not literally true -- GPT can't have seen the user's phone
* "fine-tuning on instructions at the example-level" is vague to me, particularly the "example level" part. What is this supposed to refer to? Is there a distinction between general "instruction-tuning" and "instruction-tuning ad the example level"?
* Also, "context" in Insight 5 is vague -- I wouldn't think of this kind of commonsense/world knowledge as "context" necessarily, so I'd suggest renaming this
* Formatting of Figure 2 is somewhat odd
* I'd suggest adding a line for "human performance" in Figure 2
* I'd suggest adding the few-shot CoT example (or some examples of generated CoT answers) into the main paper

**Questions:**

* Is there evaluation on this dataset with respect to an upper bound in performance given only the response, without the question? I'm curious whether answers like "I've gotta get up early" or "Some" have some spurious correlations with the labels, e.g., "no", in this dataset.
* Is the accuracy actually decreasing relative to k in Figure 4? I.e. is this difference significant? If so, why might that be?

Minor questions:
* Are there cases where examples are ambiguous, and situational context might affect the label? Did you investigate cases where the human annotators were "wrong" to identify the source of those "errors"?

**Limitations:**

Yes, mostly. More discussion on these non-binary implicatures (at the end of the discussion) would be interesting. I'd also be interested if the authors can perform analysis on the existing benchmark to see whether the labels can be predicted with only one half of the example (i.e., the response); i.e. whether there are spurious correlations within the benchmark.

---

> ### Author Rebuttal · Authors · 2023-08-09
>
> We thank the reviewer for the supportive review, saying the paper offers a _“very comprehensive evaluation”_ and that _“the finding that few-shot examples are useful mostly for conveying format [..] is very interesting”_. Below, we address questions. To summarise, we:
> - present new results showing that spurious correlations are likely not an issue
> - show the source of human errors
> - present CoT generations and how those address the gap for particularised examples
> - speculate on why instruction tuning is important
>
> We are confident the reviewer’s suggestions will contribute to a better manuscript and hope that our responses below will lead to strengthening your recommendation or letting us know what still stands in the way, so that we may further improve the submission.
>
> ### Potential spurious correlations
>
> This is an interesting suggestion, so we ran versions of the benchmark with only the question or response. Getting the implicature right from the response only does not always indicate spurious correlations, as some examples only need the response (e.g. rhetorical questions like ‘do pigs fly?’). Question-only results do always indicate spurious correlations.
>
> |Response-only|0-shot|5-shot|
> |---|---|---|
> |ChatGPT|59.2% +- 4.7|58.3% +- 6.6|
> |GPT-4|62.6% +- 1.7|65.5% +- 1.1|
>
> |Question-only|0-shot|5-shot|
> |---|---|---|
> |ChatGPT|54.3% +- 3.3|41.7% +- 12.4|
> |GPT-4|48.9% +- 10.5|53.7% +- 0.5|
>
> Models mostly perform random for question-only, so spurious correlations don't seem to be an issue. For response-only, GPT-4 5-shot gets 65%. Some examples it gets right are: “do fish swim?” and "let's hope so".
>
> ### CoT generations + gap on particularised examples
>
> For this analysis, we propose to add an appendix section that looks at the examples that GPT-4 got right with CoT and wrong with 5-shot. This answers the question about what can fill the gap on particularised examples; CoT closes it for GPT-4 (see Table 28). Explicit reasoning helps, and an example can clarify how (which we will add to the main text). We find models usually get the following wrong:
>
> > A: Is there a bus I can get to the station?
> > B: You can’t rely on it
> > Implicature: yes
>
> GPT-4 5-shot gets this wrong for all 6 templates. With CoT it gets it right for 5 of 6 templates.
>
> The CoT GPT-4 generates is useful:
>
> > Alice says 'You can't rely on it.' Alice must be implying that there is a bus, but it may not be dependable or timely. This means the response to Bob’s question is yes, but with a caution about reliability. Answer: yes
>
> ### Source of human error
>
> We agree this can be added to the main text. We will add that the score we expect a model to achieve is not 100\%, but human best. Some examples can be interpreted in different ways, so there is no true right answer, only the answer most people give. We find that part of the errors in our human eval are different interpretations of the same example, and a few annotation errors. One example that people disagree on is:
>
> A: “Was that easy to negotiate?” B: “That is as easy as shooting fish in a barrel.” Implicature: yes
>
> It depends on whether you think it’s easy to shoot fish in a barrel. We propose to look into the examples that all humans get wrong (likely annotation errors) and examples that humans disagree on (likely multiple interpretations), and summarise the results in the main paper.
>
> ### Why instruction tuning (IT) at the example-level is influential
>
> To answer the reviewer’s question on why we distinguish between benchmark-level and example-level IT; the former is where annotators write a single instruction for an entire dataset. The models are then fine-tuned on each example from the dataset with the same instruction. By contrast, in example-level IT each example in a dataset gets a new instruction, resulting in a more diverse dataset. We will clarify the distinction further in the paper.
> We think example-level IT is important for pragmatics because it provides examples of pragmatic inferences (if the instruction is ambiguous, the annotators are asked to infer the intent; see sec. 3.6 in [1]). It also provides diversity; each example is a new task with a tailored instruction (see Appendix A.2.1 on p. 26 in [1]). In the discussion we suggest future work should look into the effect of data diversity on pragmatic inference (line 297-298); we will work this out in more detail.
>
> [1] Ouyang et al., 2022
>
> ### Analysis for other types of implicatures into the main paper
>
> We appreciate this suggestion, and also find the other types interesting, but the patterns we find are not significant (overlapping confidence intervals), so it’s difficult to say anything about this in the main paper.
>
> ### Discussion on non-binary implicatures
>
> Could the reviewer clarify what they would like discussed? Much discussion is out of scope, because we focus on binary implicatures, but we could for example discuss methods for evaluating non-binary implicatures.
>
> ### Other questions
> *decreasing accuracy w.r.t to k in Fig 4?”*
> This decrease is not significant w.r.t. k=0. Reasons for no increase w.r.t k could be that the examples only clarify structure, which can be clear at k=1, and more examples might cause high variance.
>
> On using the word “context”; we respectfully disagree that we should rename this. It's meant to encompass things like commonsense and world knowledge. We'll clarify in the paper.
>
> We will clarify what human performance is in Fig 2.
>
> ### Concluding remarks
>
> We thank the reviewer for all suggestions. We propose to add to the paper:
>
> - the spurious correlations experiment
> - analysis on the source of human errors
> - a CoT completion with a discussion on how this fills the gap for particularised examples
> - why example IT might be important
>
> We believe the paper will be stronger after adding these, and thank the reviewer for suggestions. We hope you feel in a position to support the publication more strongly as a result. If not, we will gladly attempt to address additional concerns.

---

> > ### Comment · Reviewer_FPTW · 2023-08-17
> >
> > Thank you for the reply! In terms of discussion about non-binary implicatures, apologies for a vague suggestion. I do think discussing evaluation would be interesting (both metrics, and how to collect this kind of data), though don't want to draw from the focus of the paper. Thanks also for the additional analysis and examples. Will update my score as this reply has addressed many of my concerns!

---

> > > ### Author Response · Authors · 2023-08-18
> > > **Thanks for the response!**
> > >
> > > We thank the reviewer for the response and the increase in rating. We will allocate some of the additional page in a final draft of this work to discussing non-binary implicatures in more depth. We are confident that with your thoughtful review we have been able to improve the manuscript; thanks for your engagement!

---

### Official Review · Reviewer_LHbR · 2023-07-14

**Soundness:** 4 excellent
**Presentation:** 4 excellent
**Contribution:** 3 good
**Rating:** 8
**Confidence:** 4

**Summary:**

The authors analyze the behavior of LLM pragmatic understanding (capability to add and omit imformation to efficiently communicate in context) using a new evaluation protocol. They compare LLM performance to human performance and argue that instruction fine-tuning with examples may improve pragmatic understanding.

They propose using the assignment of higher likelihood to coherent utterances than similar but incoherent utterances as "resolving an implicature correctly." This includes both describing a situation where implicature understanding is necessary to essentially resolve an entailment question, whether an answer that on its face is irrelevant means "yes" or "no," or whether its a sufficient answer. For example, `Esther asked “Can you come to my party on Friday?” and Juan responded “I have to work”, which means yes.` is a failure case because the answer is wrong; the response means no. A model preffering `...which means yes.` resolves the implicature.

They assess this problem is few and zero shot settings for LLMs. They control for prompt variation effects with 6 different prompt settings, testing a fairly comprehensive set of both generative proprietary LLMs and standard  open pretrained transformers like RoBERTa. They find no model can beat human performance on their resolution task, but GPT-4 comes closest.

They conclude with a detailed analysis, with 5 insights.

**Strengths:**

Well-defined and scoped problem. Out-of-community definitions (eg, explaining pragmatics) well-motivated. Citations are sufficiently comprehensive in my opinion.

Exciting direction to explore to further understnading and engineering of LLMs.

Fully-specified approach and sufficiently broad evaluation.

Clear statement of insights.

Overal, I think this paper has clear value to the community.

**Weaknesses:**

I think it would be possible to use more than 6 "curated prompt templates" (163-172). The "we control for prompt variability" is really not central to the story of the paper but I don't really find the idea that averaging in essence over 6 prompts is sufficient to argue that that has happened. Perhaps claim of robustness should be weakened.

~They claim task-specific instruction tuning improves implicature resolution, but there's no concrete A/B test for this. They don't instruction fine-tune a model themselves for the task (not that doing so would necessarily be possible given constraints). How do we know that IFT is really driving GPT4's superior performance and not just scale or some other factor? I find the evidence presented here unconvincing~

EDIT: In light of the author's response, I do think the case for IFT driving the improvement to be a bit more convincingly made, or at least as convincingly made as can be given the inherent constraints (OpenAI's closedness for ex). I hope to see the authors response to this point integrated in the camera ready, with maybe some weakening of claims in light of the incomplete degree of knowability of this issue

**Questions:**

Maybe respond to the questions I raised implicitly in the weaknesses? I am open to being convinced.

**Limitations:**

I find limitations to be sufficiently addressed this time.

---

> ### Author Rebuttal · Authors · 2023-08-08
>
> We thank the reviewer for taking the time, and for the supportive and positive review. We are happy to see the reviewer believes the soundness and presentation are excellent and the contribution is good, stating the paper *“has clear value to the community”*. Below, we address the questions raised. To summarise;
>
> - we argue that six templates to control for sensitivity to prompt wording is significantly more than the standard in LLM literature (namely 1) and that more can become prohibitively expensive.
> - we highlight the results in the submission that show that instruction-tuning at the example-level is important for pragmatic understanding; most notably the sharp increase in performance between Cohere-base-52B and Cohere-command-52B, which is only due to instruction tuning at the example level. However, we agree that verifying this in a controlled study is an interesting future work direction (albeit outside our computational budget).
>
> We hope the reviewer will find the answers satisfactory, and will seek to make clarifying tweaks to the writing to anticipate such questions arising for future readers of the paper.
>
> ### Only six prompt templates
>
> We appreciate this suggestion, as we agree with the reviewer that in essence we still cannot be sure there is not some prompt that the model will fail on (or perform better on), and that this is an important shortcoming of LLM evaluations more generally. With the below response we’d like to show that we use many more prompt templates than is the standard in LLM literature (namely 6 instead of only 1 template), and hence believe the claims of likely robustness hold up. Additionally, using more templates quickly becomes prohibitively expensive, which is why most papers likely don’t use more than one or two.
>
> We agree with the reviewer that six prompt templates will not cover the full scope of potential wordings of the problem, we still believe it shows robustness to prompt wording and our claims are not overstated. This is mainly because:
>
> 1. We find in general a low variance w.r.t. prompt wording for models that perform well, showing that variability to wording is most likely not going to be an issue.
> 2. The standard in LLM evaluation papers is still to only report results on a single prompt template (e.g. see; many of the tasks in BIG-bench, or the highly cited zero-shot reasoning paper (Kojima et al., 2022) which uses 2 templates differing only slightly in wording, or the recent LLM MPT-7b presented by MosaicML in a blogpost, or Hu et al. 2023 Appendix A (”a fine-grained comparison …”) who use one prompt template per task, etc.).
> 3. This common pattern of using relatively few prompt templates is likely because more is prohibitively expensive (we times all our evals by 6 for each model, so instead of 600 evaluations per model if we had 1 template per example, we have 3600 evaluations per model. Taking into account that we have evaluated 17 model classes each with multiple model sizes totalling 49 different models, adding more templates runs into high costs pretty quickly). A pragmatic balance between exploring the effect of prompt templates and affording to run experiments on a budget needed to be struck.
>
> Furthermore, because our 600 examples in the dataset are naturally occurring, they already span a broad coverage of different topics and writing styles, showing that a model being able to achieve human-level performance on them is likely to be able to generalise to different examples. A small selection of 3 examples from the test set:
>
> *A: You were a smoker? B: Two packs a day.*
>
> *A: Do you think we were right? B: I think you’d lose.*
>
> *A: Do you have any ketchup left? B: We are swimming in it.*
>
> We hope this adequately addresses the concern of the reviewer regarding claims of robustness, and if not we’d be happy to consider caveating claims of robustness in the paper if the reviewer can elaborate on this still being necessary in light of our arguments above.
>
> ### Claiming task-specific instruction-tuning improves implicature resolution
>
> Thanks for bringing up a valid point of discussion here. We agree with the reviewer that A/B testing is important, but as the reviewer notes as well, this is out of scope given our computational constraints. We believe our results do sufficiently convincingly show that instruction-tuning at the example level is important. For example, Figure 3 (right) shows that base models at similar scales as IFT models perform significantly worse. We see that Cohere-command 52B significantly outperforms Cohere-base 52B, and the only difference between those models is instruction-tuning at the example level (Cohere-command is fine-tuned from Cohere-base). In fact, Cohere-command 52B outperforms other base models more than 3 times the size by a large margin (e.g. GPT-3 175B, BLOOM-176B, OPT-175B).
>
> Then the question remains, how do we know example-level IT is the driving factor for GPT-4 and the other OpenAI models? The answer is that we can’t be sure because of OpenAI’s policy of secrecy, but we can be pretty confident: We evaluated 10 models across 6 model classes and two APIs in the group example-level instruction tuned (of which GPT-4 is one). Within this group, models probably vary significantly in other training and architecture details (especially Cohere-command models versus OpenAI models). This means the most significant difference the Example IT models have with other model groups is instruction-tuning at the example level, making it likely that this factor is the driving factor in their performance. Nonetheless, we agree with the reviewer; an important future work direction would be to verify our findings by a controlled study looking into the effect of instruction fine-tuning on implicature resolution.
>
> We thank the reviewer again for taking the time, and hope the questions raised in the weaknesses section have been adequately addressed. If not, please let us know, and we will gladly look into addressing them.

---

> > ### Comment · Reviewer_LHbR · 2023-08-20
> >
> > Thank you for your thoughtful response. On the IFT claim, I am convinced, and have updated my confidence. I would encourage the authors to add this to the discussion section in the camera ready if it isn't already there! :)

---

### Official Review · Reviewer_SeJL · 2023-08-01

**Soundness:** 4 excellent
**Presentation:** 3 good
**Contribution:** 3 good
**Rating:** 7
**Confidence:** 4

**Summary:**

This paper investigates how well recent LLMs preform on resolving conversational implicatures. It presents a task on conversational implicature resolution built on top of the crowdsourced and human-annotated dataset of George and Mamidi (2020). The work presents an evaluation protocol that lays out how LLMs are evaluated on implicature resolution. The work further presents an array of experiments on various LLMs to understand whether they do well on this task and in what settings (e.g. 0-shot vs 5-shot). Results are analyzed in the form of insights and main takeaways that the authors highlight.

**Strengths:**

-	Addresses an important problem, namely studying the competence of LLMs at pragmatic inferences—here specifically implicatures
-	Extensive experiments on a wide range of recent LLMs to validate how well these LLMs perform pragmatic inferences involving implicatures


**Weaknesses:**

-	The work is poorly situated with respect to prior work on computational modeling of pragmatics and the related work section misses an important line of work which I discuss in the comments.
-	Further analysis is required in certain cases (see comments below)

**Questions:**

Intro:

Line 28: “utterances that convey something other than their literal meaning”
At the first pass of this sentence and reading “other”, I read it as suggesting implicature conveys something else different/unrelated to the literal meaning, and I had to reread it to fit it within the definition of implicature. I suggest replacing “other” with “beyond” as I feel that fits better the description of implicature.


Section 2:

As a work that is, in essence, motivated by investigating how good LLMs are at “interpreting language in context—incorporating its pragmatics”, I feel the related work section leaves out an important line of work on the computational modeling of pragmatic phenomena including implicature and the closely related phenomenon of presupposition and on understanding the limitations of modern NLP models in capturing these phenomena starting with the work of Cianflone et al 2018 on modeling adverbial presupposition, Schuster et al 2020 & Li et al 2021 on scalar implicature, the ImpPress dataset (Jeretic et al 2020) on presupposition and implicature in the context of NLI and related experiments on how good NLI models (including BERT-based models) are at capturing these phenomena, Kim et al 2021 on presupposition verifiability in the context of QA, Parrish et al 2021 with the NOPE presupposition corpus, Kabbara et al 2022 on the limitations of NLI models in resolving presupposition cases. These first come to mind. Despite these efforts focusing on (what is now) smaller LMs models (with the exception of the first mentioned work which precedes the BERT era), I believe they are important efforts that set the stage for the present work and need to be cited and imo are more relevant to the work than some of the information mentioned in the first paragraph of Section 2 (e.g., how some models are toxic, unhelpful, etc). Appendix D has a nice background on implicature from a linguistics/philosophy of language but still misses all of the work I mentioned here on modern computational modeling of pragmatic phenomena.

Section 3:

A general concern of mine is the benchmark contribution of the paper given that the same dataset has been introduced in BIG-bench. The authors are clear about how their dataset is different given that it encompasses instances that are discarded in BIG-bench (for being too “ambiguous”) but regardless the overlap between the two is obviously substantial and so I felt this is something that could benefit from a discussion here.

Section 4:

Regarding the random label experiment and looking at the related section in the appendix (I.6), two points/questions come to mind:
1)	Given that there are two labels in question, did you verify how often the label was not changed? That could eclipse the results we’re seeing here.
2)	This experiment and the close results in some cases (for 5-shot text-davinci-001 and cohere-command-52b) brings up the following question: to what extent the models are learning any notion of pragmatics here? Did you carry out some qualitative analysis in a systematic way to see whether the models are in some cases getting it right for the wrong reasons? For example, Kabbara el al 2022 showed that for presupposition-based NLI, RoBERTa and BERT were getting high accuracy performance on certain presupposition types but upon further investigation and adversarial testing, it turns out the models are often exploiting superficial cues that are not related to notions of pragmatics. I wonder to what extent this could be applicable for the task you’re exploring here.

Regarding the analysis in Appendix I.4 (varying k in-context examples), apart from the conclusion that is in the body of the paper (Section 4, Insight 2), I somewhat feel the results are so noisy for Cohere-52B and OPT-175B that it’s hard to really make sense of the results. InstructGPT3-175B is the only case where the results follow our intuition: the more examples a model sees, the better it gets at understanding the notion of implicature and so the higher the accuracy. For Cohere, we see a very noisy trajectory across the board between k = 1 and 15 (before the results start stabilizing and moving upwards) and for OPT, the results are mostly on a downward trajectory except for a chunk in the middle and end up still going downward for k>15. Would like to see if you have any thoughts on this.

Table 3: Regarding the 5-shot CoT performance

We see a variance between the text-davinci models. Generally, although specific details are not public, we can assume models got better as newer versions were released. So it makes sense to see the performance improve for the 5-shot and 5-shot COT scenarios across the 3 davinci-models. However, it’s still unclear why CoT would help marginally in one case (002) and in a rather noticeable way (003) but hurt the performance in the 001 case. Any insight there? This is even more pronounced when the drop is not trivial (7%) and is the only case among 6 (if we were to count the -0.1% as roughly not helping but not hurting)

Appendix G: Human evaluation

I’m somewhat confused by the choice of the authors to run human experiments using on prompt only. They could’ve simply chosen to run multiple experiments with at least a couple of different prompts. The authors justify this by the fact that humans are less likely (than models) to be sensitive to variations in the prompt. I feel this is at best speculative and, if anything, human experiments and pilot studies often show that human subjects can be primed to think or react in a certain way given certain wording or structure, etc.

Minor presentation note:

Line 25: I’m not sure if there’s a writing style guide that allows this form “I have to work.”. [essentially the two dots back-to-back]. Seems rather odd to me. I would think the one within the quotations need to be dropped. See here for more details:
https://www.hamilton.edu/academics/centers/writing/style/essentials/punctuation-of-quotations#:~:text=The%20final%20period%20or%20comma,the%20period%20follows%20the%20citation.


**Limitations:**

Yes

---

> ### Author Rebuttal · Authors · 2023-08-09
>
> We thank the reviewer for taking the time to review and for the in-depth suggestions. We are glad to read that the reviewer thinks our work _“addresses an important problem”_ and contains _“extensive experiments on a wide range of recent LLMs”_, rating the soundness, presentation, and contribution as good. Below we address the reviewers comments. To summarise;
>
> - we discuss the line of work on computational modeling of pragmatics, which we will incorporate in our manuscript
> - we motivate our contribution further by addressing that the BIG bench result cannot be built upon because their method raises serious questions and is not discussed in any paper.
> - we discuss that models are likely doing pragmatic inferences, but not learning to do so in-context, and present results motivating that models are not relying on spurious correlations.
>
> We hope that this will lead to strengthening your recommendation or letting us know what still stands in the way, so that we may further improve the submission.
>
> ### Situating our work
> Thank you for pointing out these works, we are happy to incorporate them. Specifically, we will expand the related work by discussing those that look into emergence of pragmatic understanding from language modeling (Jeretic et al (2020) and Parrish et al (2021)). We will cite the papers that look into explicit computational modeling for pragmatic understanding in the the related work, saying they laid the groundwork for the idea that pragmatics is difficult for computational models, and expand upon each in the background section (Cianflone et al. (2018), Schuster et al. (2020), Kim et al., (2021)). We couldn't find Li et al. (2021), could you share the title?
>
> ### Contribution in relation to the BIG-bench
> Thanks for raising this point of discussion, which we address in the [global response above](https://openreview.net/forum?id=5bWW9Eop7l&noteId=RgN3e2U0c5), as reviewer VB9K mentioned it as well.
>
> ### Random labels
> We aren't sure we understand the question about the random label experiment (_”did you verify how often the label was not changed?”_). We take it to mean; how often did the random labelling assign the correct label to the in-context examples? In the 5-shot case the label is wrong 1443 times and right 1557 times. For 1-shot, the label is wrong 265 times and right 335 times. We hope this clarifies and if not, please let us know.
>
> ### Are models learning pragmatics?
> We believe they are, though not from the in-context examples. It's an interesting question what other spurious correlations the model might rely on. Kabbara et al. (2022) point out cues that arise because the data is synthetic, like the presence of certain tokens (“exactly”) in many of the examples or similarity between premise and hypothesis (”Rene might have hidden” and “Rene hid”). Similar lexical cues will not be a part of our dataset, as they are naturally occurring and have little/no similarity between question and response. This also means the adversarial testing method Kabbara et al. use is less applicable.
> Instead, we ran an analysis to determine potential spurious correlation between question and label. If the models know what the implicature is without even looking at the response, that indicates spurious correlations. The result is as follows:
>
> |Question-only|0-shot|5-shot|
> |---|---|---|
> |ChatGPT|54.3% +/- 3.3|41.7% +/- 12.4|
> |GPT-4|48.9% +/- 10.5|53.7% +/- 0.5|
>
> From this we can conclude there are probably no spurious correlations in the questions that can be used without fine-tuning, and hence it’s unlikely the models in our study uses them.
>
> ### Noise w.r.t. increasing k (I.4)
> We agree with the reviewer that the results for the separate prompt templates w.r.t. k barely show a pattern, apart from what's mentioned in the paper. We know from the random labels experiment that the models that can do pragmatic inferences only use the in-context examples for the format. We think the reason for this noise is that giving the format of a problem doesn’t help if the failure wasn’t formatting but understanding in the first place. In such case, a model might give entirely different answers when given different in-context examples.
>
> ### 5-shot CoT performance
> We have asked ourselves this question as well. What we came up with is the following (albeit speculative):
>
> - CoT doesn’t help/hurts for Dav-1 and Dav-2 and Cohere-command-52B
> - CoT does help for Dav-3, ChatGPT, and GPT-4
>
> What distinguishes the first and the second category is RLHF. Only ChatGPT, Dav-3, and GPT-4 have undergone RLHF fine-tuning. Perhaps this enables CoT reasoning for this task.
>
> ### Human evaluation
> This is a fair point. Indeed, assuming humans are not sensitive to prompt template is speculative, but we opted for this to make the full human study _directly_ comparable to the model's results on template 2. If we had done a mix of all templates we either had to spent 6x as much on the human evals (which was not within our budget) or subsample evals, making it less comparable to part of the model study. We hope the reviewer also sees value in this design choice over the alternative. We will clarify in the paper that there is a speculative aspect to this choice.
>
> Thanks for the notes on line 25, and 28; we will update both!
>
> We would like to thank the reviewer for the thoughtful questions and we are confident that we can improve the manuscript as a result of this discussion. Specifically; we will update the related work and background. We’ll add the spurious correlation analysis in the appendix. We’ll clarify that the design choice of using only one prompt template for the human eval is speculative. Finally, we’ll update the related work and summarise why this work is a contribution in light of the BIG bench results. We hope this adequately addresses the mentioned weaknesses. If there’s discussion points left that would prevent the reviewer from raising their score, please let us know and we will further address them.

---

> > ### Comment · Reviewer_SeJL · 2023-08-17
> > **Answer to rebuttal**
> >
> > Thank you for the detailed response to my review!
> >
> > 1) Regarding the references that I mentioned, this is the complete list of papers (according to the order of appearance in my review):
> > - Cianflone et al (ACL 2018): Let’s do it “again”: A First Computational Approach to Detecting Adverbial Presupposition Triggers
> > - Schuster et al (ACL 2020): Harnessing the linguistic signal to predict scalar inferences
> > - Li et al (SCiL 2021): Predicting scalar inferences from “or” to “not both” using neural sentence encoders
> > - Jeretic et al (ACL 2020): Are Natural Language Inference Models IMPPRESsive? Learning IMPlicature and PRESupposition
> > - Kim et al (ACL 2021): Which Linguist Invented the Lightbulb? Presupposition Verification for Question-Answering
> > - Parrish et al (CoNLL 2021): NOPE: A Corpus of Naturally-Occurring Presuppositions in English
> > - Kabbara et al (COLING 2022): Investigating the Performance of Transformer-Based NLI Models on Presuppositional Inferences
> >
> > I believe these should be referenced in the main body of the paper as they all present various research efforts on either modeling the pragmatic phenomena of implicature and/or presupposition or understanding the limitations of LLMs in capturing these phenomena. And so, mentioning them properly situates the work. I would leave it to you to best decide to what level of detail to cover them in the Related Work section and referencing them in some way (e.g. grouping several by some topics) VS expanding on which in the appendix.
> >
> > 2) Are the models learning pragmatics?
> >
> > Thanks for presenting these results. The answer is convincing to me. I would think highlighting this in your paper (possibly in the body of the paper by just giving the insight and then further expanding on this in the appendix) will benefit the paper because this specific point (is the model actually learning pragmatics?) is often a point that is raised in this type of papers that attempt to highlight a competence of LLMs especially when it comes to whether they exhibit some form of “understanding”. Specifically, many in the community these days would reject that LLMs can exhibit any kind of understanding, let alone a pragmatic understanding. So such a discussion will strengthen the position of the paper.
> >
> > I replied above to points that either asked for a clarification or that I wanted to further comment on. The rest of the answers are satisfactory. I will raise my score as the rebuttal addressed most of my concerns (raised both overall score and the soundness score).

---

> > > ### Author Response · Authors · 2023-08-18
> > > **Thanks for the response!**
> > >
> > > Thanks a lot for the additional response and for listing these related works in more detail. We agree they do set the stage for our work, and merit discussion. We also agree with the reviewer that the discussion highlighting why we believe models are likely actually learning something interesting here would be an interesting addition to the paper, further motivating the contribution. We thank you for your engagement and believe we will have improved the manuscript as a result of our discussion.

---

### Author Rebuttal · Authors · 2023-08-09

This response is for **reviewer VB9K** and **reviewer SeJL**, who ask about our contribution in light of the BIG bench results. With the below we aim to motivate in further detail why we believe the BIG bench result cannot be built upon in a scientific way. Hence, our work is an important contribution validating the BIG bench result in a reproducible and methodologically sound way, and above that providing insight into what aspects of LLM training are crucial for the ability to do pragmatic inferences.

1. **The methodological approach of the task contributors in BIG bench implicatures raises serious questions as to the validity of their claims.** The reason for this is twofold:
    1. The BIG bench result likely overestimates implicature resolution performance. They show that base LLMs at a certain scale achieve above human average performance on a _subset of our dataset: the unambiguous examples_. This likely overestimates performance on our more challenging _ambiguous_ subset, which we show humans in our study resolve at significantly above chance level (72\%). Hence, the question remains; can LLMs resolve _inherently ambiguous implicatures like humans can_? We find the answer to be “no” for some type of models, and yes for other types, and believe this is an important scientific finding to be shared with the research community.
    2. We believe the human evaluation of the BIG bench task is of low quality (noting that this is impossible to fully verify because there is no information available on how the evaluation was done exactly). The average human evaluator on BIG bench implicatures achieves around 82\% performance (where ours achieves on average 86\% on a more challenging dataset), and their human best rater achieves 100\% (where our human best is 92\%). This difference between human average and best hints at poor quality average rating.
2. **the BIG bench task uses only base LLMs and no state-of-the-art fine-tuning methods**, which are the standard in almost all recent published LLMs and considered crucial to performance (e.g. [1], [2]). So even if we could take their results at face value, a question remains; _what aspects of LLMs contributes to their performance on implicatures?_ In our work we find that implicature performance emerges at a much smaller scale in models instruction fine-tuned at the example level, and that scale and prompting techniques are important.
3. **the BIG bench implicatures contribution is not a scientifically scrutinised and peer-reviewed result, as it is not described and discussed in a peer-reviewed (or any) paper** (the BIG bench tech report does not mention it apart as an entry in a list of tasks), and therefore our result reproducibly validating their result with a sound protocol is—on its own—an important contribution.

We thank both reviewers to raising this point of discussion, and we will make the above clearer in the related work section as well as in Appendix Section H.

[1] “Llama 2: Open Foundation and Fine-Tuned Chat Models”, Touvron et al., 2023

[2] “GPT-4 Technical Report”, OpenAI, 2023

---

### Decision · Program_Chairs · 2023-09-21

**Decision:**

Accept (spotlight)

**Comment:**

All reviewers have agreed that this paper presents a comprehensive evaluation providing interesting insights on the ability of LLMs to resolve conversational implicatures.

The reviewer who gave a lower score was mainly concerned about the paper's novelty because a non-peer-reviewed paper BIG BENCH has investigated the direction. In the authors' rebuttal, they explained why BIG BENCH did not satisfy the goal, and they made an additional effort (human annotation) to better position the problem. I am convinced by the authors about the paper's novelty, and all reviewers agreed that this paper provides informative insights for future LLM investigation. Therefore, I suggest accept this paper.

Given the interesting problem, carefully annotated data, and detailed discussion about conversational implicatures, I suggest this paper be presented as a spotlight paper.